# The histone chaperone SPT2 regulates chromatin structure and function in Metazoa

Giulia Saredi [1] ✉, Francesco N. Carelli[2,3], Stéphane G. M. Rolland[4,16], Giulia Furlan[2,5,13,16], Sandra Piquet[6,16], Alex Appert[2,3,16], Luis Sanchez-Pulido[7,14], Jonathan L. Price[2,5], Pablo Alcon [8], Lisa Lampersberger[2,5,15], Anne-Cécile Déclais [9], Navin B. Ramakrishna [2,5,10], Rachel Toth [1], Thomas Macartney [1], Constance Alabert[9], Chris P. Ponting[7], Sophie E. Polo [6], Eric A. Miska [2,5,11], Anton Gartner[4,12], Julie Ahringer[2,3] & John Rouse [1] ✉

Histone chaperones control nucleosome density and chromatin structure. In yeast, the H3–H4 chaperone Spt2 controls histone deposition at active genes but its roles in metazoan chromatin structure and organismal physiology are not known. Here we identify the *Caenorhabditis elegans* ortholog of SPT2 (CeSPT-2) and show that its ability to bind histones H3–H4 is important for germline development and transgenerational epigenetic gene silencing, and that *spt-2* null mutants display signatures of a global stress response. Genome-wide profiling showed that CeSPT-2 binds to a range of highly expressed genes, and we find that *spt-2* mutants have increased chromatin accessibility at a subset of these loci. We also show that SPT2 influences chromatin structure and controls the levels of soluble and chromatin-bound H3.3 in human cells. Our work reveals roles for SPT2 in controlling chromatin structure and function in Metazoa.

The basic units of chromatin are nucleosomes, consisting of 146 bp of DNA wrapped around a histone octamer comprising a tetramer of the core histones H3 and H4 flanked by two histone H2A–H2B dimers[1]. The density of nucleosomes controls the accessibility of genomic DNA to proteins involved in DNA replication, transcription and DNA repair; local nucleosome density must be altered to facilitate these key processes and restored afterwards[2–4]. A range of proteins in cell nuclei control nucleosome density and composition, including nucleosome remodeling complexes, histone readers, histone-modifying enzymes and histone chaperones. The latter group is a structurally diverse class of proteins that bind histones and regulate nucleosome assembly and composition through a variety of mechanisms. This includes shuttling histones between cytoplasm and nucleoplasm, modulating histone stability and facilitating histone eviction and deposition within nucleosomes[5,6]. Histone chaperones are usually specific for either histones H3–H4, or histone H2A–H2B, and sometimes display specificity for histone variants or posttranslational modifications[5–7]. From a structural perspective, histone chaperones shield functional histone interfaces,

[1]MRC Protein Phosphorylation and Ubiquitylation Unit, School of Life Sciences, University of Dundee, Dundee, UK. [2]Gurdon Institute, University of Cambridge, Cambridge, UK. [3]Department of Genetics, University of Cambridge, Cambridge, UK. [4]IBS Centre for Genomic Integrity at Ulsan National Institute of Science and Technology (UNIST), Ulsan, Republic of Korea. [5]Department of Biochemistry, University of Cambridge, Cambridge, UK. [6]Laboratory of Epigenome Integrity, Epigenetics and Cell Fate Centre, UMR 7216 CNRS - Université Paris Cité, Paris, France. [7]MRC Human Genetics Unit, Institute of Genetics and Cancer, University of Edinburgh, Edinburgh, UK. [8]MRC Laboratory of Molecular Biology, Cambridge, UK. [9]Molecular Cell and Developmental Biology Division, School of Life Sciences, University of Dundee, Dundee, UK. [10]Genome Institute of Singapore (GIS), Agency for Science, Technology and Research (A*STAR), Singapore, Republic of Singapore. [11]Wellcome Sanger Institute, Wellcome Genome Campus, Cambridge, UK. [12]Department of Biological Sciences, Ulsan National Institute of Science and Technology (UNIST), Ulsan, Republic of Korea. [13]Present address: Transine Therapeutics, Babraham Hall, Cambridge, UK. [14]Present address: European Molecular Biology Laboratory, European Bioinformatics Institute (EMBL-EBI), Wellcome Genome Campus, Hinxton, UK. [15]Present address: Maxion Therapeutics, Unity Campus, Cambridge, UK. [16]These authors contributed equally: Stéphane G. M. Rolland, Giulia Furlan, Sandra Piquet, Alex Appert. ✉e-mail: g.saredi@dundee.ac.uk; j.rouse@dundee.ac.uk

such as histone–DNA interaction surfaces and histone dimerization domains, that are otherwise engaged when histones are assembled into nucleosomes[5].

Because of their central role in histone metabolism, histone chaperones play crucial roles in DNA replication, repair and transcription[3,5,8]. During transcription the RNA polymerase II (RNAPII) complex disrupts nucleosomes in the DNA template, thereby 'peeling' DNA from the histone octamer[9–11]. Cryogenic-electron microscopy studies revealed that, on engaging with a nucleosome, RNAPII complexes stall at specific locations along the 'peeled' nucleosomal DNA[11,12]. During this process, the histone surfaces bound by nucleosomal DNA are transiently exposed and recognized by histone chaperones[13–15]. The histone chaperones SPT6, SPT5, ASF1 and the HIRA and FACT chaperone complexes have been implicated in promoting histone disassembly and recycling at active genes[8,16–21]. Active genes are enriched for the histone replacement variant H3.3 (refs. [22–24]), and deposition of H3.3–H4 during transcription is mediated by the HIRA complex[22,23,25]. HIRA promotes both the incorporation of new H3.3–H4 as well as the recycling of parental H3.3–H4 (ref. [18]), in a manner that specifically requires HIRA interaction with the UBN1 or ASF1 histone chaperones[18], respectively. Therefore, the preservation of chromatin structure and nucleosome density during transcription requires multiple histone chaperones with overlapping but nonequivalent functions.

Budding yeast Spt2 is a poorly understood histone chaperone implicated in histone H3–H4 recycling during transcription[26,27]. Spt2 associates with the protein-coding regions of highly expressed genes, in a manner that requires Spt6 (refs. [27,28]). Moreover, loss of Spt2 results in decreased association of H3 with these regions[27]. Yeast Spt2 was shown to bind to cruciform DNA in vitro[29], which is thought to reflect an affinity for crossed DNA helices, reminiscent of DNA at the entry–exit of a nucleosome[30]. Yeast cells lacking Spt2 show an increase in spurious transcription from cryptic intragenic start sites[27,31], and Spt2 mutations that abolish H3–H4 binding recapitulate these defects[26]. Furthermore, Spt2 synergizes with the yeast Hir (HIRA) complex in suppressing spurious transcription[27]. Therefore, Spt2 plays an important role in regulating H3–H4 recycling and chromatin structure in yeast. Little is known, however, about the roles and regulation of SPT2 beyond budding yeast. Chicken (*Gallus gallus*) SPT2 is a nonessential gene, the product of which interacts with RNA polymerase I (RNAPI) and was reported to support RNAPI-mediated transcription, as measured in vitro by nuclear run-on assay[32]. Both the DNA binding and histone binding regions of chicken SPT2 are necessary to support this function[32]. Even though almost nothing is known about SPT2 function in human cells, an X-ray crystal structure of the histone binding domain (HBD) of human SPT2 bound to a H3–H4 tetramer has been reported[26]. Replacing the HBD in yeast Spt2 with the human HBD suppresses cryptic transcription, similar to wild-type (WT) yeast Spt2, but mutating Met641 in the chimeric protein blocks this suppression[26]. These data suggest that human SPT2 can regulate H3–H4 function, at least in yeast, but similar roles in human cells have not yet been described. More recent work has shown by molecular modeling that SPT2 can also co-chaperone a histone H3–H4 dimer together with ASF1A[33].

In this Article, we dissect the in vivo function of SPT2 using both the model organism *Caenorhabditis elegans* and human cells. We combine structural modeling, biochemistry and genetics approaches to characterize how SPT2 binding to histone H3–H4 regulates chromatin structure and function in Metazoa, and we show that worm SPT2 regulates chromatin density at highly expressed genes, transgenerational epigenetic silencing and animal fertility upon heat stress. We also provide evidence that SPT2 regulates chromatin assembly in human cells.

## Results

### A *C. elegans* ortholog of the SPT2 histone chaperone

We set out to test if SPT2 histone binding activity is relevant for chromatin structure and function in Metazoa. The nematode *C. elegans* has proven a valuable system to investigate the role of histone chaperones at the cell and organism level[34–37], and we decided to interrogate a role for SPT2 in this organism first. However, no *C. elegans* ortholog of SPT2 had been reported. Iterative similarity searches revealed the uncharacterized open reading frame *T05A12.3* as a putative ortholog. Multiple sequence alignments defined three evolutionarily conserved regions of the T05A12.3 protein product. The first region spans residues 1–129 (red box) and is conserved in metazoan orthologs but not in budding yeast (Fig. 1a); the second region spans residues 250–276 (yellow box) and is conserved from yeast to humans. The functions of these domains are unknown. The third region, spanning residues 572–661, is the region of highest conservation and corresponds to the HBD found in the human and yeast Spt2 orthologs[26] (Fig. 1a, purple box, and Extended Data Fig. 1a). We used the crystal structure of the human SPT2 HBD[26] as a search template to generate a structural homology model for the corresponding region of T05A12.3, which revealed three points of similarity between the two proteins. First, the tertiary structure of the putative T05A12.3 HBD adopts an arrangement that is similar to the human HBD, which comprises two α-helices (αC1 and αC2) connected by a loop (Fig. 1b). Second, both helices and the loop contact H3–H4 in our model through residues that are conserved. For example, Glu637 and Glu638 in the worm T05A12.3 HDB correspond to Glu651 and Glu652 in the αC2 helix of the human HBD known to be required for H3–H4 binding[26]. Also, Met627 in the T05A12.3 HBD contacts histone H4 in our model; Met627 is the equivalent of Met641 in the human protein that contacts histone H4 and contributes to H3–H4 binding[26] (Fig. 1c and Extended Data Fig. 1b). Third, the most highly conserved residues in each of the two helices and loop lie at the interface with H3–H4 (Fig. 1c). Therefore, structural modeling strongly suggests that worm T05A12.3 is a H3–H4 binding ortholog of SPT2, and we refer to T05A12.3 hereafter as CeSPT-2.

We next tested if CeSPT-2 binds to histones H3–H4 in vitro. To this end we purified His₆-tagged, full-length recombinant CeSPT-2 and

**Fig. 1 | Identification of the *C. elegans* ortholog of SPT2. a**, Left, schematic representation of the three evolutionarily conserved regions found in SPT2 orthologs and in *C. elegans T05A12.3*. N-terminal, central and C-terminal (HBD) regions are shown shaded in red, yellow and violet, respectively. Right, multiple sequence alignments corresponding to the three conserved regions are shown inside colored boxes in red, yellow and violet, respectively. The amino acid coloring scheme indicates the average BLOSUM62 score (correlated to amino acid conservation) in each alignment column: black (greater than 3.5), gray (between 3.5 and 1.5) and light gray (between 1.5 and 0.5). Sequences are named according to their UniProt identifier. Species abbreviations: Q9GYK8_CAEEL, *C. elegans*; SPT2_HUMAN, *Homo sapiens*; SPT2_DROME, *Drosophila melanogaster*; SPT2_YEAST, *Saccharomyces cerevisiae*. **b**, Structural homology model for the putative HBD of *C. elegans* T05A12.3 (blue). This was generated using the crystal structure of the *H. sapiens* SPT2 HBD (yellow) in complex with the H3–H4 tetramer (shaded white) as a search template (PDB code 5BS7

(ref. [26])). The positions of the αC1 and αC2 helices and the connecting loop are shown. **c**, Same as **b**, except that only the CeSPT-2 HBD is shown, color-coded according to the degree of amino acid conservation. **d**, Coomassie gel staining of recombinant full-length HsSPT2 and CeSPT-2 produced in bacteria; one representative experiment of two. **e**, Pull-down of full-length recombinant CeSPT-2 or human HsSPT2 with beads covalently coupled to histone H3–H4 in the presence of 500 mM NaCl. One representative experiment of two is shown. **f**, H3–H4 pull-down with recombinant CeSPT-2 HBD (WT or HBM M627A), or HsSPT2 HBD (WT or HBM M641A). One representative experiment of three is shown. **g**, Coomassie gel staining of recombinant full-length CeSPT-2 (WT and HBM); one representative experiment of two. **h,i**, H3–H4 pull-down with full-length recombinant CeSPT-2 WT and HBM (**h**). Quantification (**i**) of the five independent replicates of the H3–H4 pull-down. *n* = 5, data are represented as mean ± s.d. normalized to WT. One-sided *t*-test with Welch's correction, *P* = 0.0016.

human (Hs) SPT2 (Fig. 1d) and performed a pull-down experiment using recombinant H3–H4 complex covalently coupled to beads. As shown in Fig. 1e, CeSPT-2 binds H3–H4 in vitro similar to HsSPT2, and the isolated putative HBD of CeSPT-2 also bound to H3–H4 (Fig. 1f, WT HBD). We next tested the effect of substituting Met627, the equivalent of Met641 that contacts H4 in the H3–H4 tetramer[26] (Fig. 1c and Extended Data Fig. 1b). Substituting Met627 for Ala (M627A) in the isolated CeSPT-2 HBD reduced, but did not abolish, binding to immobilized H3–H4,

similar to the M641A substitution in the HsSPT2 HBD analyzed in parallel (Fig. 1f). Similar results were obtained using purified full-length CeSPT-2 (Fig. 1g–i). We also found that CeSPT-2 binds to synthetic cruciform DNA (Extended Data Fig. 1c), similar to HsSPT2 (ref. 32) and that the CeSPT-2 M627A substitution had no apparent effect on cruciform DNA binding (Extended Data Fig. 1d). Hereafter, we refer to the histone binding-defective mutation encoding the CeSPT-2 M627A substitution as histone binding mutant (HBM).

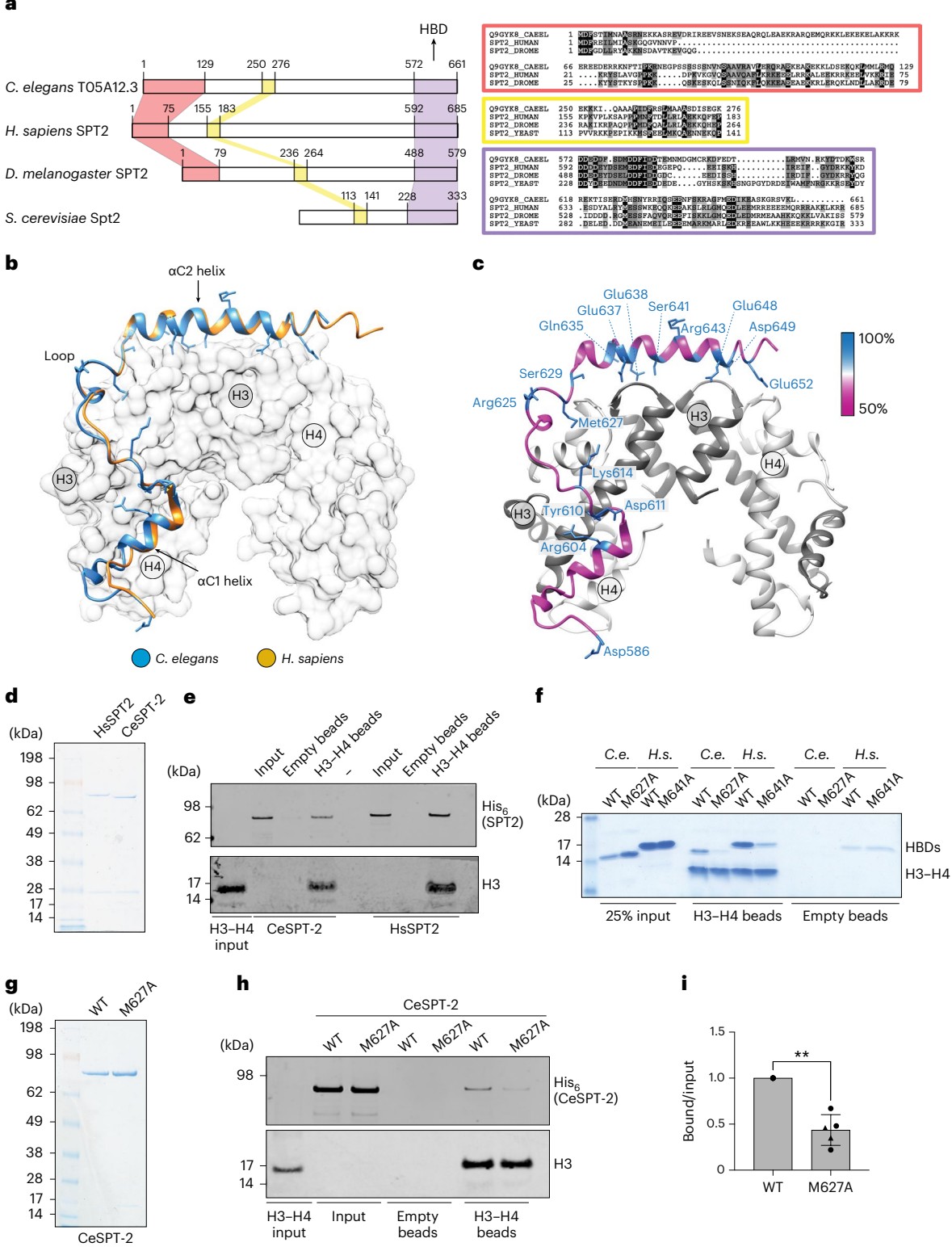

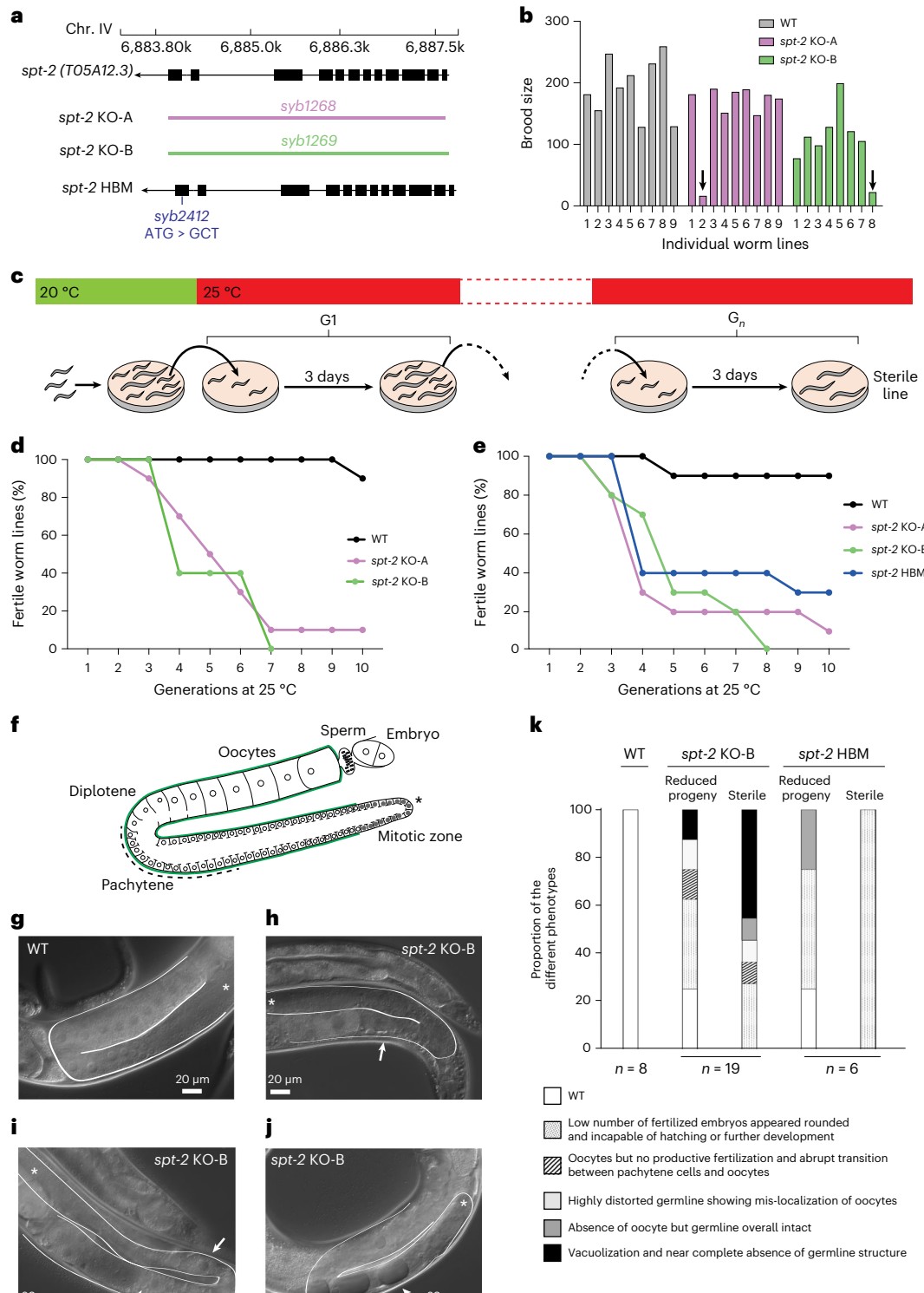

**Fig. 2 | Loss of CeSPT-2 histone binding causes germline defects and sterility.**
**a**, Schematic diagram of the WT, *spt-2*^KO and *spt-2*^HBM alleles. The genomic location of the *spt-2* (*T05A12.3*) gene on chromosome IV is shown. Colored bars indicate the deleted regions in the *spt-2* gene in the KO strains. **b**, The brood size of worms from the indicated strains, grown at 25 °C from the L4 stage, is shown. Arrows indicate worms with low brood size. The number of worms used is indicated in the figure. **c–e**, Transgenerational sterility assay (**c**). Three L4 stage worms of the indicated genotypes were shifted to 25 °C and grown at that temperature for the indicated number of generations. Every generation (3 days), three L3–L4 worms were moved to a new plate. A worm line was considered sterile when no progeny was found on the plate. Ten plates per genotype were used, *n* = 10. The transgenerational sterility assay was performed using WT, *spt-2*^KO-A and *spt-2*^KO-B

worms (**d**), or using WT, *spt-2*^KO-A, *spt-2*^KO-B and *spt-2*^HBM worms (**e**). **f**, Diagram of the *C. elegans* germline (green) and embryos; figure adapted from ref. 38. **g–j**, Single WT and *spt-2*^KO-B L4 worms were shifted to 25 °C and grown at that temperature for the indicated number of generations; every generation (3 days), one L4 worm was moved to a new plate and the number of progeny was assessed, as indicated in Extended Data Fig. 2d,e. Siblings of the worms that, 3 days after the L4 stage, showed reduced or no progeny were subjected to microscopic observation 1 day after the L4 stage. Germline from a WT worm (**g**); germlines from *spt-2*^KO-B worms, with arrows indicating an abrupt transition between pachytene cells and oocytes (**h**), mis-localized oocyte (upper arrow) and abnormally small oocyte (lower arrow) (**i**), and vacuolization (**j**). **k**, Quantification of germline defects; numbers indicated in the figure. G$_n$, number of generations.

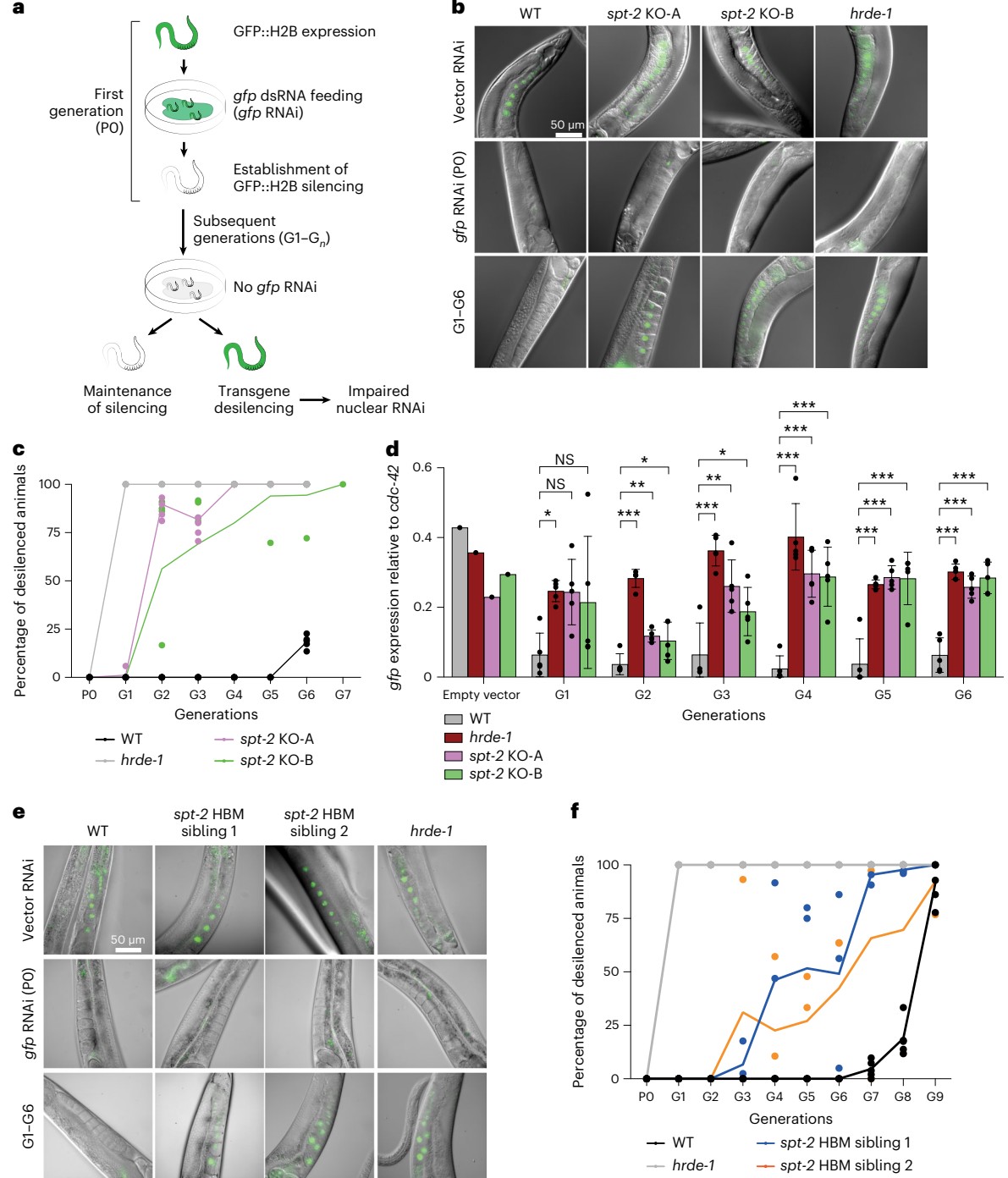

**Fig. 3 | CeSPT-2 histone binding is required for transgenerational gene silencing. a**, Schematic diagram of transgenerational RNAi; see Extended Data Fig. 3 for the transgenerational RNAi assay described previously[41] (figure adapted from ref. [41]). **b**, Representative images of the germline of the *gfp::h2b* reporter worm strains without RNAi (vector RNAi), after *gfp* RNAi treatment and in the subsequent generations after removal of the dsRNA (G1–G6). **c**, Quantification of GFP::H2B expression. Worms showing 'dim' GFP intensity during microscopy inspection were considered as silenced. The graph indicates the mean percentage of desilenced worm lines across the replicate lines, and the percentage of desilenced worms in each replicate is indicated by a dot; *n* = 5. **d**, *gfp::h2b* mRNA expression measured by qPCR, normalized relative to *cdc-42*. *n* = 1 for 'empty vector' datapoint; *n* = 5 for G1–G6; data are represented

as mean ± s.d. Statistical testing: one-way ANOVA, calculated for each generation independently, with Dunnett's post-tests. NS: *P* > 0.05; \**P* < 0.05, \*\**P* < 0.01, \*\*\**P* < 0.0001. One-way ANOVA results, G1: 0.0588, G2: 4.34 × 10⁻⁸, G3: 1.19 × 10⁻⁴, G4: 4.22 × 10⁻⁶, G5: 3.71 × 10⁻⁶, G6: 9.75 × 10⁻⁸. Exact *P* values of Dunnett's post-test comparisons shown in the figure are reported in Supplementary Table 3. **e**, Representative microscopy images of the germline of the indicated worm strains without RNAi (vector RNAi), after *gfp* RNAi treatment and in the subsequent generations after removal of the dsRNA (G1–G9). **f**, Quantification of GFP::H2B expression, as in **c**. The graph indicates the mean percentage of desilenced worm lines across the replicate lines, and the percentage of desilenced worms in each replicate is indicated by a dot; *n* = 5 (WT and *hrde-1* mutant); *n* = 3 (for each of the *spt-2*^HBM siblings).

We expected CeSPT-2 to be a nuclear protein given that it binds H3–H4 and DNA. Analysis of a worm strain in which green fluorescent protein (GFP) was inserted at the 5′ end of the *T05A12.3* gene showed that GFP-tagged CeSPT-2 is a widely expressed protein found, for example, in the head, germline, hypodermis, intestine and vulva cell nuclei (Extended Data Fig. 1e, upper row). Moreover, knock-in of the HBM mutation did not affect GFP::CeSPT-2 expression or localization (Extended Data Fig. 1e, lower row). Taken together, the data above indicate that CeSPT-2 is a widely expressed, nuclear protein that appears to be the ortholog of the SPT2 histone H3–H4 chaperone.

## CeSPT-2 deficiency causes temperature-dependent sterility

To study the impact of CeSPT-2 on worm development and fertility, two independently derived *spt-2* null strains, with the open reading frame being eliminated, were constructed, hereafter referred to as *spt-2*[KO-A] and *spt-2*[KO-B] (KO, knockout; Fig. 2a and Extended Data Fig. 2a). The *spt-2* null strains were viable, and their progeny size was comparable to WT under standard growth conditions (20 °C, Extended Data Fig. 2b). However, when worms were grown at 25 °C for one generation, we noticed that a low proportion of worms produced far fewer progeny than WT (Fig. 2b and Supplementary Table 1). We next investigated if this apparent fertility defect became more pronounced in subsequent generations (Fig. 2c). As shown in Fig. 2d,e (and Extended Data Fig. 2e), the proportion of sterile *spt-2* null worms increased progressively with each generation, so that after approximately ten generations at 25 °C very few or no fertile worms remained. We also generated a knock-in worm strain harboring the HBM mutation that reduces CeSPT-2 binding to H3–H4 (*spt-2*[HBM]) (Fig. 2a). The *spt-2*[HBM] worms also showed an increased incidence of sterility when grown at 25 °C for several generations (Fig. 2e and Extended Data Fig. 2c,e), although to a lesser extent than the *spt-2* null strains probably because CeSPT-2 HBM shows residual binding to H3–H4 (Fig. 1f,h). To be certain that the sterility observed in the *spt-2*[HBM] strain is a direct consequence of M627A mutation, we reverted the Ala627 HBM mutation to WT (Met627). The resulting strain (*spt-2*[HBM] A627M) lost the sterility phenotype associated with *spt-2*[HBM], indicating that the sterility is due to loss of CeSPT-2 histone binding capacity (Extended Data Fig. 2c). Of note, *gfp::spt-2* worms do not show increased temperature-dependent sterility as compared to WT worms (Extended Data Fig. 2e), indicating that GFP-tagged CeSPT-2 is functional.

To investigate the morphology of the germline in *spt-2*-defective worms, we grew *spt-2* mutant worms for several generations at 25 °C and subjected the siblings of worms that either showed a reduced number of progeny or were sterile to microscopic analysis (Extended Data Fig. 2d,e). In WT nematode gonads, germ cells transit in an orderly manner from the mitotic stem cell state to the various stages of meiotic prophase (pachytene and diplotene) and eventually become fully mature oocytes[38] (Fig. 2f,g). In contrast, the germline of *spt-2* mutant worms, whose siblings had a decreased progeny size or were sterile,

shows a wide array of defects at 25 °C, including the following: oocytes but no productive fertilization, and abrupt transition between pachytene cells and oocytes (Fig. 2h); a highly distorted germline showing mis-localization of oocytes (Fig. 2i) and vacuolization (Fig. 2j). Taken together, these data show that the onset of sterility in *spt-2*-defective worms is associated with pleiotropic defects in germline development (Fig. 2k).

## CeSPT-2 is required for transgenerational gene silencing

The sterility seen in *spt-2*-defective worms after several generations at elevated temperature was reminiscent of worms harboring defects in nuclear RNA interference (RNAi)[39,40]. Nuclear RNAi is a pathway in which small double-stranded (ds)RNAs trigger heritable gene silencing[41,42]. The establishment of silencing involves dsRNA-mediated dicing of the target messenger RNA, while the maintenance and transgenerational inheritance of the silenced state involves the RNA-dependent synthesis of secondary small interfering (si)RNAs and changes in chromatin state in the RNAi target gene(s), such as histone H3 methylation at Lys9 and Lys23 (refs. 43–46). In prevailing models, the Argonaute family protein HRDE-1 binds secondary small RNAs and is required for the transgenerational inheritance of gene silencing after the dsRNA trigger has been removed[41,42].

We investigated a role for CeSPT-2 in the nuclear RNAi pathway, using a reporter strain we described previously[41]. In this system, a *gfp::h2b* single-copy transgene that is constitutively expressed in the worm germline can be silenced by feeding worms with bacteria expressing double-stranded *gfp* RNA (*gfp* RNAi)[41] (Fig. 3a and Extended Data Fig. 3). Analysis of GFP fluorescence showed that silencing of the *gfp::h2b* reporter transgene occurs normally in *spt-2* null and *hrde-1*-defective worms grown on bacteria expressing the *gfp* RNAi (P0; Fig. 3b,c). After removing the bacteria, silencing was maintained for five generations in the WT worms, but the transgene was desilenced in the first generation (G1) of *hrde-1*-defective worms. Strikingly, the reporter transgene was robustly desilenced in both of the *spt-2* null strains from the second generation (G2) after the removal of the *gfp* RNAi, as measured by GFP fluorescence (Fig. 3b,c) and mRNA abundance (quantitative polymerase chain reaction, qPCR; Fig. 3d). We also found that the *spt-2*[HBM] strain showed transgene desilencing albeit slightly later than in the null strains (Fig. 3e,f). Taken together, these data show that CeSPT-2 histone binding is required for the transgenerational inheritance of epigenetic gene silencing.

## CeSPT-2 controls chromatin accessibility of its target genes

To gain insight into the genomic regions that might be regulated by CeSPT-2, we sought to identify its chromatin occupancy genome wide. We performed chromatin immunoprecipitation and sequencing (ChIP–seq) in synchronized adult worms expressing endogenously tagged GFP::CeSPT-2, using WT worms as control for antibody specificity.

**Fig. 4 | CeSPT-2 histone binding controls chromatin accessibility.**
**a**, Distribution of GFP::CeSPT-2 ChIP–seq peaks in different genome locations. **b**, Metagene profile of GFP::CeSPT-2 occupancy levels within the coding region of its target genes, compared to non CeSPT-2 target genes. **c**, GFP::CeSPT-2 chromatin occupancy regions were divided in quartiles, and gene expression levels (measured in synchronized WT adult worms) were plotted in each quartile, and compared to CeSPT-2 nontarget genes. The box represents the interquartile range, the whiskers the minimum and maximum (excluding outliers). *n*, nontargets: 20,117; first quartile: 545; second quartile: 544; third quartile: 544; fourth quartile: 544. Statistical differences: Benjamini–Hochberg-corrected pairwise Mann–Whitney test. ***$P < 10^{-16}$. **d**, Chromatin accessibility at CeSPT-2 target genes in *spt-2*[KO-A] and *spt-2*[HBM] adult worms, measured by ATAC–seq and expressed in reads per million. TSS, transcription start site; TTS, transcription termination site. Two independent replicates of the ATAC–seq experiment were performed. **e**, Example of ATAC–seq and GFP::CeSPT-2 ChIP–seq tracks. The ATAC–seq signal across the gene body of *ddo-2* in WT, *spt-2*[KO-A] and *spt-2*[HBM]

worms is indicated, with the GFP::CeSPT-2 binding profile shown in the bottom profile. **f**, Fraction of genes with significantly increased accessibility in *spt-2*[KO-A] (dark gray) among CeSPT-2 nontarget and target genes. CeSPT-2 targets were divided in quartiles based on their CeSPT-2 enrichment. Statistical difference between CeSPT-2 targets and nontargets was measured by a chi-squared test. Benjamini–Hochberg corrected *P* values. *P*, nontargets versus first quartile: 0.0513; nontargets versus second quartile: 0.0035; nontargets versus third quartile: <10⁻¹⁶; nontargets versus fourth quartile: <10⁻¹⁶. **g,h**, Volcano plot of gene expression levels of CeSPT-2 target genes (**g**) and nontargets (**h**) in *spt-2* null versus WT worms. Red points indicate genes with FDR <0.001 and $\log_2$ fold change >1 or <−1. In Extended Data Fig. 4e: validation by qPCR using independently collected RNA. **i**, GO analysis of the genes upregulated in the indicated *spt-2* mutants. KO_UP, genes upregulated in spt-2 KO worms. 'p.adjust' values indicate Benjamini–Hochberg *P* values obtained through an hypergeometric test.

Around 88% of the regions enriched for CeSPT-2 binding lie within genic regions (5,299/6,003 sites), with the remaining sites found in intergenic (~7%, 258 sites) or repetitive (~4%, 446 sites) sequences (Fig. 4a). GFP::CeSPT-2 enrichment was observed over the entire length of what

we hereby call 'CeSPT-2 target genes' (2,177, Supplementary Table 2), with an apparent enrichment for the 3′ end of the gene (Fig. 4b). We found that CeSPT-2 target genes are highly expressed, and their expression positively correlates with CeSPT-2 binding (Fig. 4c).

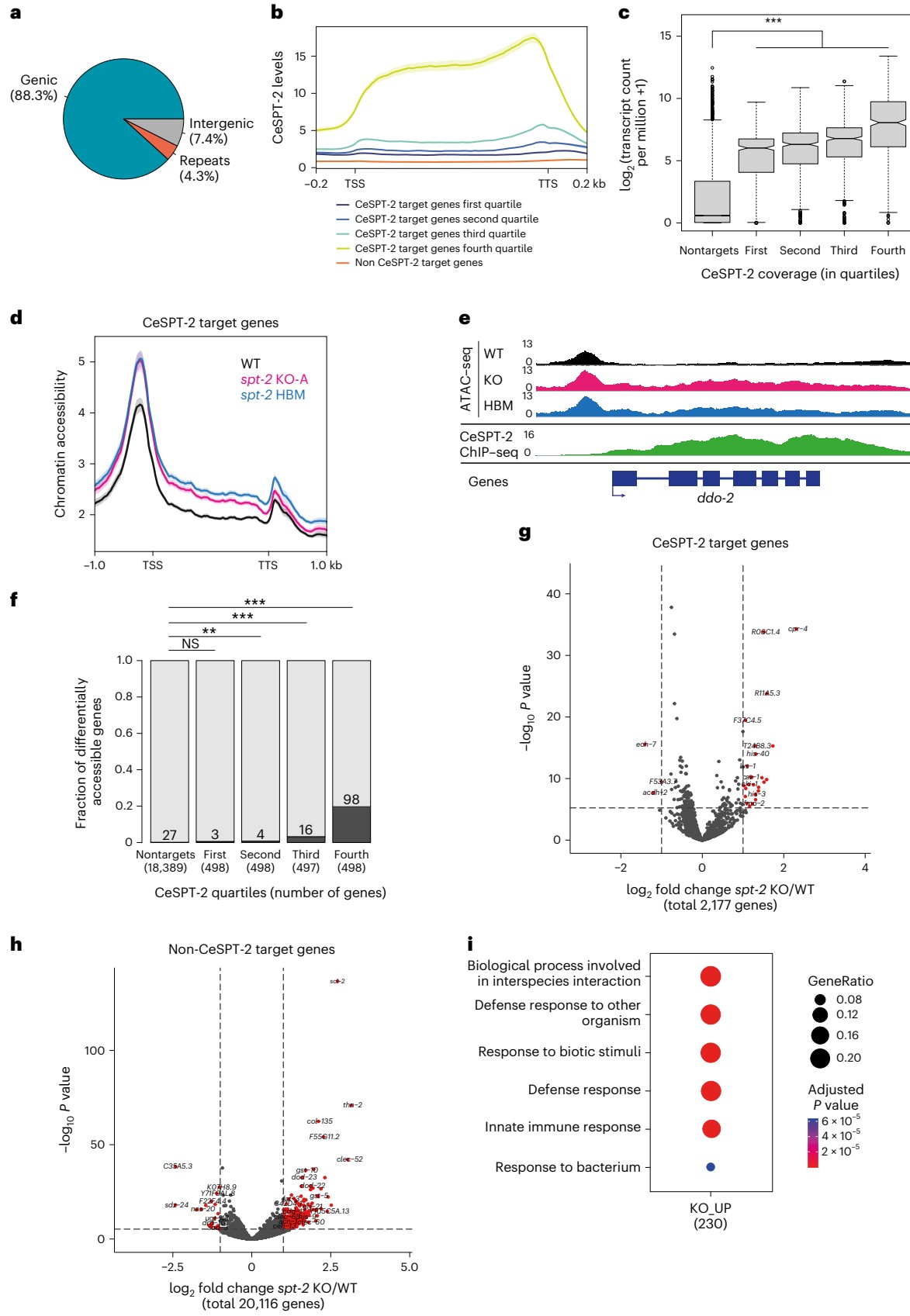

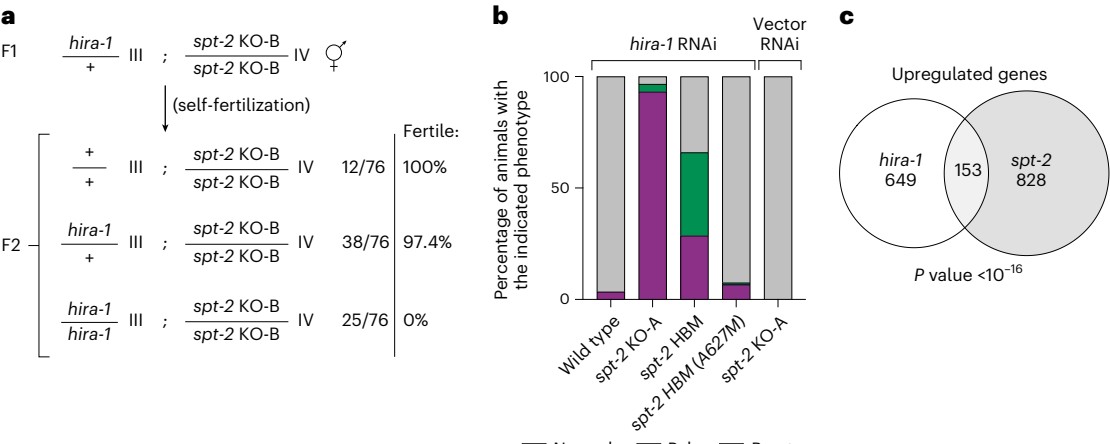

**Fig. 5 | Interplay between SPT2 and HIRA histone chaperones. a**, *spt-2*[KO-B] worms were crossed with *hira-1*/+ heterozygote worms; the full cross strategy is outlined in Extended Data Fig. 5. The numbers of progeny of *hira-1*/+;*spt-2*[KO-B] worms of each indicated genotypes are shown, together with the percentage of fertile worms. **b**, Percentage of burst worms or worms showing a protruding vulva (Pvl) phenotype on treatment with *hira-1* RNAi, or empty vector RNAi, as indicated. Forty worms were scored per replicate, and three independent replicates of the experiment were performed. **c**, Venn diagram showing the overlap between upregulated genes in *spt-2* null and *hira-1* null worms. Hypergeometric test; enrichment: 3.085, *P* value <10[-16].

Conversely, when we analyzed CeSPT-2 enrichment in the top 5% most highly expressed genes, we found that 74.8% of these genes (833/1,114) are CeSPT-2 targets (Supplementary Table 2).

To investigate whether CeSPT-2 and its histone H3–H4 binding activity affect chromatin accessibility of its target genes, we performed assay for transposase-accessible chromatin with sequencing (ATAC–seq) in WT, *spt-2*[KO-A] and *spt-2*[HBM] adult worms. In both null and HBM *spt-2* mutants, we found that chromatin accessibility increased across the entire length of the gene body of a subset of CeSPT-2 bound genes (KO: 121/1,991, HBM: 105/1,991, overlap: 88; Fig. 4d–f and Supplementary Table 2). In contrast, no obvious increase in chromatin accessibility was observed at genomic regions that are not enriched for CeSPT-2 (Extended Data Fig. 4a). Notably, most genes showing a significant increase in their accessibility in *spt-2* mutants had high levels of CeSPT-2 binding (Fig. 4f). Differentially accessible (DA) genes in *spt-2*[KO] worms are highly transcribed (Extended Data Fig. 4b) and are enriched for 'metabolic processes' and 'nucleosome assembly' Gene Ontology (GO) terms (Extended Data Fig. 4c). Together, we conclude that CeSPT-2 binds and influence the chromatin architecture of some of the most highly transcribed genes in *C. elegans*. The finding that chromatin accessibility is affected in both *spt-2*[KO-A] and *spt-2*[HBM] mutants indicates that histone binding activity is needed for chromatin accessibility regulation.

To evaluate a potential role for CeSPT-2 in gene expression, we performed mRNA-seq of *spt-2*[KO-A] and *spt-2*[HBM] mutant adult worms. We observed 605 upregulated and 40 downregulated protein-coding genes in *spt-2* null worms compared to WT, but only 30 up- and 3 downregulated protein-coding genes in the *spt-2*[HBM] mutant (Fig. 4g,h, Supplementary Table 2 and Extended Data Fig. 4d,e). Of the 605 genes upregulated in the null mutant, only 40 were CeSPT-2 targets, suggesting that most gene expression alterations are indirect (Fig. 4g,h). The upregulated genes were strongly enriched for GO terms related to 'defense response' (Fig. 4i), indicating that loss of *spt-2* leads to the activation of a stress response, similar to what was observed in response to pathogens as well as ultraviolet and ionizing radiation[47,48]. Of note, the expression profiles of genes involved in the RNAi pathway is largely unaffected (Extended Data Fig. 4f), and it is therefore unlikely that CeSPT-2 affects *gfp::h2b* silencing (Fig. 3b–f) via decreased expression of RNAi effectors. Taken together, these data show that CeSPT-2 binds to a range of highly expressed target genes, and its histone binding activity controls chromatin accessibility at these loci; loss of *spt-2* results in a global stress response.

**Interplay between the CeSPT-2 and HIRA-1 histone chaperones**
Given the reported roles of the HIRA complex in preserving chromatin structure at active genes[18,22,23], we examined whether CeSPT-2 and HIRA-1 may overlap functionally. *hira-1* null worm strains are viable but show pleiotropic defects, including low brood size and morphological abnormalities (such as small pale bodies and protruding vulvas 'Pvl')[35,36]. We set out to test the impact of the concomitant loss of CeSPT-2 and HIRA-1 on worm development and viability. First, we crossed *spt-2*[KO-B] and *hira-1* mutant[36] worms and we observed that *hira-1;spt-2* double-null worms are born at Mendelian ratios, but were completely sterile (Fig. 5a and Extended Data Fig. 5). In a complementary experiment, we depleted

**Fig. 6 | SPT2 binding to H3–H4 regulates the levels of soluble and chromatin-bound H3.3 in human cells. a**, Distribution of HsSPT2 between soluble and chromatin fractions of U-2 OS cells; one representative experiment of two. **b**, GFP–HsSPT2 binding to chromatin in fixed or pre-extracted U-2 OS cells assessed by high-content microscopy. Graph shows the mean GFP intensity normalized to the fixed sample; *n* = 2, data are represented as mean of two independent experiments. Right, representative images. **c**, Schematics of the SNAP-tag assay for new histone H3.3 incorporation. **d**, Western blot showing SPT2 levels in WT, SPT2 KO-3 and KO-4 clones and in the rescued cell lines after reintroduction of untagged SPT2 WT or HBM; one representative experiment of two. **e,f**, Fluorescence intensity of new TMR-labeled SNAP-H3.3 in cell nuclei of the indicated U-2 OS SNAP-H3.3 cell lines after direct fixation (**e**) or pre-extraction (**f**). Data from each one of the four independent replicates is indicated with a different symbol, showing the reproducible trend of SPT2[WT] rescuing new H3.3-SNAP levels to a higher degree than SPT2[HBM]. Data are represented as mean ± s.d. from four independent experiments and normalized to WT. Statistical testing: one-way ANOVA with Dunnett's post-test. NS, *P* > 0.05; *\*P* < 0.05, \*\**P* < 0.01, \*\*\**P* < 0.0001. One-way ANOVA *P* value results: KO-3 direct fixation, 0.0164; KO-4 direct fixation, 0.0510; KO-3 pre-extraction, 0.0017; KO-4 pre-extraction, 0.0139. Exact *P* values of Dunnett's post-test comparisons are reported in Supplementary Table 3. **g**, Chromatin accessibility levels at DA genes in WT and SPT2 KO cells, measured by ATAC–seq and expressed in reads per million. The chromatin accessibility profile of a control dataset of 100 random, non-DA genes is shown in Extended Data Fig. 6m. **h**, Example of ATAC–seq tracks on histone genes in WT and SPT2 KO cells.

HIRA-1 by RNAi from WT, *spt-2* null or *spt-2*[HBM] worms grown at 25 °C. We noted that the Pvl phenotype, reported in *hira-1* null worms[36], was not observed in WT worms treated with *hira-1* RNAi (Fig. 5b), possibly due to partial HIRA-1 depletion. However, HIRA-1 depletion in *spt-2*

null or *spt-2*[HBM] worms resulted in a dramatic increase in the incidence of vulvar protrusions and of burst nematodes (Fig. 5b). The morphological defects of the *spt-2*[HBM] worms were fully rescued in the reverted (A627M) strain (Fig. 5b). Comparison of the transcriptomes of *spt-2* null

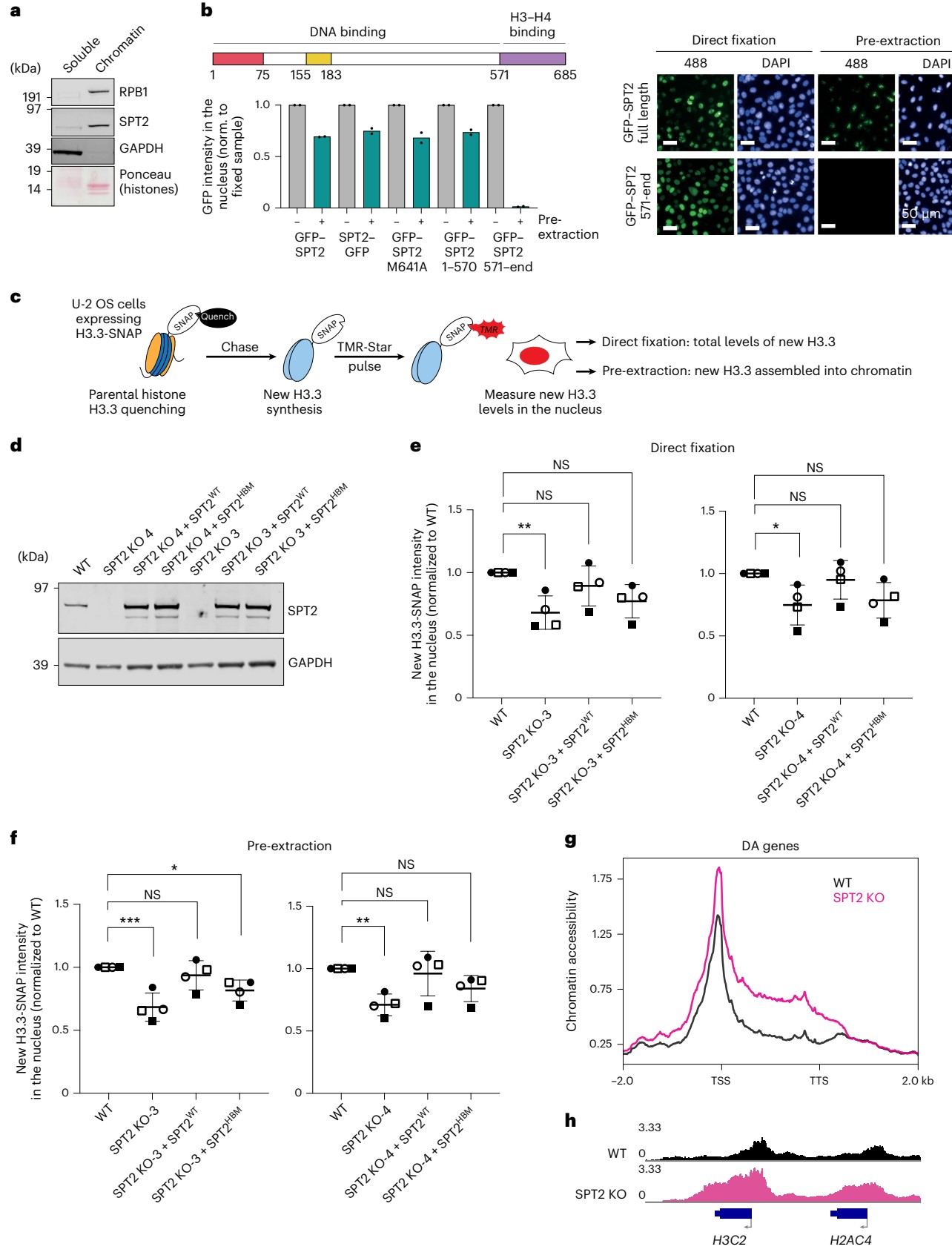

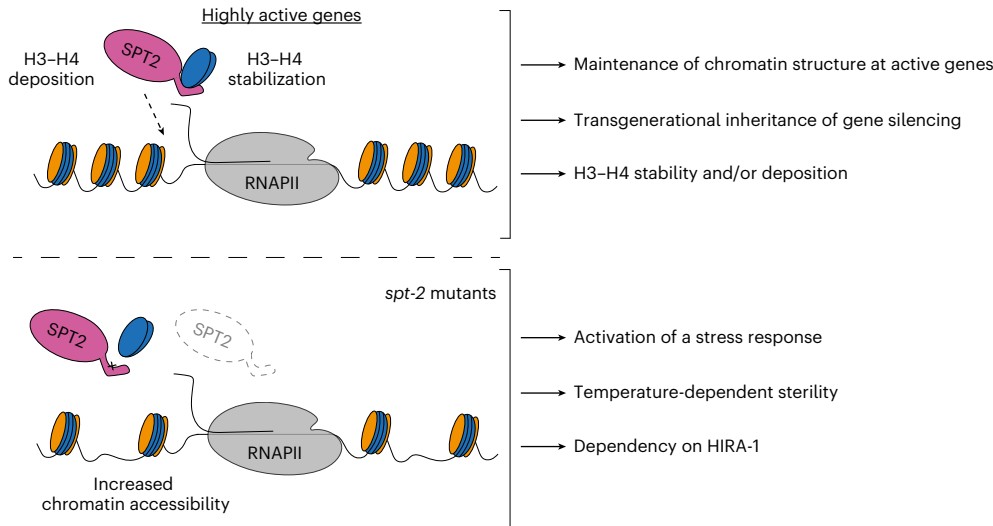

**Fig. 7 | SPT2 regulates chromatin structure and organism fitness.** SPT2 localizes to highly active genes where it promotes the maintenance of chromatin structure via its histone H3–H4 binding capacity. Here, it may act via stabilizing new histone H3.3 before their incorporation, as well as via directly promoting their incorporation into chromatin. In worms, the interaction of CeSPT-2 with H3–H4 is required for the transgenerational inheritance of gene silencing and to promote the integrity of the germline under heat stress conditions. Loss of CeSPT-2 in worms activates a stress response, potentially due to aberrant transcription, and makes worms reliant on the HIRA-1 histone chaperone.

(this study) and *hira-1* null worms (dataset from ref. 35) revealed a statistically significant overlap of the genes upregulated in both mutants (Fig. 5c). These data point to functional overlap between CeSPT-2 and HIRA-1 in worms.

## Impact of SPT2 on chromatin structure in human cells

The data above show clear roles for CeSPT-2 activity in controlling chromatin structure in worms, and we next explored a potential role in human cells. Affinity-purified antibodies raised against HsSPT2 recognized a band of approximately 75 kDa in extracts of U-2 OS cells, that was reduced in intensity by three different HsSPT2-specific siRNAs (Extended Data Fig. 6a). Fractionation experiments showed that endogenous HsSPT2 is strongly enriched in the chromatin fraction of U-2 OS cells (Fig. 6a). Furthermore, fluorescence analysis of U-2 OS cells after pre-extraction of soluble proteins showed that HsSPT2 tagged with GFP at either the N-terminus or C-terminus associates with chromatin (Fig. 6b). Full-length HsSPT2 bearing an M641A mutation that reduces histone binding, or an HsSPT2 deletion fragment (amino acids (a.a.) 1–570) lacking the HBD, bound chromatin similar to full-length, WT HsSPT2, indicating that interaction with H3–H4 is dispensable for chromatin association (Fig. 6b). Consistent with this idea, a deletion fragment corresponding to the HBD alone (a.a. 571–end) localized to the nucleus but was not retained on chromatin (Fig. 6b). Biochemical fractionation recapitulated the above observations (Extended Data Fig. 6b,c).

We then tested whether HsSPT2 influences chromatin structure and deposition of histone H3.3, which is known to be enriched at active genes[23]. We employed a U-2 OS reporter cell line stably expressing a SNAP-tagged H3.3 (ref. 49) that can be covalently labeled in vivo using cell-permeable compounds to distinguish between pre-existing and newly synthesized protein pools (Fig. 6c). New H3.3 incorporation into chromatin in these cells is reduced after depletion of HIRA (Extended Data Fig. 6d), as previously shown[18,23]. As shown in Extended Data Fig. 6d,e, siRNA-mediated depletion of HsSPT2 with three different siRNAs reduces the amount of new H3.3 incorporated into chromatin (pre-extraction, Extended Data Fig. 6d), with a slight albeit nonsignificant reduction in total H3.3 levels also observed (direct fixation, Extended Data Fig. 6e). We note that HsSPT2 knockdown does not affect HIRA stability, nor does it impair HIRA binding to chromatin (Extended

Data Fig. 6f,g). Moreover, H3.3-SNAP mRNA levels in HsSPT2-depleted cells are comparable to control cells (Extended Data Fig. 6h).

To investigate these phenotypes further, we disrupted the *SPTY2D1* gene in the H3.3-SNAP U-2 OS reporter cells, to obtain three independent SPT2 KO clones (Extended Data Fig. 6i). Lentiviruses were used to re-introduce untagged SPT2 (WT or a HBM allele, HBM: M641A, E651A, E652A[26]; Fig. 6d) into clones KO-3 and KO-4. We found that the total amount of new H3.3-SNAP, and the amount of new H3.3 in chromatin, is reduced in SPT2 KO cells (Fig. 6e,f and Extended Data Fig. 6j,k). Re-expression of SPT2[WT] restores the level of total and chromatin-associated H3.3-SNAP to WT levels, while expression of SPT2[HBM] results in a lower level of rescue (Fig. 6e,f and Extended Data Fig. 6j,k). The impact of SPT2 on new H3.3 levels does not correlate with significant changes in H3.3-SNAP mRNA as measured by qPCR (Extended Data Fig. 6l). From these data, we conclude that human SPT2 controls the levels of H3.3 histones and their incorporation into chromatin through its histone binding activity. We note that U-2 OS cells carry mutations in the DAXX–ATRX histone H3.3 deposition complex[22,50,51]. DAXX–ATRX has mainly been implicated in histone deposition in heterochromatin regions[22,50,51] though, and mutations in DAXX or ATRX are unlikely to influence how SPT2 regulates chromatin assembly at active genes; this, however, remains to be formally proven.

Last, we performed ATAC–seq to investigate chromatin accessibility levels in SPT2 KO U-2 OS cells. Our analysis identified 84 genes that are DA in SPT2 KO cells (false discovery rate (FDR) >0.05), with 82 of these genes showing increased accessibility. Genes that were DA in SPT2 KO cells showed increased accessibility over the entire gene body (Fig. 6g), as we observed in *spt-2* mutant worms (Fig. 4d). Intriguingly, we noticed that DA genes were enriched for histone genes, with 29 out of 84 (34.5%) histone genes being more accessible in SPT2 KO cells (Fig. 6h and Supplementary Table 2). Together, these data show that SPT2 is required for chromatin assembly and maintenance in human cells.

## Discussion

In this study, we demonstrate that SPT2 controls chromatin structure and function in Metazoa. Through bioinformatic analyses and structural modeling we identified the previously unannotated protein T05A12.3 as the worm ortholog of CeSPT-2 (Fig. 1a–c), and we found that recombinant CeSPT-2 binds H3–H4 in vitro in a manner similar

to the yeast and human orthologs (Fig. 1d–i). Chromatin profiling revealed that CeSPT-2 targets a range of genomic sites in worms, most (~88%) lying within genic regions (Fig. 4a,b). Consistent with what has been observed in yeast[27], genes enriched for CeSPT-2 are among the mostly highly active genes in worms (Fig. 4c), possibly reflecting a propensity of CeSPT-2 to bind accessible chromatin regions. A subset of CeSPT-2 target genes showed increased chromatin accessibility over the entire bodies of these genes in spt-2 null and spt-2[HBM] worms (Fig. 4d,e), with a direct correlation between CeSPT-2 binding levels and extent of increased accessibility (Fig. 4f); therefore, CeSPT-2 activity helps limit the accessibility levels of chromatin in highly active genomic regions. The expression of most of the CeSPT-2 target genes was unaffected in the spt-2 mutant worms (Fig. 4g). What, then, is the role of CeSPT-2 in limiting chromatin accessibility at these genes? Perhaps CeSPT-2 activity prevents harmful consequences of an excessively open chromatin such as spurious transcription, as observed for yeast Spt2. This might lead to the production of aberrant transcripts resembling foreign nucleic acids[52]. While chromatin accessibility levels were comparable between spt-2 KO and spt-2 HBM worms (Fig. 4d,e), spt-2 KO worms mounted a more dramatic transcriptional upregulation response than spt-2 HBM worms (Fig. 4h and Extended Data Fig. 4d), which suggests that histone-independent functions of CeSPT-2 may contribute to transcriptional regulation. These possibilities will be interesting to investigate.

The idea that CeSPT-2 binding to H3–H4 restricts chromatin accessibility at highly expressed genes is supported by our demonstration that RNAi-mediated transgenerational gene silencing of a gfp::h2b reporter transgene requires CeSPT-2 histone binding activity (Fig. 3a–f). This finding, to our knowledge, identifies CeSPT-2 as the first histone chaperone required for nuclear RNAi in C. elegans. How chromatin structure controls the transgenerational inheritance of gfp::h2b silencing is unclear[53]: while histone posttranslational modifications associated with silenced chromatin are observed at the RNAi target locus in response to dsRNA[43–46,54], the SET-25 and SET-32 H3K9/K23 methyltransferases are only required for maintenance of silencing in the first generation after removal of the dsRNA trigger, and are dispensable afterwards[43,55]. We speculate that CeSPT-2 is recruited to the open chromatin of the active transgene, where it can control silencing of the gfp::h2b reporter by limiting chromatin accessibility, as shown for the endogenous CeSPT-2 target genes (Fig. 4d,f), and/or preserve repressive histone H3 PTMs associated with RNAi-mediated gene silencing.

Besides SPT2, the histone chaperones FACT, SPT6, SPT5, ASF1 and HIRA have all been implicated in preserving chromatin structure at active genes[8,16–21]. One possible scenario to explain the multiplicity of chaperones acting at transcribed genes is that SPT2 could work together with the other histone chaperones, by binding different intermediates of histone H3–H4 in a relay during nucleosome assembly and disassembly. In an alternative scenario, not incompatible with the previous one, specific subsets of genes may be particularly reliant on the joint functions of SPT2 together with other histone chaperones—highly expressed genes, for example, where histone turnover is highest[56–59]. Yet again, SPT2 activity could become essential in response to stress or stimuli, and this may underlie the heat stress-dependent sterility observed in spt-2 mutant worms (Fig. 2b–e). Concomitant loss of CeSPT-2 and HIRA-1 leads to synthetic lethality in worms (Fig. 5a) and the synthetic defects depend on CeSPT-2 binding to H3–H4 (Fig. 5b). This is in accordance with the observation, in budding yeast, that compound mutants of SPT2 and HIR complex genes have synthetic growth defects[27]. How SPT2 fits mechanistically in the wider network[5] of histone chaperones is still unclear. We speculate that, at active genes, SPT2 could function both in the recycling of 'old' H3.3–H4, as well as in the deposition of new H3.3–H4, together with ASF1 and HIRA[18,20,23,25,26,33]. The chaperoning of new H3.3 histones by SPT2 may also protect histones from degradation thus maintaining new H3.3 levels. Proteomic analysis of the histone chaperone network[5] in cells lacking SPT2 will

potentially shed light into SPT2 function and its interaction with other histone chaperones.

We made the important finding that, in human U-2 OS cells, depleting or deleting SPT2, or weakening SPT2 histone binding capacity, reduces the total levels of newly synthesized H3.3 (Fig. 6c–f and Extended Data Fig. 6d,e), suggesting that the ability of SPT2 to bind H3–H4 may influence the stability of these histones. In principle, the reduction in total H3.3 could account for the decrease in H3.3 chromatin incorporation in SPT2-deficient cells, without needing to invoke a role for SPT2 in the active deposition of H3.3–H4 into nucleosomes. By way of comparison, however, it is interesting to note that mutations in histone H3 (H3 R63A, K64A) that impair binding to the MCM2 histone chaperone affect total levels of SNAP-tagged H3.1 without obviously decreasing H3.1 incorporation[60]. Therefore, it is still possible SPT2 participates in loading H3–H4 into chromatin as well as influencing total histone levels. We note that SPT2 has been identified as interactor of both H3.1 and H3.3 in human cells[33] and it could therefore have a broader function in stabilizing and/or incorporating histone H3 in a variant-independent manner.

Faithful maintenance of chromatin structure is essential to safeguard epigenetic information and cell identity, and to preserve genome stability and organism viability (Fig. 7). We show the importance of CeSPT-2 in maintaining germline fertility under heat stress conditions, and its role in preserving chromatin structure at highly active genes and ensuring the transgenerational inheritance of gene silencing. Together, our work highlights the importance of understanding how the concerted action of histone chaperones come together to preserve chromatin structure and organism fitness.

## Online content

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

## Methods

### Computational protein sequence analysis

Multiple sequence alignments were generated with the program T-Coffee using default parameters[61], slightly refined manually and visualized with the Belvu program[62]. Profiles of the SPT2 evolutionarily conserved regions as hidden Markov models (HMMs) were generated using HMMer. Profile-based sequence searches were performed against the Uniref50 protein sequence database[63] using HMMsearch[64,65].

### Structural modeling and conservation analysis

The crystal structure of human SPT2 (Protein Data Bank (PDB) code 5BS7)[26] was used as a search template to generate a structural homology model for *C. elegans* SPT2 using the homology-model server SWISS-MODEL (https://swissmodel.expasy.org/)[66,67]. A multiple protein sequence alignment was generated by MUSCLE[68,69]. UCSF Chimera[70] was used to align the atomic models of human and *C. elegans* set using the 'Match-Maker' function. Protein sequence conservation was mapped onto the structural model and figures generated in Chimera.

### Recombinant protein purification

All recombinant SPT2 proteins were expressed in Rosetta 2(DE3) pLysS BL21 *Escherichia coli* bacteria (Novagen, 71401). Plasmids were transformed into Rosetta BL21 cells according to the manufacturer's instructions, and transformed bacteria were grown in Luria-Bertani (LB) medium supplemented with 35 µg ml$^{-1}$ chloramphenicol and 50 µg ml$^{-1}$ of kanamycin.

### Full-length CeSPT-2 (WT and HBM) and HsSPT2 (His$_6$-tagged).

Expression of 2xFlag–SUMO–CeSPT-2–His$_6$ or of 2xFlag–SUMO–HsSPT2–His$_6$ was induced by growing bacteria overnight at 20 °C with 200 µM isopropyl β-ᴅ-1-thiogalactopyranoside (IPTG). After cell lysis (lysis buffer: 50 mM Tris–HCl pH 8.1, 500 mM NaCl, 10 mM imidazole pH 8, 10% glycerol and 0.1 mM tris(2-carboxyethyl)phosphine (TCEP) supplemented with protease inhibitors), lysates were incubated with NiNTA beads for 1 h rotating top-down at 4 °C, and subsequently washed with lysis buffer. Proteins were eluted by addition of 500 mM imidazole (elution buffer: 50 mM Tris–HCl pH 8.1, 500 mM NaCl, 500 mM imidazole pH 8, 10% glycerol and 0.1 mM TCEP supplemented with protease inhibitors) and Flag M2 beads were immediately added to the eluate, and incubated for 2 h rotating top-down at 4 °C. The M2 beads were washed twice with lysis buffer supplemented with 10 mM MgCl$_2$ and 2 mM ATP to remove folding chaperones. Flag-tagged proteins were eluted twice by adding lysis buffer containing 0.5 mg ml$^{-1}$ Flag peptide (MRC PPU Reagents and Services). The eluate was incubated with 200 nM His$_6$–Ulp1 (prepared as previously described[71]) rotating top-down for 1 h at 4 °C. The cleaved 2xFlag–SUMO tag was removed by re-binding the eluate on Flag M2 beads rotating top-down for 1 h at 4 °C and collecting the unbound fraction. Proteins were concentrated in an Amicon Ultra Centrifugal tube with a molecular weight cutoff of 30 kDa and frozen in liquid nitrogen.

### CeSPT-2 HBD (a.a. 552–661) and HsSPT2 HBD (a.a. 571–685) WT and HBMs.

Expression of His$_{14}$–SUMO–CeSPT-2 HBD or His$_{14}$–SUMO–HsSPT2 HBD was induced by growing bacteria overnight at 20 °C with 400 µM IPTG. After cell lysis (lysis buffer: 20 mM Tris–HCl pH 7.5, 200 mM NaCl, 40 mM imidazole pH 8 and 0.1 mM TCEP supplemented with protease inhibitors), lysates were incubated with NiNTA beads for 2 h rotating top-down at 4 °C and washed with lysis buffer. Proteins were eluted by addition of 500 mM imidazole (elution buffer: 20 mM Tris–HCl pH 7.5, 500 mM NaCl, 500 mM imidazole pH 8 and 0.1 mM TCEP supplemented with protease inhibitors). The eluate was incubated with 60 µM His$_6$–Ulp1 to cleave the His$_{14}$–SUMO tag and dialyzed overnight against dialysis buffer (20 mM Tris–HCl pH 7.5, 500 mM NaCl and 0.1 mM TCEP). The cleaved His$_{14}$–SUMO tag was removed by re-binding the eluate on NiNTA beads and collecting the unbound

fraction containing the SPT2 HBD. Proteins were concentrated in a Amicon Ultra Centrifugal tube with molecular weight cutoff of 3 kDa and frozen in liquid nitrogen.

### Histone H3–H4 pull-down

Protein LoBind tubes were used at all steps. For each reaction, 10 µl of Pierce NHS-activated magnetic beads (ThermoFisher, 88826) were prepared according to the manufacturer's instructions. Briefly, the beads were washed with 1 ml of cold 1 mM HCl and then were resuspended in 1 ml of Coupling Buffer (500 mM NaCl and 200 mM NaHCO$_3$, pH 8.3). Then 20 pmol of recombinant histone H3.1–H4 tetramer (NEB, M2509S) was immediately added to the tube. The reaction was incubated overnight at 4 °C with top-to-bottom rotation. The following day, the beads were washed twice with 0.1 M glycine (pH 2.0) and once with MilliQ water. One milliliter of quenching buffer (500 mM ethanolamine and 500 mM NaCl pH 8.5) was added to the beads, followed by 2 h incubation at 4 °C. The beads were then washed once with MilliQ and twice with binding buffer (20 mM Tris–HCl pH 8.0, 500 mM NaCl, 0.1% Triton X-100 and 0.1 mM TCEP). Then 5 pmol of recombinant SPT2 proteins was added to the beads in 800 µl binding buffer and incubated for 1.5 h at 4 °C, rotating top-down. The beads were then washed three times with washing buffer (20 mM Tris–HCl pH 8.0, 0.1% Triton X-100 and 0.1 mM TCEP) supplemented with either 700 mM (●) or 800 mM (▲) NaCl, each time rotating for 15 min on a top-down wheel at 4 °C. The beads were then boiled for 10 min in 1× lithium dodecyl sulfate (LDS) Sample buffer (supplemented with 150 mM dithiothreitol (DTT)), and the supernatant was loaded on a 4–12% NuPage Bis-Tris gel. For the H3–H4 pull-down using the isolated HBD, the protocol was the same as above, except for the following differences: NHS-activated Sepharose 4 Fast Flow beads (Cytiva, 17090601) instead of magnetic beads were used; 1 nmol of recombinant *Xenopus laevis* H3(Δ1–40 a.a.)-H4 (a gift from R. Sundaramoorthy and T. Owen-Hughes) and 0.5 nmol of recombinant CeSPT-2 or HsSPT2 HBDs were used; the eluted material was run on a home-made 18% acrylamide gel and then stained with InstantBlue (Abcam) Coomassie protein staining.

### Electrophoretic mobility shift assay

**Preparation of labeled HJ substrates.** An equimolar mixture of all four oligonucleotides (J3b(40), J3h(40), J3r(40) and J3x(40)) was 5′-$^{32}$P-labeled, then annealed by slow cooling. The four-way junction was purified by electrophoresis on a native 8% polyacrylamide gel and recovered by the crush-and-soak method followed by ethanol precipitation. For binding assays, the substrate (10 nM) was incubated with increasing amounts of SPT2 at 37 °C for 15 min in binding buffer (25 mM Tris–HCl pH 8.0, 55 mM NaCl, 1 mM ethylenediaminetetraacetic acid (EDTA), 1 mM DTT, 0.1 mg ml$^{-1}$ bovine serum albumin and 1% glycerol), then mixed with loading buffer (Ficoll 400, 2.5% final) and run on a native 8% polyacrylamide gel at 8 V cm$^{-1}$ for 2 h. The gel was then dried, exposed to a Storage Phosphor screen and quantified with a Typhoon FLA 9500 phosphorimager (GE Healthcare).

Oligonucleotide sequences (5′–3′) are:

J3b(40): AGGGATCCGTCCTAGCAAGGGGCTGCTACCGGAAGCT-TAC

J3h(40): GTAAGCTTCCGGTAGCAGCCTGAGCGGTGGTTGAATTCAC

J3r(40): GTGAATTCAACCACCGCTCAACTCAACTGCAGTCTAGAAC

J3x(40): GTTCTAGACTGCAGTTGAGTCCTTGCTAGGACGGATCCCT

### General *C. elegans* maintenance and strains

The list of strains used in this study is provided below. Unless otherwise indicated, worms were grown on 1× nematode growth media (NGM) plates (3 g l$^{-1}$ NaCl, 2.5 g l$^{-1}$ peptone, 20 g l$^{-1}$ agar, 5 mg l$^{-1}$ cholesterol, 1 mM CaCl$_2$, 1 mM MgSO$_4$, 2.7 g l$^{-1}$ KH$_2$PO$_4$ and 0.89 g l$^{-1}$ K$_2$HPO$_4$) seeded with OP50 bacteria. Worms were routinely kept at 20 °C. All *spt-2* alleles used in this study were generated by SunyBiotech using clustered regularly interspaced short palindromic repeats (CRISPR)–Cas9

technology. All *spt-2* mutant strains were back-crossed six times against the reference WT N2 strain.

To genotype *spt-2* KO worms (*syb1268* and *syb1269*), primers GSA2, GSA3 and GSA4 were used to amplify the *T05A12.3* gene from genomic DNA obtained from lysing single worms, using a Quick-Load Taq 2× Master Mix (M0271L); extension time, 1 min.

GSA2: ATTCCGGTTCTCGACAATCG
GSA3: CCATTTCGGTATCTATCGGC
GSA4: AATGGGGTGAGAAGGTTGAC

To genotype the presence of the M627A mutation, two separate PCR reactions were set up, using a Quick-Load Taq 2× Master Mix, to observe the presence of either WT (Met627, ATG) or mutant 627 (Ala627, GCT) codon. Primers GSA93 and GSA92 are used to identify the presence of the M627A mutation, while GSA93 and GSA95 to identify the WT codon.

GSA92: ACGATATCAGAACGAGAcGCT
GSA93: GGAGACGGAAATAGAAATGTGG
GSA95: GACGATATCAGAACGAGATATG

### Analysis of GFP::SPT-2 tissue expression

For the analysis of the tissue expression of GFP::SPT-2, L4 animals were mounted in M9 buffer with 0.25 mM levamisole on 2% agarose pads. Imaging was performed at 20 °C using a microscope equipped with a ×63 1.25 numerical aperture oil lens (Imager M2; Carl Zeiss) and a charge-coupled device camera (Axiocam 503 mono). Nomarski and GFP *Z*-stacks (2 μm per slice) were sequentially acquired using the Zeiss acquisition software (ZEN v.3.1 blue edition). The same LED intensity and acquisition time was used for all images (Nomarski (50%, 20 ms), GFP (50%, 100 ms)).

### Brood size analysis

Worms at the L4 stage were singled onto 1× NGM plates with a thin layer of OP50 bacteria and allowed to grow at either 20 or 25 °C. Worms were passaged to a new plate every 12–24 h to keep generations separated, and fertilized eggs and L1 larvae were counted until each adult was not laying eggs anymore. Worms that were not alive by the end of the experiment were discarded from the analysis.

### Transgenerational sterility assay

Worms of the indicated genotypes were grown at 20 °C before shifting to 25 °C at the L4 stage. Either one or three L4 larvae (as indicated in the figure legend) were placed on each 1× NGM plate seeded with OP50 bacteria and grown at 25 °C. Every 3 days, three L3–L4 larvae were transferred to a new plate. Worms were considered sterile when no progeny was found on the plate. Ten plates per genotype were used in each repetition of the assay as shown in Fig. 2d,e and Extended Data Fig. 2c; while the indicated number of lines was used for the assays in Extended Data Fig. 2e. The progeny of each worm line was scored after 6 days of growing at 25 °C. Scoring criteria (as shown in Extended Data Fig. 2e) were as follows: WT—worms were starved, no more bacteria present; medium—~100 worms, not starved, bacteria are still present; few—~30 worms, not starved, bacteria are still present; sterile—no progeny. Assays shown in Fig. 2d,e and Extended Data Fig. 2c were performed by G.S., while assays shown in Extended Data Fig. 2e were performed by S.G.M.R.

### *hira-1* RNAi

RNAi was performed by feeding worms with HT115 bacteria containing L4440 plasmids that express dsRNA. The plasmid expressing *hira-1* dsRNA was obtained from a commercial RNAi library (clone III-2P01, Source Bioscience), and the empty L4440 vector was used as a control. dsRNA expression was induced by adding IPTG to a final 3 mM concentration in LB medium supplemented with 50 μg ml⁻¹ ampicillin, and RNAi bacteria were seeded on 1× NGM plates at OD₆₀₀ equal 1. When the plate was dry, six L4 worms of the indicated genotypes were added to the plate and allowed to grow at 25 °C until the next generation (2 days).

New *hira-1* RNAi, or vector RNAi, plates were prepared as described above. Forty L2–L3 worms per genotype were singled on the RNAi plates and allowed to grow at 25 °C for 4 or 5 days, after which plates were blindly scored for Pvl phenotype and presence of burst worms.

### Transgenerational memory inheritance

The transgenerational memory inheritance assay was described previously[72]. Three L4 larvae per genotype were plated on *gfp* RNAi-expressing bacteria (five replicates for each *spt-2*ᴷᴼ strain, or three replicates for each *spt-2*ᴴᴮᴹ sibling) or empty vector L4440 bacteria (three replicates). Generation 1 (G1) animals were analyzed under a fluorescence microscope, and one silenced animal per replicate per genotype was plated onto HB101 bacteria. At each generation, one silenced animal was singled from each plate to produce the next generation, and the remaining adult progeny was analyzed under a Kramer FBS10 fluorescence microscope. Animals were collected in M9 buffer, washed twice, quickly fixed in 70% ethanol and deposited onto a glass slide coated with a 2% agarose pad. At least 35 animals per replicate per genotype were counted at each generation. Germline nuclear GFP brightness was scored as 'on' or 'off' by visual inspection (dim GFP expression was scored as 'off'). Representative images were taken on a SP8 confocal fluorescence microscope (Leica) at ×40 magnification.

### GFP::CeSPT-2 ChIP–seq

Animals of the indicated genotype (grown at 20 °C) were bleached, and embryos allowed to hatch overnight in M9 buffer. Approximately 300,000 worms were used per condition, divided in six 15-cm plates seeded with thick HB101 bacteria. Synchronized adult animals (70 h post-seeding) were washed four times in M9. Worms were flash-frozen in liquid nitrogen to obtain 'popcorns' that were ground using a BioPulverizer (MM400 Mixer Mill, Retsch) liquid nitrogen mill, until adult worms were broken in three to four pieces each. Worm powder was resuspended in cold phosphate-buffered saline (PBS; supplemented with Roche cOmplete protease inhibitors). Freshly prepared ethylene glycol bis(succinimidyl succinate) (EGS) solution was added to a final 1.5 mM concentration and incubated rotating for 8 min, followed by crosslinking with 1% methanol-free formaldehyde (ThermoFisher, 28908) for 8 min and quenching with glycine (final concentration 125 mM). The crosslinked chromatin was then washed twice in PBS (with protease inhibitors) and once in FA buffer (50 mM HEPES/KOH pH 7.5, 1 mM EDTA, 1% Triton X-100, 0.1% sodium deoxycholate and 150 mM NaCl) supplemented with protease (Roche Complete) and phosphatase (PhosStop) inhibitors. Chromatin was then sonicated using a Bioruptor (Diagenode) on high mode, 30 cycles at 30 s on, 30 s off. A 30-μl aliquot was decrosslinked to confirm enrichment of DNA fragments between 100 and 300 bp. To decrosslink, chromatin was spun down at maximum speed and the supernatant was transferred to a new tube and resuspended in FA buffer supplemented with 2 μl of 10 mg ml⁻¹ RNase A (incubation at 37 °C for 10 min) followed by Proteinase K treatment (incubation at 65 °C for 1 h). DNA concentration was measured using the Qubit assay (Qubit dsDNA High Sensitivity assay, ThermoFisher). The ChIP reaction was assembled as follows: 20 μg of DNA were used per ChIP reaction, together with 10 μg of GFP antibody (ab260) in 1 ml of FA buffer (with protease and phosphatase inhibitors) supplemented with 1% sarkosyl. ChIP reactions were incubated overnight rotating at 4 °C. For each reaction, 40 μl of protein A magnetic beads slurry were blocked overnight in 1 ml of FA buffer (with protease and phosphatase inhibitors) supplemented with 10 μl of transfer RNA (Merck, R5636). The following day, beads were added to the immunoprecipitation reaction and incubated for a further 2 h rotating at 4 °C. The beads were then washed with 1 ml of the following buffers (ice cold), each time rotating 5 min at 4 °C: twice with FA buffer (with protease and phosphatase inhibitors); once with FA buffer supplemented to 500 mM NaCl; once with FA buffer supplemented to 1 M NaCl; once with TEL buffer (10 mM Tris–HCl pH 8, 250 mM LiCl, 1% NP-40, 1% sodium deoxycholate and

1 mM EDTA) and finally twice in TE buffer (pH 8). DNA was eluted twice with 60 µl of ChIP elution buffer (1% SDS and 250 mM NaCl in TE buffer pH 8), each time incubating for 15 min at 65 °C (300 rpm shaking). Eluted samples were treated with 2 µl of 10 mg ml⁻¹ RNase A for 1 h at 37 °C and then decrosslinked overnight at 65 °C with 1.5 µl of 20 mg ml⁻¹ Proteinase K. DNA was purified with PCR purification columns, and DNA concentration was measured with the Qubit assay. Sequencing library preparation was performed as previously described[73] using a modified Illumina TruSeq protocol. Briefly, DNA fragments were first treated with end repair enzyme mix (NEB, E5060) for 30 min at 20 °C in 50 µl volume, and DNA fragments were subsequently recovered with one volume of AMPure XP beads (Beckman Coulter, cat. no. A63880) mixed with one volume of 30% of PEG$_{8000}$ in 1.25 M NaCl. DNA was eluted in 17 µl of water and further 3′ A-tailed using 2.5 units of Klenow 3′ to 5′ exo(−) (NEB, cat M0212) in 1× NEB buffer 2 supplemented with 0.2 mM dATP for 30 min at 37 °C. Illumina Truseq adaptors were ligated to the DNA fragments by adding 25 µl of 2× Quick Ligase buffer (NEB, M2200), 1 µl of adaptors (1 µl of a 1:250 dilution of Illumina stock solution), 2.5 µl water and 1.5 ml of NEB Quick ligase (NEB, M2200). The reaction was incubated 20 min at room temperature, and 5 µl of 0.5 M EDTA was added to inactivate the ligase enzyme. DNA was purified using 0.9 volumes of AMPure XP beads, and DNA fragments were eluted in 20 µl water. One microliter of DNA was used to set up a qPCR reaction to determine the number of cycles needed to get amplification to 50% of the plateau, which is the cycle number that will be used to amplify the library. One microliter of DNA was mixed with 5 µl of KAPA Syber Fast 2× PCR master mix and 1 µl of 25 µM Truseq PCR Primer cocktail. Each library (19 µl) was then amplified with 20 µl of KAPA HiFi HotStart Ready Mix (KM2605) and subsequently size selected to a size between 250 bp and 370 bp. To achieve this, DNA was first purified with 0.7 volumes of AMPure beads to remove all fragments above 370 bp, keeping the supernatant and discarding the beads. All DNA fragments were then recovered from the supernatant by adding 0.75 volumes of beads and 0.75 volumes of 30% PEG$_{8000}$ in 1.25 M NaCl, and eluted in 40 µl of water. To recover fragments above 250 bp, 0.8 volumes of beads were added to bind the library. DNA was then eluted in 10 µl of water, quantified using a dsDNA HS Qubit kit, and the size distribution of the libraries was analyzed using an Agilent TapeStation.

### Processing of sequencing data

ATAC−seq and ChIP−seq reads were preprocessed using trim-galore (v.0.6.7; available at ref. [74]) and mapped on the *C. elegans* (wormbase release WS285) or human genome (hg38) using bwa-mem (v.0.7.17)[75]. Reads with mapping quality (MAPQ) higher than 10 were extracted using Samtools[76]. ATAC−seq peaks were called in *C. elegans* by first generating read depth-normalized coverage tracks with MACS2 (ref. [77]; v.2.7.1; settings −nomodel −extsize 150 −shift −75 −keep-dup all −SPMR −nolambda) and then running YAPC[73] (v.0.1) using bigwig tracks from both replicates for each condition as input. CeSPT-2 ChIP−seq peaks were called using MACS2 (settings −SPMR −gsize ce −keep-dup all −nomodel −broad) using the GFP ChIP−seq samples from WT animals as controls. The final CeSPT-2 peak set was defined by the strict intersection of the broad peaks called on each replicate.

RNA-seq data were aligned on annotated transcripts (Wormbase release WS285; gene types included: protein coding, long intergenic noncoding RNAs and pseudogenes) using kallisto[78] (v.0.45.0) to estimate gene expression (in transcripts per million). Coverage profiles were obtained by aligning reads to the genome using STAR[79] (v.2.7.1a).

### Annotation of CeSPT-2 bound genes

A gene was considered an CeSPT-2 target if more than 50% of the length of its longest annotated transcript was covered by a CeSPT-2 peak. The average CeSPT-2 coverage over CeSPT-2 targets was calculated using coverageBed from the BEDTools suite[80] (v.2.30.0). Coverage plots over gene models were produced using the DeepTools suite[81] (v.3.5.1).

### ATAC−seq

**ATAC−seq in *C. elegans*.** Animals of the indicated genotype (grown at 20 °C) were bleached, and embryos allowed to hatch overnight in M9 buffer. Approximately 12,000 L1 worms were seeded onto 15-cm plates seeded with a thick lawn of HB101 bacteria. Gravid adult worms were collected 70 h post-seeding and washed three times in M9 buffer. To assess that worm synchronization was equal between samples, 10 µl of worm slurry was fixed in cold methanol overnight at −20 °C; the remaining slurry was frozen in 'popcorns'. The methanol-fixed worms were stained with 4′,6-diamidino-2-phenylindole (DAPI; final 1 µg ml⁻¹) for 10 min, washed three times with PBS/0.1% Tween and rehydrated overnight at 4 °C in PBS/0.1% Tween. Worms were then mounted on a slide and inspected on a Leica SP8 ultraviolet microscope. ATAC−seq was performed as previously described[73]. Frozen worms (three to four frozen popcorns) were broken by grinding in a mortar and pestle. The frozen powder was thawed in 10 ml Egg buffer (25 mM HEPES pH 7.3, 118 mM NaCl, 48 mM KCl, 2 mM CaCl$_2$ and 2 mM MgCl$_2$) and washed twice with that buffer by spinning 2 min at 1,500*g*. After the second wash, the pellet was resuspended into 1.5 ml of Egg buffer containing 1 mM DTT, protease inhibitors (Roche complete, EDTA free) and 0.025% of Igepal CA-630. Samples were dounced 20 times in a 7-ml stainless tissue grinder (VWR) and then spun at 200*g* for 5 min to pellet large debris. Supernatant containing nuclei was transferred into a new tube, and nuclei were counted using a hemocytometer. One million nuclei were transferred to a new 1.5-ml tube and spun at 1,000*g* for 10 min, the supernatant was removed, and the nuclei resuspended in 47.5 µl of Tagmentation buffer (containing 25 µl of 2× Tagmentation buffer from Illumina and 22.5 µl of water) before adding 2.5 µl of Tn5 enzyme (Illumina Nextera kit) and incubated for 30 min at 37 °C. Tagmented DNA was then purified using MinElute column (Qiagen) and amplified in ten cycles using the Nextera kit protocol. Amplified libraries were finally size selected using AMPure beads to recover fragments between 150 and 500 bp and sequenced.

**ATAC−seq in human U-2 OS cells.** ATAC−seq was performed using the Active Motif ATAC-seq kit (53150) according to the manufacturer's protocol. Briefly, WT or SPT2 KO (KO-3, KO-4 and KO-6) U-2 OS cells were grown in three separate technical batches for 2 weeks to obtain independent replicates. On the day of the ATAC−seq experiment, 80,000 cells were collected by trypsinization, washed in ice-cold PBS and then resuspended in 100 µl of ice-cold ATAC lysis buffer. After centrifugation, the cell pellet was resuspended in 50 µl of Tagmentation Master Mix containing Tn5 transposomes assembled with adaptors, and incubated at 37 °C for 30 min. DNA was then purified on columns and eluted in 35 µl of purification elution buffer. The tagmented DNA was then amplified with Q5 polymerase (ten cycles), according to the Active Motif ATAC−seq kit, using i5 and i7 indexed primers. The PCR reaction products were then purified using SPRI beads (1.2× sample volume). The size distribution of the libraries was checked using a TapeStation, and a further round of DNA purification with SPRI beads was performed (0.5× sample volume) to remove fragments above 600 bp. Libraries were sequenced on an Illumina HiSeq instrument.

### Analysis of DA genes (ATAC−seq)

**ATAC−seq analysis in *C. elegans*.** We used DiffBind[82] (v.3.8.0) to identify genes showing differential accessibility in the *spt-2*[KO-A] and the *spt-2*[HBM] strains compared to the WT. For this analysis, we had to redefine gene coordinates to avoid overlaps, which would be otherwise merged in DiffBind. The boundaries of the longest transcript for each gene were trimmed to remove overlaps with neighboring transcripts; if more than 50% of the length of the transcript was trimmed, the whole gene locus was removed from the analysis. Genes were defined as DA if the |log fold change (LFC)| >0 and FDR <0.05, and their enrichment evaluated at non-CeSPT-2 targets and at CeSPT-2 targets (only 1,991/2,177 could be analyzed due to overlapping gene intervals) divided in four equally sized groups based on their CeSPT-2 levels.

**ATAC-seq analysis in human cells.** We used DiffBind[82] (v.3.8.0) to identify genes showing differential accessibility in the any SPT2 KO strain compared to the WT. We used the human Gencode (release 21) gene annotation focusing on the following gene classes: long intergenic noncoding RNA, processed_pseudogene, protein_coding, transcribed_processed_pseudogene, transcribed_unprocessed_pseudogene, unitary_pseudogene, unprocessed_pseudogene. To avoid overlaps between gene loci, we used the same procedure described for *C. elegans*. Genes were defined as DA if the |LFC| >0 and FDR <0.05. Two replicates (SPT2 KO-6 Replicate 2 and SPT2 KO-4 Replicate 1) were excluded from the analysis due to a lower signal-to-noise ratio compared to other replicates from KO strains. A WT sample (WT Replicate 3) was also excluded as it showed differences in its accessibility profile compared to the other WT replicates. These three excluded samples all clustered separately from other WT and mutant samples from the same strain both in PCA and in correlation heatmaps.

## Collection of adult worms for total RNA extraction
Animals of the indicated genotype (grown at 20 °C) were bleached, and embryos were allowed to hatch overnight in M9 buffer. Approximately 2,000 L1 worms were seeded onto 10-cm plates seeded with HB101 bacteria. Gravid adult worms were collected 70 h post-seeding and washed three times in M9 buffer, and the worm pellet was resuspended in 1 ml of TRIzol (ThermoFisher, 15596026). This mixture was quickly thawed in a water bath at 37 °C, followed by vortexing and freezing in liquid nitrogen; five freeze–thaw cycles were performed. Then, 200 µl of chloroform was added, and the mixture was vortexed for 15 s and incubated at room temperature for 3 min. The mixture was spun down for 15 min at 15,000*g* in a refrigerated centrifuge (4 °C). The upper aqueous phase was carefully moved to a new 1.5-ml tube and RNA was retrieved by isopropanol precipitation. Briefly, 1 µl of glycogen and 500 µl of isopropanol was added and, after vortexing, the mixture was incubated overnight at −20 °C. The next day, the mix was centrifuged for 15 min at 15,000*g*, 4 °C and the supernatant was removed. One milliliter of 70% EtOH was added, and the tube was again centrifuged for 5 min at 15,000*g*, 4 °C. After carefully removing the supernatant, the RNA pellet was allowed to dry at room temperature for a maximum of 5 min and then resuspended in 40 µl diethyl pyrocarbonate-treated water (Invitrogen). The RNA quality was assessed on an Agilent TapeStation, using the RNA ScreenTape kit according to the manufacturer's instructions (Agilent, 5067-5576 and 5067-5577).

## Poly(A) enrichment, library preparation and sequencing
Total RNA extracted from adult worms was treated with Turbo DNA-free kit (Invitrogen, AM1907) according to the manufacturer's protocol. RNA concentration was measured with a Qubit instrument, and 1 µg of total RNA was used to make libraries. Poly(A) tail selection was performed using NEBNext Poly(A) mRNA Magnetic Isolation Module (NEB, E7490S), and libraries for next-generation sequencing were prepared using the NEBNext Ultra II Directional RNA Library Prep kit with Sample Purification Beads (NEB, E7765S). Libraries were indexed using NEBNext Multiplex Oligos for Illumina Index Primers Set 1 and 2 (E7335S and E7500S). Sequencing was performed on a NovaSeq instrument, 100 bp paired-end sequencing.

## Differential gene expression
We used DESeq2 (ref. 83; v.1.34.0) to identify genes differentially expressed in the *spt-2*[KO] and the *spt-2*[HBM] strains compared to the WT. Genes were defined as differentially expressed if the |LFC| >0 and adjusted *P* value <0.001. Volcano plots were generated using the EnhancedVolcano R package (available at ref. 84). GO-enrichment analysis of differentially expressed genes was performed using cluster-Profiler[85]. The overlap between genes upregulated in *spt-2* and *hira-1* mutant[35] was tested using a hypergeometric test. The list of RNAi effector genes was compiled from ref. 86.

## Reverse transcription qPCR
For the retro-transcription reaction, 1 µl of 50 µM random hexamers and 1 µl of 10 µM dNTPs were added to 200–500 ng of RNA in a 10 µl volume reaction. The reactions were heated for 5 min at 65 °C to denature secondary structures and allow random primer annealing. Then, 4 µl of 5× first strand buffer, 2 µl of 0.1 M DTT, 1 µl of RNaseOUT and 1 µl of SuperScript RT III were added. This mix was incubated on a thermocycler with the following program: 10 min at 25 °C, 1 h at 50 °C and 15 min at 70 °C. Before proceeding with the qPCR reaction, the mix was diluted 1:5 with Sigma water. qPCR was performed with the TB Green Premix Ex Taq II (Tli RNase H Plus, RR820L Takara) in a 384-well plate, using a Bio-Rad CFX384 real-time PCR system.

qPCR primer sequences (5′–3′) are as follows:
GSA144 gfp_ex1_F: GTGAAGGTGATGCAACATACGG (from ref. 72)
GSA145 gfp_ex1/2junction_R: ACAAGTGTTGGCCATGGAAC (from ref. 72)
GSA146 cdc-42 F: CTGCTGGACAGGAAGATTACG (from ref. 72)
GSA147 cdc-42 R: CTCGGACATTCTCGAATGAAG (from ref. 72)
GSA213 R06C1.4 F1: GTGTACGATCGTGAAACCGG
GSA214 R06C1.4 R1: CGAAGGTTGCGTCCATTGAA
GSA215 Y57G11B.5 F1: GCTTGTAATGCCGAGACGAG
GSA216 Y57G11B.5 R1: TGGAACATTTCGAGACGGGA
GSA217 asp-1 F1: GATTCCAGCCATTCGTCGAC
GSA218 asp-1 R1: TGATCCGGTGTCAAGAACGA
GSA221 clec-66 F1: TGCCATGACTAAATTCGCCG
GSA222 clec-66 R1: ACGCTCTCTTCTGTTGGTCA
GSA223 F08B12.4 F1: GAAAAGCGTCTTGGAAGGGG
GSA224 F08B12.4 R1: TTACTGGTGGTTTTGCTCGC
GSA225 C14C6.5 F1: CTACGACAATGGCACCAACC
GSA227 C14C6.5 R1: TTCATTCCTGGGCAGTCACT
GSA227 skr-10 F1: TGAGAGAGCTGCAAAGGAGA
GSA228 skr-10 R1: TGGAAGTCGATGGTTCAGCT
GSA350 H3.3-SNAP F2: CCTGGCTCAACGCCTACTTT (from ref. 60)
GSA351 H3.3-SNAP R2: GGTAGCTGATGACCTCTCCG (from ref. 60)
GSA344 gapdh F: CAAGGCTGTGGGCAAGGT (from ref. 87)
GSA345 gapdh R: GGAAGGCCATGCCAGTGA (from ref. 87)

## Human cell culture
All cells were kept at 37 °C under humidified conditions with 5% $CO_2$. Parental U-2 OS cells (female cells; source ATCC, HTB-96) were grown in Dulbecco's modified Eagle medium (Life Technologies) supplemented with 100 U ml[−1] penicillin, 100 µg ml[−1] streptomycin, 1% L-glutamate (Invitrogen) and 10% fetal bovine serum. U-2 OS SNAP-H3.3 cells were described before[49] and were grown in the medium described above supplemented with 100 µg ml[−1] G418 (Formedium, G4181S). To generate stable U-2 OS Flp-In Trex cell lines, hygromycin was used to select for the integration of GFP–SPT2 constructs at the Flp-In recombination sites in U-2 OS Flp-In Trex cells (obtained from K. Haynes's laboratory, Harvard). U-2 OS Flp-In Trex cells expressing GFP-tagged SPT2 were grown in the medium described above supplemented with 100 µg ml[−1] hygromycin and 10 µg ml[−1] blasticidin. Expression of GFP–SPT2 was induced by addition of 1 µg ml[−1] of tetracycline for 24 h. All cell lines tested negative for mycoplasma contamination. HEK293 cells were obtained from K. Helin's laboratory (BRIC, Copenhagen).

## siRNA transfection
Twenty-four hours before transfection, 80,000 U-2 OS cells were seeded per six-well dish. siRNA transfection was performed with RNAiMax reagent (Invitrogen) according to the manufacturer's protocol, and all siRNAs were used to a final concentration of 50 nM. Cells were collected 48 h post-transfection. siRNAs were synthetized by Dharmacon.

siRNA sequences (5′–3′) are the following:
ON-TARGETplus Non-targeting siRNA #1 (Dharmacon, D-001810-01-05)

siSPT2 no. #1: GACCTATGACCGCAGAAGA
siSPT2 no. #2: GTTACAATGGGATTCCTAT
siSPT2 no. #3: GAGAATTCCTTGAACGAAA
siHIRA no. #1: GAAGGACUCUCGUCUCAUG (from ref. 88)

## CRISPR–Cas9 KO of the *SPTY2D1* gene

To knock out the *SPTY2D1* gene from U-2 OS H3.3-SNAP cells, we transfected 2 µg of pX459 *SPTY2D1* exon3 KO Antisense Guide 2 plasmid (DU74304) using PEI MAX (Polysciences). Twenty-four hours post-transfection, medium was replaced with fresh medium containing 2 µg ml$^{-1}$ puromycin and cells were allowed to grow for another 48 h in selection conditions. Single cells were sorted into 96-well plates using a MA900 cell sorter (Sony Biosciences) and allowed to grow until they formed visible colonies. Cells were then expanded and approximately 30 colonies were screened by western blotting for the absence of SPT2 band; 11 of these colonies showed no WT SPT2 band and were further tested by immunoprecipitation using our in-house produced SPT2 antibody. DNA was then extracted from three positive clones to amplify a PCR fragment encompassing exon 3 of *SPTY2D1*, which contains the CRISPR guide target site, using primers GSA370 and GSA371; this PCR fragment was cloned into a StrataClone vector (Stratagene Blunt PCR Cloning kit 240207) according to the manufacturer's instructions, and analyzed by sequencing using T7 and T3 primers. Three clones (SPT2 KO-3, SPT3 KO-4 and SPT2 KO-6) lacked a WT *SPTY2D1* allele, instead harboring frameshift mutations and/or insertions leading to missense and/or premature stop codons.

PCR primers sequences:
GSA370 SPTY2D1 ex3 F1: ACTTGTTCACATGACTTCCTCATTT GATTCC
GSA371 SPTY2D1 ex3 R1: GTTCAAACTGCTTTTTCTCAGCCAGC

## Lentiviral transduction

Lentiviral rescue plasmids are DU71837 (pLV(exp) puro CMV SPT2) and DU71838 (pLV(exp) puro CMV SPT2 M641A E651A E652A). HEK293 cells were transfected with rescue plasmids using the calcium phosphate method as follows: 10 µg of lentiviral transfer vector was mixed with 8 µg psPAX2, and 4 µg MD2.G plasmids in 500 µl of 250 mM CaCl$_2$ solution at room temperature, then an equal amount of 2× HBS (280 mM NaCl, 50 mM HEPES and 1.42 mM Na$_2$HPO$_4$, pH 7.05) was added dropwise with continuous vortexing to generate a fine-particle precipitate. After a brief incubation (2–5 min), the transfection mix was added dropwise to the HEK293 cells in freshly changed medium, followed by gentle mixing. The medium was replaced 16 h after transfection and viral supernatants were collected 72 h post-transfection, filtered through a 0.45-µm filter and used for transduction on the same day. U-2 OS H3.3-SNAP SPT2 KO-3 and SPT2 KO-4 cells were transduced with a mix of viral supernatant and polybrene (8 µg ml$^{-1}$ final concentration). Selection with 2 µg ml$^{-1}$ puromycin was started after 24 h, and after 2 days all cells in the control, nontransduced well died. SPT2 KO cells reconstituted with SPT2 complementary DNA were subsequently grown in medium containing 2 µg ml$^{-1}$ puromycin and 100 µg ml$^{-1}$ G418. SPT2 re-expression was confirmed by western blot and by immunofluorescence.

## Soluble and/or chromatin fractionation

To analyze soluble and chromatin-bound proteins by western blotting, cells were washed in cold PBS, scraped, and centrifuged at 1,500*g* for 10 min at 4 °C. The pellet was incubated on ice for 10 min in CSK buffer (10 mM PIPES pH 7, 100 mM NaCl, 300 mM sucrose and 3 mM MgCl$_2$)/0.5% Triton X-100, supplemented with Roche Complete Protease Inhibitor cocktail and PhosSTOP Roche phosphatase inhibitors, followed by 5 min centrifugation at 1,500*g*, 4 °C (soluble fraction). The pellet was washed with CSK/0.5% Triton, before resuspending in 1× LDS buffer supplemented with 10 mM MgCl$_2$, 150 mM DTT and 250 units of Pierce Universal Nuclease (ThermoFisher, 88702), and incubated at 37 °C shaking for 1 h (chromatin fraction). To analyze chromatin-bound proteins by microscopy, cells were washed with cold PBS and then with cold CSK buffer, followed by pre-extraction with cold CSK/0.5% Triton X-100 buffer for 5 min, and then washed again with CSK and PBS before fixation with 4% formaldehyde for 10 min.

## RNA extraction from human cells

RNA was extracted from U-2 OS cells according to the Qiagen RNeasy Mini Extraction kit. DNaseI digestion was performed on columns using the Qiagen RNase-Free DNase Set.

## In vivo labeling of SNAP-tagged histone H3.3

A total of 10,000 U-2 OS cells stably expressing H3.3-SNAP were seeded in each well of a 96-well CELLSTAR plate with µClear flat bottom (Greiner). For labeling newly synthesized SNAP-tagged H3.3, pre-existing SNAP-tagged histones were first quenched by incubating cells with 10 µM of the nonfluorescent SNAP reagent (SNAP-cell Block, New England Biolabs) for 30 min at 37 °C, followed by a 30-min wash in fresh medium (warmed to 37 °C) and a 2-h chase. The SNAP-tagged histones neo-synthesized during the chase time were then pulse-labeled by incubating cells with 2 µM of the red-fluorescent SNAP reagent SNAP-cell TMR star (New England Biolabs) for 15 min at 37 °C, followed by a 30-min wash in fresh medium (quench-chase-pulse protocol). Cells were then either directly fixed in 4% paraformaldehyde for 10 min, or pre-extracted in CSK buffer containing 0.5% Triton X-100 for 5 min before fixation in 4% paraformaldehyde and DAPI staining. Cells were then imaged with an Olympus ScanR high-content automated microscope (equipped with a ×20 dry objective) and analyzed Olympus ScanR Analysis software. Nuclei were delineated on the basis of DAPI staining. For the experiments shown in Fig. 6e,f that use SPT2 KO cells recomplemented with untagged SPT2 cDNA (WT or HBM), cells were stained with anti-SPT2 antibody (1:1,000, overnight; secondary: Alexa Fluor antibody). When measuring H3.3-SNAP intensity in the rescued cell lines, a gate was created in the ScanR Analysis software based on SPT2 levels, so that only recomplemented cells that had SPT2 levels similar to the endogenous levels were considered. For siRNA experiments shown in Extended Data Fig. 6d,e, the same protocol was used except for the following differences: following TMR-Star labeling, cells are washed for 45 min in fresh medium instead of 30 min. Samples were observed with a Leica DMI6000 epifluorescence microscope using a Plan-Apochromat ×40/1.3 oil objective. Images were captured using a CCD camera (Photometrics) and MetaMorph Software. Fiji software was used for image analyses using custom macros. Nuclei were delineated on the basis of DAPI staining.

## Quantitative high-content microscopy analysis of GFP–SPT2 binding to chromatin

To analyze GFP–SPT2 intensity on chromatin, U-2 OS Flp-In cells grown in 96-well plates (Greiner) were treated with 1 µg ml$^{-1}$ tetracycline for 24 h to induce expression of GFP-tagged SPT2. Cells were then either directly fixed with 4% paraformaldehyde (10 min at room temperature), or pre-extracted for 5 min on ice with CSK/0.5% Triton buffer (supplemented with Roche Complete Protease inhibitor and PhosSTOP Phosphatase inhibitors) to remove soluble proteins, before fixation for 10 min with 4% paraformaldehyde at room temperature. DNA was stained with DAPI, and images were acquired using an Olympus ScanR high-content automated microscope (equipped with a ×20 dry objective) and analyzed Olympus ScanR Analysis software.

Images shown in Fig. 6b were acquired using a Perkin Elmer Operetta high-content automated microscope (equipped with a ×20 dry objective). The contrast for the images shown in Fig. 6b (right) was applied equally to the directly fixed and pre-extracted samples of each cell line; contrast was instead adjusted independently for the different cell lines, which express different levels of GFP-tagged HsSPT2 (as shown in Extended Data Figs. 4b and 6b).

## Western blotting and antibodies

Primary antibodies:

| Target | Company | Cat. no./clone | Dilution western blot | Dilution immunofluorescence |
|--------|---------|----------------|----------------------|------------------------------|
| SPT2 | DSTT | DA010 ($0.2\,\mu g\,\mu l^{-1}$) | 1:2,000 | 1:1,000 |
| GAPDH | Cell Signaling | 2118S, clone 14C10 | 1:5,000 | – |
| GFP | Abcam | ab290 | 1:2,000 | – |
| RPB1 | Cell Signaling | clone D8L4Y, 14958S | 1:2,000 | – |
| HIRA | Active Motif | 39457 | 1:1,000 | – |
| His$_6$ | Abcam | ab18184 | 1:1,000 | – |
| H3 | Abcam | ab1971 | 1:1,000 | – |

For western blotting, secondary IRDye LI-COR antibodies were used at a dilution of 1:15,000. Images were acquired using an Odyssey CLx LI-COR scanner and further processed using the LI-COR Empiri-aStudio Software (v.2.3.0.154). For immunofluorescence, secondary Alexa Fluor antibodies were used (1:1,000 dilution).

## Human SPT2 antibody production

Polyclonal SPT2 antibodies were raised in sheep by the MRC PPU Reagents and Services Unit (University of Dundee) and purified against the SPT2 antigen a.a. 385–685 (after depleting antibodies recognizing the epitope tags). Sheep DA010, third bleed, was used in this study. Sheep were immunized with the antigens followed by four further injections 28 days apart. Bleeds were performed 7 days after each injection.

## Statistical analysis

The difference of variance between two populations was measured using Prism 9 software, and a Welch's correction was applied when the variances were not equal. *P* values are provided and defined in figure legends. Multiple comparisons were performed with ordinary one-way analysis of variance (ANOVA) with Dunnett's post-tests. Whenever not indicated in the figure legends, exact *P* values of post-test comparisons are reported in Supplementary Table 3. The overlap between genes upregulated in *spt-2* and *hira-1* mutant worms (Fig. 5c) was calculated

| Genotype | Strain name | Source |
|----------|-------------|--------|
| WT N2 | | CGC |
| *spt-2(syb1268) IV* | JRG30 | This work, from SunyBiotech |
| *spt-2(syb1269) IV* | JRG31 | This work, from SunyBiotech |
| *spt-2(syb2412[M627A]) IV* | JRG48 | This work, from SunyBiotech |
| *spt-2(syb1735[mAID-gfp::spt-2(wt)]) IV* | JRG44 | This work, from SunyBiotech |
| *spt-2(syb2435[mAID-gfp::spt-2(M627A)) IV* | JRG45 | This work, from SunyBiotech |
| *spt-2(syb2412,syb4133[A627M]) IV* | PHX4133 | This work, from SunyBiotech |
| *mjls31 II* | SX461 | From ref. 41 |
| *mjls31 II; hrde-1(tm1200) III* | SX3448 | From ref. 72 |
| *mjls31 II; spt-2(syb1268) IV* | JRG49 | This work |
| *mjls31 II; spt-2(syb1269) IV* | JRG50 | This work |
| *mjls31 II; spt-2(syb2412[M627A]) IV* sibling 1 | JRG51 | This work |
| *mjls31 II; spt-2(syb2412[M627A]) IV* sibling 2 | JRG52 | This work |
| *hira-1(uge29) III* | FAS45 | From ref. 36 |

using a hypergeometric test. All tests related to Figs. 4c,f and 5c and Extended Data Fig. 4b were two-sided. Exact *P* values for ANOVA test shown in Fig. 3d were calculated in Excel using the FDIST function.

*C. elegans* strains:

Plasmids:

| Plasmid name | DSTT reference |
|--------------|----------------|
| 2xFlag–SUMO–CeSPT-2–His$_6$ full-length | DU70523 |
| 2xFlag–SUMO–CeSPT-2–His$_6$ full-length M627A | DU70525 |
| 2xFlag–SUMO–HsSPT2–His$_6$ full-length | DU70522 |
| His$_{14}$–SUMO–CeSPT-2 HBD (a.a. 551–662) | DU70520 |
| His$_{14}$–SUMO–CeSPT-2 HBD (a.a. 551–662) M627A | DU70521 |
| His$_{14}$-SUMO-HsSPT2 HBD (aa. 571–685) | DU70518 |
| His$_{14}$–SUMO–HsSPT2 HBD (a.a. 571–685) M641A | DU70519 |
| pcDNA5–FRT/TO–GFP | DU13156 |
| pcDNA5–FRT/TO–GFP–SPT2 | DU63205 |
| pcDNA5–FRT/TO–SPT2–GFP | DU63486 |
| pcDNA5–FRT/TO–GFP–SPT2 1–570 | DU63519 |
| pcDNA5–FRT/TO–GFP–SPT2 571–end | DU63517 |
| pcDNA5–FRT/TO–GFP–SPT2 M641A | DU63497 |
| pLV(exp)–puro–CMV–SPT2 | DU71837 |
| pLV(exp)–puro–CMV–SPT2 M641A, E651A, E652A (3xHBM) | DU71838 |
| L4440 | DU70356 |
| pX459 *SPTY2D1* exon3 KO Antisense Guide 2 | DU74304 |
| L4440–*gfp* | From ref. 72 |
| L4440–*hira-1* | Source Bioscience |

## Reporting summary

Further information on research design is available in the Nature Portfolio Reporting Summary linked to this article.

## Data availability

All plasmids and antibodies generated in this study can be requested to the MRC PPU DSTT at https://mrcppureagents.dundee.ac.uk/reagents-from-papers/Rouse-SPT2-paper-1. All NGS datasets have been deposited on GEO with accession number GSE224802. The previously published crystal structure of SPT2 HBD in complex with H3–H4 (ref. 26) was retrieved from PDB under the ID 5BS7. Other materials generated in this work will be made available upon request. Source data are provided with this paper.

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

## Acknowledgements

We are grateful to members of the Labib laboratory for invaluable advice on recombinant protein purification and *C. elegans* techniques, and to members of the Rouse laboratory for fruitful discussion. We thank K. Rasmussen for help with ATAC–seq in human cells; R. Sundaramoorthy and T. Owen-Hughes (University of Dundee) for the gift of recombinant *Xenopus* histones; B. Meier and F. Pelisch for help with *C. elegans* genetics; K. Labib for helpful comments on the manuscript and for the gift of the His₆–Ulp1 plasmid; V. Alvarez for help with microscopy and A. Knebel for advice on protein purification. We thank the technical support of the MRC PPU including the DNA Sequencing Service, Tissue Culture team, Reagents and Services team, and the Flow Cytometry & Cell Sorting Facility at the University of Dundee; we thank the (EPI)2 Imaging platform - UMR7216 Epigenetics and Cell Fate centre (Paris) for access to instruments; we also thank N. Wood for help with cloning, and F. Brown and J. Hastie for SPT2 antibody production and purification. We acknowledge the excellent support teams and admin staff in MRC PPU and the School of Life Sciences (University of Dundee) where most of this work was done. This work was supported by the Medical Research Council (grant number MC_UU_00018/5) and the pharmaceutical companies supporting the Division of Signal Transduction Therapy Unit (Boehinger-Ingelheim, GlaxoSmithKline and Merck KGaA) (J.R.); the Wellcome Trust (grant number 217170) and the MRC (grant number MR/S021620/1) (J.A.); the Korean Institute for Basic Science (grant number IBS-R022-A2-2023) (A.G. and S.G.M.R.); Cancer Research UK (grant number C13474/A27826) and the Wellcome Trust (grant number 219475/Z/19/Z) (E.A.M.); the European Research Council (grant number ERC-2018-CoG-818625) (S.E.P.); the Medical Research Council (grant number MC_UU_00007/15) (C.P.P.); the European Research Council (ERC-2016-StG-715127) (C.A.); the Medical Research Council (grant number MC_U105192715 to L. Passmore). G.S. was supported by an EMBO Long-Term Fellowship (ALTF 951-2018) and a SULSA ECR Development Fund; this project has received funding from the European Union's Horizon 2020 research and innovation programme under the Marie Skłodowska-Curie grant agreement no. 845448 (G.S.). G.F. was supported by an EMBO Long-Term Fellowship (ALTF 1132-2018). P.A. was supported by an EMBO Long-Term Fellowship (ALTF 692–2018). For the purpose of open access, the MRC Protein Phosphorylation and Ubiquitylation Unit has applied a CC BY public copyright license to any Author Accepted Manuscript version arising. The funders had no direct role in study design, data collection and analysis, decision to publish or preparation of the manuscript.

## Author contributions

G.S. and J.R. conceived the project. F.N.C. analyzed ChIP–seq, ATAC–seq and mRNA-seq datasets under the supervision of J.A. S.G.M.R. and A.G. analyzed GFP::SPT-2 distribution and transgenerational germline defects. G.F. and L.L. performed the transgenerational gene silencing microscopy assay under the supervision of E.A.M. S.P. assessed SNAP-H3.3 levels after siRNA transfection, under the supervision of S.E.P. A.A. performed ATAC–seq in *C. elegans*, and A.A. and G.S. performed GFP::SPT-2 ChIP–seq. J.L.P. and N.B.R. performed the initial RNA-seq analysis. L.S.-P. and C.P.P. identified the *C. elegans* SPT2 ortholog. A.-C.D. performed electrophoretic mobility shift assay assays. P.A. performed molecular modeling of CeSPT-2. R.T. performed the cloning of human expression vectors. T.M. designed the CRISPR–Cas9 strategy to knock out the *SPTY2D1* gene in human cells. C.A. provided high-content microscopy support. G.S. performed the remaining experiments. G.S. and J.R. wrote the manuscript.

## Competing interests

The authors declare no competing interests. E.A.M. is a founder and director of STORM Therapeutics Ltd. STORM Therapeutics had no role in the design of the study and collection, analysis and interpretation of data as well as in writing the manuscript.

## Additional information

**Extended data** is available for this paper at https://doi.org/10.1038/s41594-023-01204-3.

**Correspondence and requests for materials** should be addressed to Giulia Saredi or John Rouse.

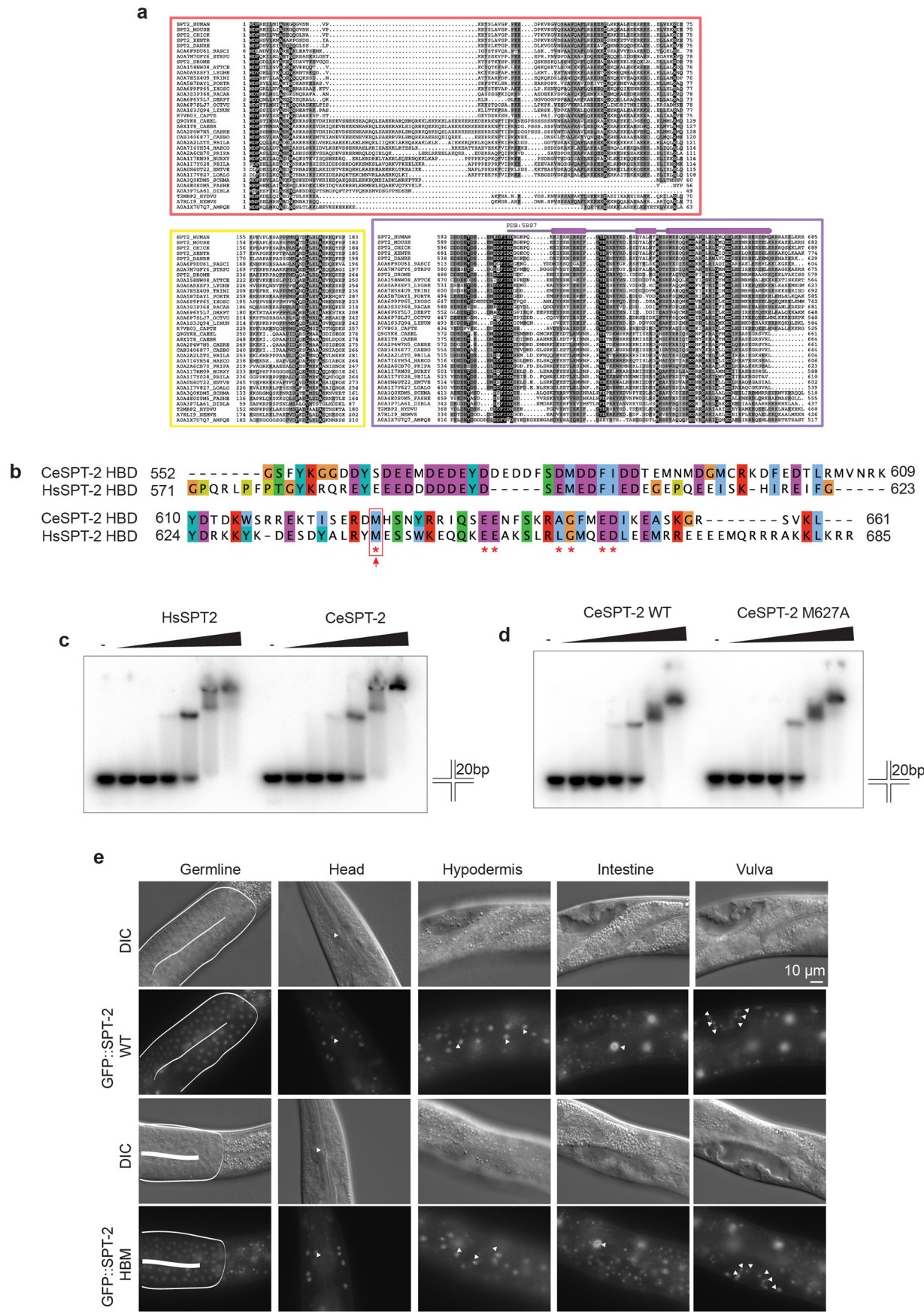

**Extended Data Fig. 1 | See next page for caption.**

**Extended Data Fig. 1 | Conservation and biochemical analysis of CeSPT-2.**
**a**, Multiple protein sequence alignments corresponding to the three conserved regions found in metazoan SPT2 proteins are shown inside coloured boxes in red, yellow, and violet, respectively. Above the alignment that corresponds to the conserved SPT2 C-terminal region (in violet), the secondary structure of the experimentally determined H3-H4 HDB of human SPT2 (PDB: 5BS7 ref. 26) is shown (cylinders correspond to alpha helices). The amino acid colouring scheme indicates the average BLOSUM62 score (correlated to amino acid conservation) in each alignment column: black (greater than 3.5), grey (between 3.5 and 1.5) and light grey (between 1.5 and 0.5). Sequences are named according to their UniProt identifier, and species abbreviations are defined in the Supplementary Information file. **b**, Alignment of human and worm SPT2 histone binding domains. Red asterisks, residues in HsSPT2 reported to be required for the interaction with histones. Red arrow/box indicated the position of the Met residue histone binding defective mutation: M641 (human) / M627 (worm). **c, d**, Electrophoretic mobility shift assay (EMSA) showing binding of full-length recombinant CeSPT-2 or HsSPT2 (c), or of CeSPT-2 wild type and HBM (d), to synthetic cruciform DNA. Each arm of the four-way DNA junction contains 20 base pairs. For each panel, one representative experiment out of two is shown. **e**, Expression of GFP::CeSPT-2 WT or HBM in the indicated tissues of L4 larvae; one representative experiment of two. Images were taken with the same intensity and acquisition time.

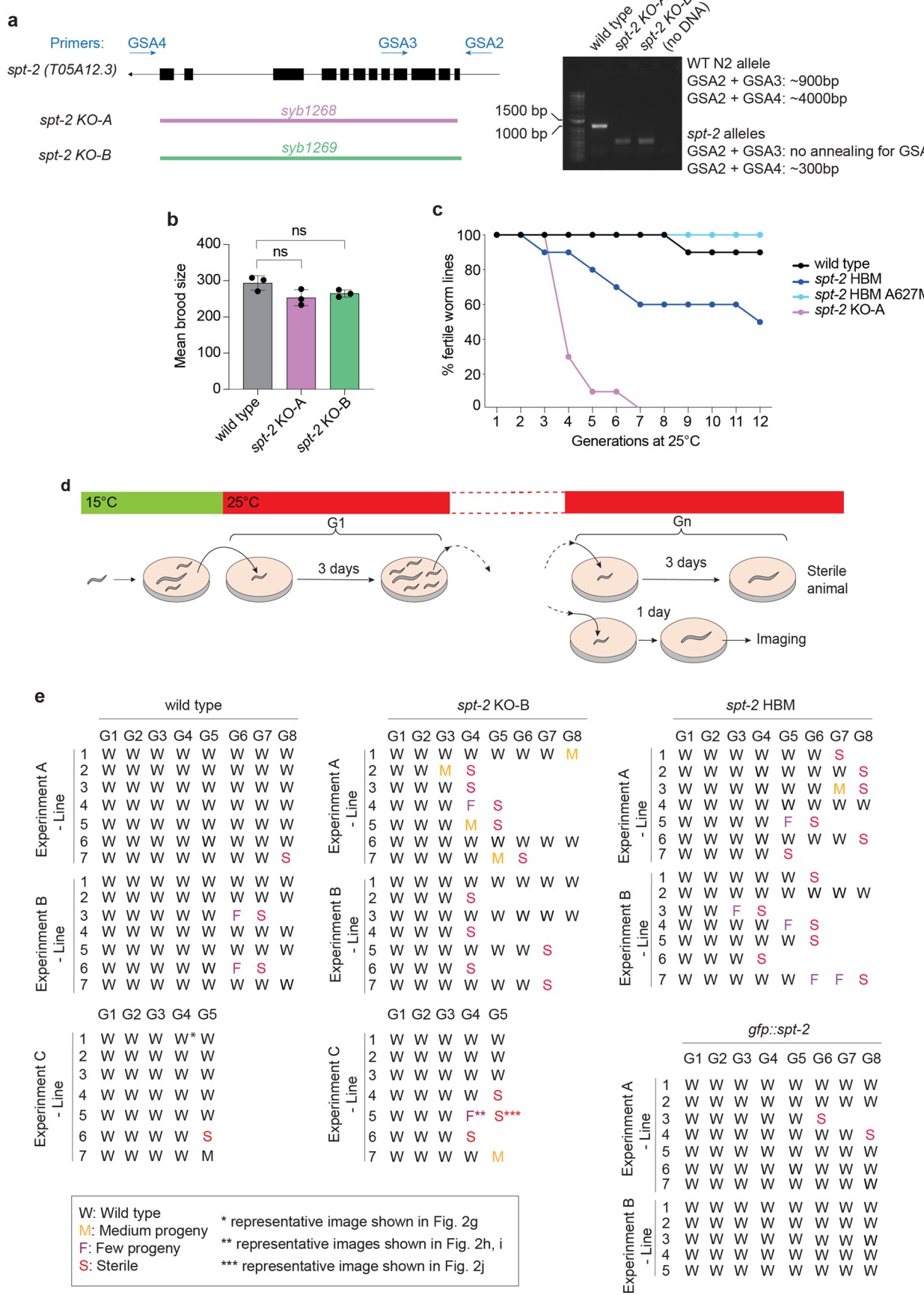

**Extended Data Fig. 2 | See next page for caption.**

**Extended Data Fig. 2 | Further characterization of *spt-2* mutant worms.**
**a**, Schematic of the genotyping strategy for identifying *spt-2* null alleles; primer sequences are provided in the Methods section, together with the strategy for genotyping the *spt-2*^HBM allele. **b**, Mean brood size of the indicated worm strains grown at 20 °C. Worms at the L4 stage were singled onto plates and their total brood was counted. Each point represents the mean brood size of at least 5 worms, and the brood counting experiment was independently repeated three times. n = 3; data are represented as mean ± S.D. Statistical testing: one-way ANOVA with Dunnett's post-tests. ns: P > 0.05. **c**, Transgenerational sterility assay. Three L4 stage worms of the indicated genotypes were shifted to 25 °C and grown at that temperature for the indicated number of generations. Every generation (3 days), three L3-L4 worms were moved to a new plate. A worm line was

considered sterile when no progeny was found on the plate. Ten plates per genotype were used. **d, e**, As indicated in the schematics of the experiment (d), single wild type and *spt-2* mutant L4 worms were shifted to 25 °C and grown at that temperature for the indicated number of generations; every generation (3 days), one L4 worm was moved to a new plate. Siblings of the worms that, 3 days after the L4 stage, showed reduced or no progeny were subjected to microscopic observation 1 day after the L4 stage. The progeny of each worm line was scored after 6 days of growing at 25 °C, and the progeny score indicated in panel (e); scoring criteria: wild type = worms were starved, no more bacteria present; medium = ~100 worms, not starved, bacteria are still present; few = ~30 worms, not starved, bacteria are still present; sterile = no progeny. The indicated number of independent worm lines were used.

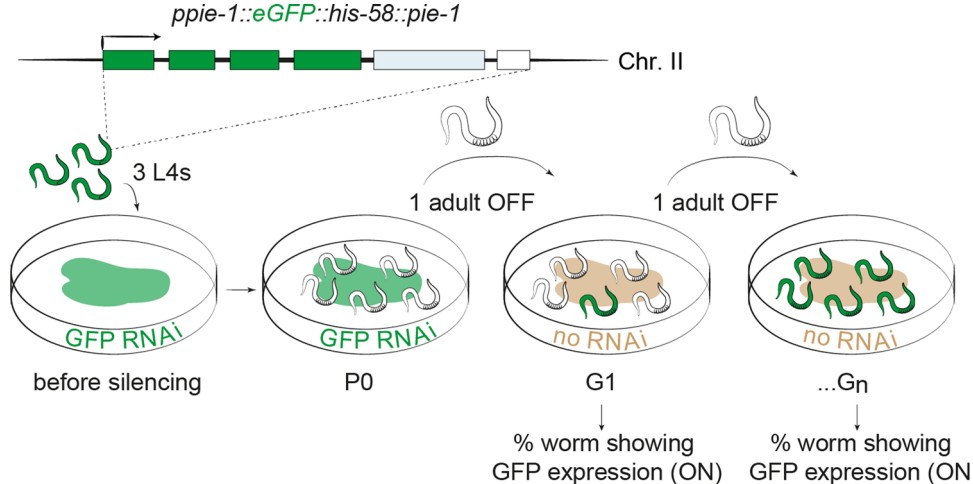

**Extended Data Fig. 3 | Transgenerational RNAi inheritance assay.** Schematics of the RNAi transgenerational inheritance assay, as described previously[72]. Figure adapted from ref. [72].

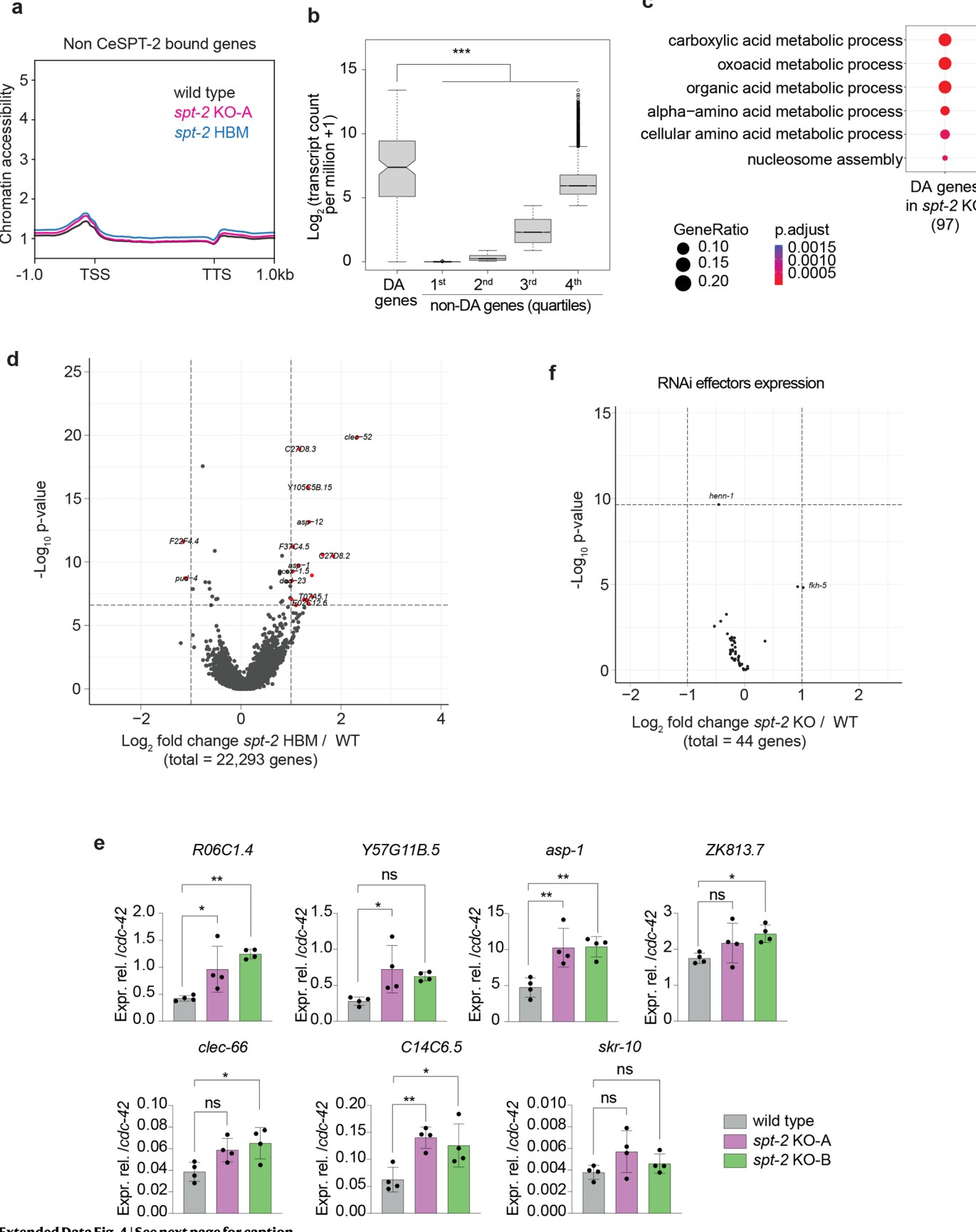

**Extended Data Fig. 4 | See next page for caption.**

**Extended Data Fig. 4 | ATAC-seq and mRNA-seq analysis in *spt-2* mutant worms. a**, Chromatin accessibility at CeSPT-2 non-target genes in *spt-2*[KO-A] and *spt-2*[HBM] adult worms, measured by ATAC-seq and expressed in reads per million. **b**, Expression levels of differentially accessible (DA) genes in *spt-2*[KO] worms, compared to non-DA genes (divided in quartiles). Boxes represent the interquartile range; the thick line represents the median; the whiskers extend up to 1.5-fold of the interquartile range. n = DA gene: 150; 1st quartile: 5536; 2nd: 5536; 3rd: 5536; 4th: 5536. Statistical testing: Mann–Whitney test with Benjamini-Hochberg correction. P values = DA vs 1st: $< 10^{-16}$; DA vs 2nd: $< 10^{-16}$; DA vs 3rd: $< 10^{-16}$; DA vs 4th: $< 10^{-7}$. **c**, Gene Ontology analysis of the DA genes in *spt-2*[KO] worms. 'p.adjust' values indicate Benjamini-Hochberg P values obtained through an hypergeometric test. **d**, Volcano plot of gene expression levels in *spt-2* HBM versus wild type worms. Red points indicate genes with FDR<0.001 and $\log_2$ fold change >1 or <−1. **e**, qPCR analysis for the indicated genes, relative to *cdc-42* expression. Each point indicates an independent replicate with worms harvested on different days, n = 4. The RNA used for the qPCR was harvested independently from the RNA used for the mRNA-seq analysis. Statistical testing: one-way ANOVA with Dunnett's post-test. ns: P > 0.05, *: P < 0.05, ** P < 0.01, *** P < 0.0001. One-way ANOVA results, *R06C1.4*: P = 0.0038; *Y57G11B.5*: P = 0.0269; *asp-1*: P = 0.0035; *ZK813.7*: P = 0.0669; *clec-66*: P = 0.0255; *C14C6.5*: P = 0.0094; *skr-10*: P = 0.1607. Exact P values of Dunnett's post-tests are reported in Supplementary Table 3. Data are represented as mean ± S.D. **f**, Expression levels of RNAi effectors genes.

**Extended Data Fig. 5 | Genetic interaction between HIRA-1 and CeSPT-2.** Description of the entire cross of *spt-2*[KO-B] worms with *hira-1* null worms.

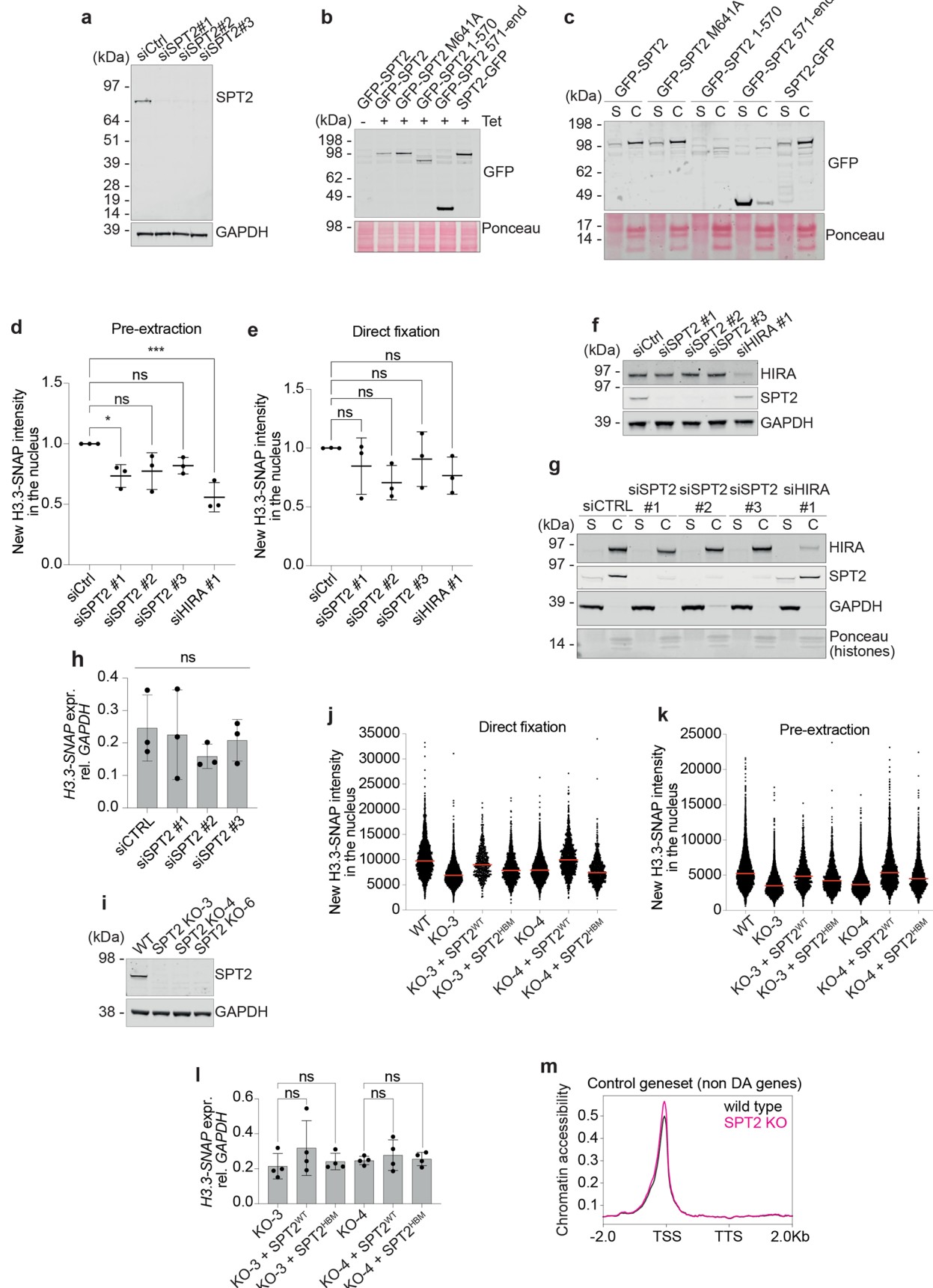

**Extended Data Fig. 6 | See next page for caption.**

**Extended Data Fig. 6 | Validation of SPT2 antibodies and siRNAs, and controls for H3.3-SNAP experiments. a**, Western blot against HsSPT2 in total extracts of SPT2-depleted U-2 OS cells; one representative experiment of three. **b**, Expression of GFP-tagged HsSPT2, full-length or mutants, in U-2 OS Flp-In cells; one representative experiment of two. **c**, Soluble/Chromatin fractionation of U-2 OS Flp-In cells expressing GFP-tagged HsSPT2, full-length or mutants; one representative experiment of two. **d**, **e**, H3.3-SNAP U-2 OS cells were treated with the indicated siRNAs, and newly synthetized H3.3-SNAP was labelled 48 hours post-transfection. New TMR-labelled H3.3-SNAP fluorescence intensity after (d) pre-extraction or (e) direct fixation is shown. n = 3, data represented as mean ± S.D. Statistics: one-way ANOVA with Dunnett's post-test. ns: P > 0.05, *: P < 0.05, ***: P < 0.0001. One-way ANOVA results, Pre-extraction: P = 0.0048, Direct fixation: P = 0.3447. Supplementary Table 3: P values of Dunnett's post-tests. **f**, Total extracts of siRNA-treated U-2 OS cells, 48 hours post-transfection. One representative experiment of two. **g**, Soluble/Chromatin fractionation of siRNA-treated U-2 OS cells, 48 hours post-transfection. One representative experiment of two. **h**, qPCR for H3.3-SNAP expression in cells harvested

48 hours post-transfection. n = 3, data represented as mean ± S.D. Statistics: one-way ANOVA with Dunnett's post-test. One-way ANOVA results, P = 0.4767. Supplementary Table 3: P values of Dunnett's post-tests. **i**, SPT2 levels in the indicated H3.3-SNAP U-2 OS cell lines; one representative experiment of three. **j**, **k**, New TMR-labelled SNAP-H3.3 fluorescence intensity in the indicated U-2 OS SNAP-H3.3 cell lines after (j) direct fixation or (k) pre-extraction. Red line: median. One representative experiment of four, medians from all experiments plotted in Fig. 6e, f. Direct fixation, WT: n = 2487, KO-3: n = 3891, KO-3 + SPT2$^{WT}$: n = 572, KO-3 + SPT2$^{HBM}$: n = 1278, KO-4: n = 4693, KO-4 + SPT2$^{WT}$: n = 1302, KO-4 + SPT2$^{HBM}$: n = 924; Pre-extraction, WT: n = 4617, KO-3: n = 6673, KO-3 + SPT2$^{WT}$: n = 1382, KO-3 + SPT2$^{HBM}$: n = 1864, KO-4: n = 6725, KO-4 + SPT2$^{WT}$: n = 2101, KO-4 + SPT2$^{HBM}$: n = 1472. **l**, qPCR for H3.3-SNAP expression. n = 4, data represented as mean ± S.D. Statistics: one-way ANOVA with Dunnett's post-test, calculated independently on SPT2 KO-3 or KO-4 cells. One-way ANOVA results: KO-3: P = 0.3754, KO-4: P = 0.7420. Supplementary Table 3: P values of Dunnett's post-tests. **m**, ATAC-seq levels at a control subset of 100 random, non-differentially accessible genes.

# Reporting Summary

## Statistics

For all statistical analyses, confirm that the following items are present in the figure legend, table legend, main text, or Methods section.

| n/a | Confirmed | |
|---|---|---|
| ☐ | ☒ | The exact sample size (*n*) for each experimental group/condition, given as a discrete number and unit of measurement |
| ☐ | ☒ | A statement on whether measurements were taken from distinct samples or whether the same sample was measured repeatedly |
| ☐ | ☒ | The statistical test(s) used AND whether they are one- or two-sided<br>*Only common tests should be described solely by name; describe more complex techniques in the Methods section.* |
| ☒ | ☐ | A description of all covariates tested |
| ☐ | ☒ | A description of any assumptions or corrections, such as tests of normality and adjustment for multiple comparisons |
| ☐ | ☒ | A full description of the statistical parameters including central tendency (e.g. means) or other basic estimates (e.g. regression coefficient) AND variation (e.g. standard deviation) or associated estimates of uncertainty (e.g. confidence intervals) |
| ☐ | ☒ | For null hypothesis testing, the test statistic (e.g. *F*, *t*, *r*) with confidence intervals, effect sizes, degrees of freedom and *P* value noted<br>*Give P values as exact values whenever suitable.* |
| ☒ | ☐ | For Bayesian analysis, information on the choice of priors and Markov chain Monte Carlo settings |
| ☒ | ☐ | For hierarchical and complex designs, identification of the appropriate level for tests and full reporting of outcomes |
| ☒ | ☐ | Estimates of effect sizes (e.g. Cohen's *d*, Pearson's *r*), indicating how they were calculated |

*Our web collection on statistics for biologists contains articles on many of the points above.*

## Software and code

Policy information about availability of computer code

| Data collection | Microscopy image acquisition was performed with an Imager M2 (Carl Zeiss) (C. elegans images in Fig. 2g-j); an SP8 confocal fluorescence microscope (Leica) (Fig. 3b,e); a Perkin Elmer Operetta imaging software (Fig. 6b, right); an Olympus ScanR automated microscope (Fig. 6b, e,f; Extended Data Fig. 6j,k); a DMI6000 epifluorescence microscope (Leica) and MetaMorph Software (Extended Data Fig. 6d,e). An Odissey CLx LI-COR for used for western blotting. A Typhoon FLA 9500 phosphorimager was used to image EMSA gels; qPCR was performed using a CFX384 real-time PCR system (Bio-Rad). |
|---|---|
| Data analysis | Statistic analysis was performed using Prism 9. Western blot processing was performed using LI-COR EmpiriaStudio software (version 2.3.0.154). Data analysis relied on published software as described in the Methods section. |

For manuscripts utilizing custom algorithms or software that are central to the research but not yet described in published literature, software must be made available to editors and reviewers. We strongly encourage code deposition in a community repository (e.g. GitHub). See the Nature Portfolio guidelines for submitting code & software for further information.

## Data

Policy information about availability of data

All manuscripts must include a data availability statement. This statement should provide the following information, where applicable:
- Accession codes, unique identifiers, or web links for publicly available datasets
- A description of any restrictions on data availability
- For clinical datasets or third party data, please ensure that the statement adheres to our policy

All plasmids and antibodies generated in this study can be requested to the MRC PPU DSTT: https://mrcppureagents.dundee.ac.uk/reagents-from-papers/Rouse-SPT2-paper-1
All NGS datasets have been deposited on GEO with accession number GSE224802.
The previously published crystal structure of SPT2 HBD in complex with H3-H4 was retrieved from PDB under the ID 5BS7.
Other materials generated in this work will be made available upon request.

## Human research participants

Policy information about studies involving human research participants and Sex and Gender in Research.

| | |
|---|---|
| Reporting on sex and gender | No human research participants were used in this study. |
| Population characteristics | No human research participants were used in this study. |
| Recruitment | No human research participants were used in this study. |
| Ethics oversight | No human research participants were used in this study. |

Note that full information on the approval of the study protocol must also be provided in the manuscript.

# Field-specific reporting

Please select the one below that is the best fit for your research. If you are not sure, read the appropriate sections before making your selection.

☒ Life sciences ☐ Behavioural & social sciences ☐ Ecological, evolutionary & environmental sciences

For a reference copy of the document with all sections, see nature.com/documents/nr-reporting-summary-flat.pdf

# Life sciences study design

All studies must disclose on these points even when the disclosure is negative.

| | |
|---|---|
| Sample size | The sample size used in each experiment was not predetermined or formally justified for statistical power. Samples sizes used in our experiments followed the conventions used in the field. |
| Data exclusions | Three mRNA-seq samples were removed after quality control as they did not meet the standard to be used for differential gene expression analysis (also described in the Methods section).<br>Three human ATAC-seq samples were removed: Two replicates (SPT2 KO-6 Replicate 2 and SPT2 KO-4 Replicate 1) were excluded from the analysis due to a lower signal to noise ratio compared to other replicates from KO strains. A wild type sample (WT Replicate 3) was also excluded as it showed differences in its accessibility profile compared to the other wild type replicates. These three excluded samples all clustered separately from other wild-type and mutant samples from the same strain both in PCA and in correlation heatmaps (also described in the Methods section). |
| Replication | The number of replicates for each experiments are indicated in the relevant figure legend. |
| Randomization | Sample randomization was not applicable in this study; as far as we know, randomization is not something that is applied to the kind of experiments that we present in this paper. |
| Blinding | With the exception of the experiment in Fig. 5b, no blinding was performed in this study. All our samples were however processed in parallel and the analysis was independent of the genotype or condition used. |

# Reporting for specific materials, systems and methods

We require information from authors about some types of materials, experimental systems and methods used in many studies. Here, indicate whether each material, system or method listed is relevant to your study. If you are not sure if a list item applies to your research, read the appropriate section before selecting a response.

## Materials & experimental systems

| n/a | Involved in the study |
|---|---|
| ☐ | ☒ Antibodies |
| ☐ | ☒ Eukaryotic cell lines |
| ☒ | ☐ Palaeontology and archaeology |
| ☐ | ☒ Animals and other organisms |
| ☒ | ☐ Clinical data |
| ☒ | ☐ Dual use research of concern |

## Methods

| n/a | Involved in the study |
|---|---|
| ☐ | ☒ ChIP-seq |
| ☒ | ☐ Flow cytometry |
| ☒ | ☐ MRI-based neuroimaging |

## Antibodies

| | |
|---|---|
| Antibodies used | Primary antibodies: Human SPT2 (in-house produced, sheep DA010, 3rd bleed; WB: 1:2000, IF: 1:1000); GAPDH (Cell Signalling, clone 14C10, 2118S; WB: 1:5000); GFP (Abcam, ab290; WB: 1:2000); RPB1 (Cell Signalling, clone D8L4Y, 14958S; WB: 1:2000); HIRA (Active Motif, 39457; WB: 1:1000); His6 (Abcam, ab18184; 1:1000), H3 (Abcam, ab1971; 1:1000).<br>Secondary antibodies: IRDye antibodies (LI-COR) for western blotting, 1:15,000; Alexa Fluor antibodies (ThermoFisher) for immunofluorescence, 1:1000. |
| Validation | The human SPT2 antibody was validated by siRNA-mediated depletion of SPT2 (Extended Data Fig. 6a).<br>The HIRA 39457 antibody was validated by western blotting after siRNA depletion (Extended Data Fig. 6f,g).<br>The GFP ab290 antibody has been extensively used to perform ChIP-seq, such as in Gal et al. 2021 Cell Reports, and is further validated as per the manufacturer's website: https://www.abcam.com/gfp-antibody-ab290.html<br>The RPB1 14958S antibody is validated as per the manufacturer's website: https://www.cellsignal.com/products/primary-antibodies/rpb1-ntd-d8l4y-rabbit-mab/14958?N=0+102236+4294956287&Nrpp=200&No=4600&fromPage=plp<br>The GAPDH 2118S antibody is validated as per the manufacturer's website: https://www.cellsignal.com/products/primary-antibodies/gapdh-14c10-rabbit-mab/2118<br>The His6 ab18184 antibody is validated as per the manufacturer's website: https://www.abcam.com/6x-his-tag-antibody-hish8-ab18184.html<br>The H3 ab1791 antibody is validated as per the manufacturer's website: https://www.abcam.com/histone-h3-antibody-nuclear-marker-and-chip-grade-ab1791.html |

## Eukaryotic cell lines

Policy information about cell lines and Sex and Gender in Research

| | |
|---|---|
| Cell line source(s) | U-2 OS cells were obtained from ATCC. HEK293 cells were obtained from Kristian Helin's lab. U-2 OS Flp-In cells were obtained from Karmella Haynes' lab. |
| Authentication | U-2 OS Flp-In cells over-expressing different versions of GFP-tagged SPT2 were validated by western blotting. Confirmation of the knock-out of the SPTY2D1 gene was performed by western blot and by sequencing. |
| Mycoplasma contamination | All human cell lines used in this study were routinely tested negative for mycoplasma. |
| Commonly misidentified lines (See ICLAC register) | No commonly misidentified cells were used in this study. |

## Animals and other research organisms

Policy information about studies involving animals; ARRIVE guidelines recommended for reporting animal research, and Sex and Gender in Research

| | |
|---|---|
| Laboratory animals | Caenorhabditis elegans nematodes. In our Methods section, we indicate the 'age' of the C. elegans worms by stating the larval stage at which the worms are used; for adult worms, we state how many days post-larval stage the worms were used. |
| Wild animals | No wild animals were involved in the study. |
| Reporting on sex | The C. elegans nematodes used in this study are hermaphrodites.<br>U-2 OS human cells are derived from a female donor. |
| Field-collected samples | No samples were collected from the field. |
| Ethics oversight | No ethical approval was required for working with C. elegans animals. |

Note that full information on the approval of the study protocol must also be provided in the manuscript.

# ChIP-seq

## Data deposition

☒ Confirm that both raw and final processed data have been deposited in a public database such as GEO.

☒ Confirm that you have deposited or provided access to graph files (e.g. BED files) for the called peaks.

| Data access links<br>*May remain private before publication.* | https://www.ncbi.nlm.nih.gov/geo/query/acc.cgi?acc=GSE224802 |
|---|---|

| Files in database submission | |
|---|---|
| | GSE243274 Investigating how the SPT2 histone chaperone affects chromatin structure in human osteosarcoma cells.<br>GSM7782685 WT, replicate 1<br>GSM7782686 WT, replicate 2<br>GSM7782687 WT, replicate 3<br>GSM7782688 SPT2 KO-3, replicate 1<br>GSM7782689 SPT2 KO-3, replicate 2<br>GSM7782690 SPT2 KO-3, replicate 3<br>GSM7782691 SPT2 KO-4, replicate 1<br>GSM7782692 SPT2 KO-4, replicate 2<br>GSM7782693 SPT2 KO-4, replicate 3<br>GSM7782694 SPT2 KO-6, replicate 2<br>GSM7782695 SPT2 KO-6, replicate 3<br>------------------------------------------------------------------<br>GSE224799 Investigating how the SPT-2 histone chaperone affects chromatin structure in C. elegans.<br>GSM7032552 WT, replicate 1<br>GSM7032553 WT, replicate 2<br>GSM7032554 spt-2 KO-A, replicate 1<br>GSM7032555 spt-2 KO-A, replicate 2<br>GSM7032556 spt-2 M627A, replicate 1<br>GSM7032557 spt-2 M627A, replicate 2<br>------------------------------------------------------------------<br>GSE224800 Identification of GFP-SPT-2 binding sites in the Caenorhabditis elegans genome<br>GSM7032558 gfp::spt-2 C. elegans, GFP, replicate 1<br>GSM7032559 gfp::spt-2 C. elegans, GFP, replicate 2<br>GSM7032560 WT N2 C. elegans, GFP, replicate 1<br>GSM7032561 WT N2 C. elegans, GFP, replicate 2<br>GSM7032562 WT N2 C. elegans, input<br>GSM7032563 GFP-SPT-2 C. elegans, input<br>------------------------------------------------------------------<br>GSE224801 Differential gene expression in spt-2 mutant worms.<br>GSM7032564 WT, 20C, replicate 1<br>GSM7032565 WT, 20C, replicate 3<br>GSM7032566 spt-2 KO A, 20C, replicate 1<br>GSM7032567 spt-2 KO A, 20C, replicate 3<br>GSM7032568 spt-2 KO B, 20C, replicate 1<br>GSM7032569 spt-2 KO B, 20C, replicate 2<br>GSM7032570 spt-2 KO C, 20C, replicate 3<br>GSM7032571 WT, 20C, replicate 4<br>GSM7032572 WT, 20C, replicate 5<br>GSM7032573 WT, 20C, replicate 6<br>GSM7032574 spt-2 M627A, 20C, replicate 1<br>GSM7032575 spt-2 M627A, 20C, replicate 2 |

| Genome browser session<br>(e.g. UCSC) | BigWig files have been provided in the GEO database. |
|---|---|

## Methodology

| Replicates | Two independent replicates of the ChIP-seq experiment were performed, with worms from the two replicates collected independently on different days. |
|---|---|
| Sequencing depth | - GFP IP (gfp::spt2) rep 1 32,169,120 pairs<br>- GFP IP (gfp::spt2) rep 2 29,237,604 pairs<br>- GFP IP (WT) rep 1 24,313,284 pairs<br>- GFP IP (WT) rep 2 28,602,884 pairs<br>- input gfp::spt-2 29,410,305 pairs<br>- input WT 28,175,667 pairs |
| Antibodies | GFP antibody: ab290 |
| Peak calling parameters | CeSPT-2 ChIP-seq peaks were called using MACS2 (settings: --SPMR --gsize ce --keep-dup all --nomodel --broad) using the GFP ChIP-seq samples from wild type animals as controls. The final SPT-2 peak set was defined by the intersection of the broad peaks called on |

each replicate.

A gene was considered an CeSPT-2 target if more than 50% of the length of its longest annotated transcript was covered by a CeSPT-2 peak. The average CeSPT-2 coverage over CeSPT-2 targets was calculated using coverageBed from the BEDTools suite77 (v.2.30.0). Coverage plots over gene models were produced using the DeepTools suite78 (version 3.5.1).

**Data quality**

ChIP-seq reads were preprocessed using trim-galore (version 0.6.7) and mapped on the C. elegans genome (wormbase release WS285) using bwa-mem (version 0.7.17)70. Reads with mapping quality (MAPQ) higher than 10 were extracted using Samtools71. ChIP-seq peaks were called using MACS2 (settings: --SPMR --gsize ce --keep-dup all --nomodel --broad).

**Software**

Trim-galore (version 0.6.7); bwa-mem (version 0.7.17); BEDTools suite75 (v.2.30.0); DeepTools suite76 (version 3.5.1).

