## [Peer Review File · Nature Structural & Molecular Biology]

Peer Review Information

Manuscript Title: The histone chaperone SPT2 regulates chromatin structure and function in Metazoa

Corresponding author name(s): John Rouse, Giulia Saredi

Reviewer Comments & Decisions:

Decision Letter, initial version:

Message: 16th Mar 2023

Dear Dr. Rouse,

Thank you again for submitting your manuscript "The histone chaperone activity of SPT2 controls chromatin structure and function in Metazoa". We now have comments (below) from the 3 reviewers who evaluated your paper. In light of those reports, we remain interested in your study and would like to see your response to the comments of the referees, in the form of a revised manuscript.

You will see that all reviewers appreciate the discovery and characterisation of the worm SPT2 orthologue, commending on how well-written the manuscript is. However, all three reviewers raise important technical issues and serious mechanistic/functional concerns that must be addressed in a revision. More specifically, both reviewer #1 (R#1) and #3 (R#3) note that there is little supporting mechanistic evidence about how loss of *spt-2* induces fertility loss in worms and leads to defective transgenerational RNAi-induced silencing. R#1 additionally comments on the lack of pertinent epistasis experiments between *hira-1* and *spt-2* that would substantially strengthen this work, especially if performed in both the worm and human cells. Furthermore, R#3 notes that there is limited mechanistic insight in how *spt-2* exerts its roles (points 9, 10, 12, 15) and questions the disconnect between the modest functional implications on gene expression upon loss of *spt-2* and macroscopic phenotypes (how does *spt-2* lead to a global stress response; point 16). Finally, R#3 comments on the lack of corresponding, directly comparable experiments in both human and *c.elegans* *spt-2* (e.g. ATAC-seq and RNA-seq in human cells, H3.3. experiments in the worm; see final comments from before point 17 onwards). We editorially agree that substantially strengthening the mechanistic and functional insight according to these guidelines of the referees will be paramount for a successful peer-review process of this story. We also expect you to address all technical issues raised, following the suggestions of all three reviewers (primarily R#2 and R#3). As always, please be sure to address/respond to all concerns of the referees in full in a point-

by-point response and highlight all changes in the revised manuscript text file. If you have comments that are intended for editors only, please include those in a separate cover letter.

We expect to see your revised manuscript within 6 months. If you cannot send it within this time, please contact us to discuss an extension; we would still consider your revision, provided that no similar work has been accepted for publication at NSMB or published elsewhere.

Reporting Summary:

When submitting the revised version of your manuscript, please pay close attention to our [href="https://www.nature.com/nature-portfolio/editorial-policies/image-integrity">Digital Image Integrity Guidelines. and to the following points below:](https://www.nature.com/nature-portfolio/editorial-policies/image-integrity)

Please note that all key data shown in the main figures as cropped gels or blots should be presented in uncropped form, with molecular weight markers. These data can be aggregated into a single supplementary figure item. While these data can be displayed in a relatively informal style, they must refer back to the relevant figures. These data should be submitted with the final revision, as source data, prior to acceptance, but you may want to start putting it together at this point.

SOURCE DATA: we urge authors to provide, in tabular form, the data underlying the graphical representations used in figures. This is to further increase transparency in data reporting, as detailed in this editorial

(<http://www.nature.com/nsmb/journal/v22/n10/full/nsmb.3110.html>). Spreadsheets can be submitted in excel format. Only one (1) file per figure is permitted; thus, for multi-paneled figures, the source data for each panel should be clearly labeled in the Excel file; alternately the data can be provided as multiple, clearly labeled sheets in an Excel file. When submitting files, the title field should indicate which figure the source data pertains to. We encourage our authors to provide source data at the revision stage, so that they are part of the peer-review process.

DATA AVAILABILITY: this journal strongly supports public availability of data. All data used in accepted papers should be available via a public data repository, or alternatively, as Supplementary Information. If data can only be shared on request, please explain why in your Data Availability Statement, and also in the correspondence with your editor. Please note that for some data types, deposition in a public repository is mandatory - more information on our data deposition policies and available repositories can be found below: <https://www.nature.com/nature-research/editorial-policies/reporting-standards#availability-of-data>

Deposition of deep sequencing and microarray data is mandatory, and the datasets must be released prior to or upon publication. To avoid delays in publication, dataset accession numbers must be supplied with the final accepted manuscript and appropriate release dates must be indicated at the galley proof stage.

[Redacted]

Sincerely,

Dimitris Typas

Associate Editor
Nature Structural & Molecular Biology
ORCID: 0000-0002-8737-1319

Referee expertise:

Referee #1: C.elegans chromatin

Referee #2: H3.3, HIRA, histone chaperones, chromatin accessibility

Referee #3: H3.3, histone chaperones, chromatin accessibility

Reviewers' Comments:

Reviewer #1:

Remarks to the Author:

This study reports on the phenotypic consequences of deleting or mutating the histone chaperon SPT-2 in *C. elegans*. After making the case that the gene T05A12.3 is the *C. elegans* orthologue of SPT2, the authors demonstrate broad and nuclear expression of the protein, where it binds to highly expressed genes, and as such influence their chromatin composition. Complete loss of the protein (or in minor way corrupting its chromatin association) affects the worm's fertility (in a pleiotropic fashion) at elevated temperature and the inheritance of gene silencing induced by RNAi. The worm is also used to report on a genetic link between SPT-2 and the histone chaperone HIRA, while mammalian cells are used to demonstrate evolutionary conservation of SPT2's role in controlling H3/H3.3 levels at active genes.

Overall, I find the paper very well written and the elegantly designed experiments carefully executed. In my opinion, the data is convincing and in support of the claims made in the abstract and title. I would also like to acknowledge the authors for not overstating; the study was a real pleasure to read.

I guess the only reservation I have relates to the novelty in relation to the broad readership of the journal. Some of the presented data (mostly the cellular work) is completely expected based on what was previously reported in yeast, whereas the additional phenotypic consequences of SPT-2 loss are either somewhat generic or without depth or mechanistic underpinning: i.e. the work does not reveal in what way loss of spt-2 induces fertility issues (e.g. by identifying the triggers), nor does it provide insight on how the perturbed chromatin structure in spt-2 animals affect transgenerational inheritance of RNAi-induced gene silencing. All the relevant follow-up questions are raised in the discussion section but I guess I would have expected some more insight into the underlying biology to make the manuscript an obvious candidate for NSMB.

As for suggestions to improve the manuscript: the part on the genetic interaction of spt-2 and hira-1 needs some improvement. Given the fact that both spt-2 and hira-1 null worm strains are viable, I consider an RNAi approach (with admitted potential partial levels of depletion, line 299) sub-standard. The observation of a mutant RNAi phenotype of spt-2 mutant animals (while I realize it is not in establishing silencing and, if true, it would actually work against the reported outcomes) adds to this thought. The corroborating

genetic experiment is also easy: either create (new) null alleles in hira-1 null worms through Crispr (which may not be possible) or by analysing double mutant animals generated by crossings (or demonstrate non-mendelian segregation of the alleles when in combination).

Reviewer #2:

Remarks to the Author:

This is an interesting manuscript identifying and characterizing the metazoan orthologs of the Spt2 histone chaperone, originally described in yeast. The authors convincingly show that they identified an Spt2 protein in *C.elegans*, which binds H3-H4 histones, and that this protein controls histone dynamics on chromatin (and to some extent transcriptional activity), a function it shares with its human counterpart, with phenotypical consequences on long-term fertility in worms.

The manuscript is well-written, experiments are sound and conclusions are generally convincing. I can recommend considering this paper for publication, although I would still encourage the authors to take into account the detailed points below to make their study an even stronger piece.

Specific comments:

1) Epigenomic mapping of CeSPT-2 is performed by using a GFP-tagged construct in ChIP-seq. Have the authors confirmed that the a GFP::CeSTP2 construct is functional (i.e that this construct can rescue a mutant and/or that the endogenously tagged worms have a WT phenotype)?

2) The authors show that high CeSPT-2 occupancy genes display relatively higher transcript counts. Have they conversely checked if all highly active genes are more enriched for CeSPT2? This would be interesting to understand if there is some specificity to CeSPT2 recruitment beyond it being attracted to transcribed regions.

3) Related to the effect of CeSPT2 mutants in RNAi silencing, have the authors specifically looked at a potential up/downregulation of RNAi pathway effectors?

4) The notion that one should expect "more chromatin accessibility to result in increased gene expression" is not obvious. Other studies cited by the authors in contrast find lowered transcriptional activity when chaperones are disrupted (eg the human HIRA complex). This could simply be because a nucleosome-depleted chromatin substrate is less structured and is in fact detrimental to high-rate transcriptional activity. Perhaps the authors should consider this as an alternative explanation to the relatively modest effect of CeSPT2 mutations on transcriptional activity of CeSTP2 target genes?

5) Have the authors checked that HsSPT2 does not impact deposition of H3.1/.2? The conclusions of the current study stand regardless, but this would enlighten a putatively more general role for SPT2 in histone deposition, versus a dedicated role to transcriptionally active units. I do not believe that steady-state epigenomic occupancy/accessibility data can really tackle this question, but SNAP-tag would be a good tool.

6) Related to this question, have the authors checked whether CeSPT2 (specifically)

affects the H3.3 variant in worms? This could be done by measuring global levels of tagged H3.3 on chromatin, encouraged by the fact that KD of HsSPT2 does result in a global decrease in H3.3-SNAP and that the Steiner lab previously reported such an effect in H3.3-GFP-bearing, HIRA-impaired worms (Delaney et al., 2018). Of particular interest, certain of these H3.3 variants appear to be germline-specific. While no direct sterility phenotype was reported for these histone mutants, it is unclear whether such phenotype could be progressively acquired across generations.

7) The text states that upregulated genes analyzed in Fig 5g are enriched for stress-response transcripts, but this is not shown in the Figure or a supplement.

Reviewer #3:

Remarks to the Author:

This manuscript from Saredi et al explores the function of the poorly studied histone chaperone SPT2 in *C. elegans* and human (U2OS osteosarcoma) cell culture model systems. The main conclusions are that SPT2 is enriched at active genes to help maintain nucleosome occupancy. This involves supporting H3.3 deposition in U2OS cells. SPT2 depletion or deletion leads to general stress response and eventually sterility in *C. elegans*. The organismal phenotype is not causally linked with the molecular phenotype put forth, nor can the global stress response (concluded based on RNA-seq) be causally linked to loss of SPT2 or its histone binding activity. Overall, this is a loosely associated set of experiments on SPT2 function which would benefit from improved data and statistical analysis.

The first major focus of this manuscript is the identification of a putative SPT2 orthologue in *C. elegans*. I am generally satisfied that the authors have done this. They show that both full length and the histone binding domain of *C. elegans* SPT2 binds to H3-H4, and that a previously identified mutation reduces binding to H3-H4.

1. Figure 1i is lacking statistics (general point to be addressed throughout the manuscript).
2. Please show DAPI co-stain for nuclei for Fig. S1.

The authors have *spt-2* KO strains generated to assess organismal phenotype. They find that *spt-2* KO results in low-penetrance sterility and germline defects.

3. Details of strategy and validation must be provided. Ideally, this would include a western blot, either with the new antibody generated in this study or the commercially available antibody from Abcam, if these antibodies are able to recognize the *C. elegans* protein.
4. Can the authors please explain the large experimental variability between Fig. 2e and Fig. S2b? Also, what is the biological explanation for the low penetrance observed?
5. The authors list many pleiotropic defects in the germline – can these be quantified?
6. The authors should provide morphological assessment of the HBM mutants with respect to germline defects.

The authors claim that SPT2 is required for transgenerational maintenance of RNAi. Rationale for studying this connection is that *spt-2* KO and *hrde-1* defective strains have a similar phenotype. The observations presented here are somewhat tangential to the

manuscript and do not provide insight into how *spt-2* KO leads to de-silencing of GFP (the reporter transgene used in their assay).

7. All data presented here requires statistics.

8. It is not clear why *spt-2* KO attenuates RNAi-mediated silencing, nor how this data integrates with the rest of the manuscript.

The manuscript turns to genomic approaches to understand SPT2 function on chromatin. The major conclusion is that SPT2 is enriched at highly expressed genes and helps to maintain chromatin occupancy. However, loss of SPT2 has little effect on transcription of its targets, and instead results in a global stress response.

9. Is transcription required for SPT2 recruitment to chromatin? What does an SPT2 ChIP-seq look like in the presence of molecules that inhibit either transcription initiation or elongation?

10. Figure 4 title states that histone binding activity of SPT2 controls chromatin accessibility. This is not formally demonstrated. The authors should perform ChIP-seq of the HBM to determine whether they can make this claim.

11. Figure is lacking statistics throughout.

12. SPT2 enrichment seems generally associated with transcription elongation compared with recruitment to the promoter (see Fig. 4e). However, *spt-2* KO effects chromatin accessibility at the promoter. How do the authors characterize / explain the effects on chromatin accessibility in the *spt-2* KO?

13. Fig. 4h – authors state they are displaying RNA-seq of SPT2 non-target genes (line 238) and then state that only 40 of 605 upregulated genes are SPT2 targets. Please clarify which gene sets are used for this analysis.

14. Line 245 – HBM RNA-seq data is under-analyzed and under-represented. Volcano plots such as those shown for *spt-2* KO in Fig. 4h should also be shown for the HBM strain.

Based on the ATAC-seq shown in Fig. 4d and Fig. S3B, loss of SPT2 or HBM show very similar increases in chromatin accessibility. Why, then, are the changes in transcription more subtle? Authors state on line 247 that “80% of the genes upregulated in *spt2*-HBM worms are also upregulated in the *spt2*-KO strain”. What are these numbers?

15. Based on Fig. 4f, it appears that the genes that are upregulated shown in Fig. 4h do not show corresponding changes in chromatin accessibility. Fig. 4h displays only a fold-change in expression of *spt-2* KO compared to WT, but the relative level of expression for these genes is unclear. I would like to know the expression level of these genes in *spt-2* KO compared to the quartiles of expression in WT as shown in Fig. 2c.

16. If I understand the argument that the authors are making correctly, the global stress response is not due to increased chromatin accessibility at active genes. Based on the supplement, non-SPT2 targets show very little change in chromatin accessibility in the absence of SPT2. It is not clear, then, why loss of SPT2 should result in global stress response, and this critical point is never adequately addressed in this manuscript.

The final section of the manuscript transitions to the study of the role of SPT2 in H3.3 deposition in a human osteosarcoma cell line model. There is a general lack of concordance between the *C. elegans* section and the human cell line work. First, I do not understand why the authors do not perform ATAC-seq and/or RNA-seq in the cell line model to validate the *C. elegans* result. Further, *C. elegans* also expresses H3.3, so the authors could have determined the effect of SPT2 loss on H3.3 in that model. Regardless, the main conclusion of this section is that SPT2 controls both H3.3 levels and its incorporation into chromatin. These data are generally poorly supported, with little insight

into how either of those two functions is carried out by SPT2.

17. The authors use truncation constructs to study the structure-function of SPT2. They use GFP-tagged SPT2 constructs and fluorescence microscopy (either without or with pre-extraction) to conclude functional effects of different parts of SPT2 in recruitment to chromatin. First, I would like to see orthogonal methods presented here, like the biochemical fractionation shown in Fig. 5a, or SPT2 ChIP to show recruitment to chromatin. Further, could something like FRET be used to gain a more quantitative view of how well and with what kinetics SPT2 is retained on chromatin? Second, the authors state that SPT2(1-570) lacking the histone binding domain is still recruited to chromatin to a similar degree as full length protein (line 263) based on staining shown in Fig. S4c. I'd like to see this same experiment with the HBM, and to know what is recruiting / retaining SPT2 at chromatin if it is not interaction with histones.

18. Line 306 – authors refer to SPT2 as a histone chaperone, but I do not think there is sufficient evidence either in the literature or this submission to make that claim.

19. Is there any evidence to suggest that SPT2 would have preference for canonical vs H3.3 interaction? To my understanding, the mutant being used in this study is involved in H4 interaction.

20. Based on Fig. 5d, the authors conclude that newly synthesized H3.3 protein levels are lower in SPT2 KD compared to WT cells. What role does SPT2 play in synthesizing or stabilizing H3.3 protein? Is reduced H3.3 deposition due to a transcriptional defect in these cells? It is difficult to conclude that SPT2 has a role in H3.3 deposition if SPT2 is affecting H3.3 protein levels.

21. In general, data from the cell line model are difficult to interpret because the genomic assessment of SPT2 KD in this model are incomplete.

Author Rebuttal to Initial comments

Response to Reviewers

We thank the three Reviewers their thoughtful and constructive comments on our study. We have attempted to address all of their comments, which resulted in a range of new experiments which we have included in the revised manuscript. We believe the new data greatly strengthen the manuscript. New data are shown in revised figures: 2k, S2e, S4b, c, d, f, 5a, 6b, d-h, Fig. S6b-e, i-m.

Reviewer #1

Overall, I find the paper very well written and the elegantly designed experiments carefully executed. In my opinion, the data is convincing and in support of the claims made in the abstract and title. I would also like to acknowledge the authors for not overstating; the study was a real pleasure to read.

We are glad the Reviewer felt so positive about our study.

I guess the only reservation I have relates to the novelty in relation to the broad readership of the journal. Some of the presented data (mostly the cellular work) is completely expected based on what was previously reported in yeast, whereas the additional phenotypic consequences of SPT-2 loss are either somewhat generic or without depth or mechanistic underpinning: i.e. the work does not reveal in what way loss of spt-2 induces fertility issues (e.g. by identifying the triggers), nor does it provide insight on how the perturbed chromatin structure in spt-2 animals affect transgenerational inheritance of RNAi-induced gene silencing. All the relevant follow-up questions are raised in the discussion section but I guess I would have expected some more insight into the underlying biology to make the manuscript an obvious candidate for NSMB.

This is a fair point, and ideally we would like to be at a more advanced stage in terms of the molecular mechanisms underlying SPT2 function. In mitigation though, it's important to note that almost nothing was known about SPT2 in Metazoa when we started; there was no known worm orthologue, and there was no evidence that SPT2 affects chromatin structure. It took multiple collaborating teams almost 4 years of hard work to show convincingly that the histone binding capacity of SPT2 in worms and human is involved in maintaining chromatin structure and protecting cell and organism function in collaboration with HIRA. We feel that even without further mechanistic insights, this work is of sufficiently broad interest and novelty for a first paper on the topic. In the revised version of the SPT2 paper, we have expanded the section of human SPT2; we made SPT2 KO cell lines and rescued them with wild-type and the HBM mutant, and did ATAC-seq and other experiments which point to a role for SPT2 histone binding in influencing chromatin structure in humans. We feel that this view across evolution of SPT2 as a *bona fide* chromatin regulator will be of sufficient novelty and general interest to justify publication in NSMB.

As for suggestions to improve the manuscript: the part on the genetic interaction of spt-2 and hira-1 needs some improvement. Given the fact that both spt-2 and hira-1 null worm strains are viable, I consider an RNAi approach (with admitted potential partial levels of depletion, line 299) sub-standard. The observation of a mutant RNAi phenotype of spt-2 mutant animals (while I realize it is not in establishing silencing and, if true, it would actually work against the reported outcomes) adds to this thought. The corroborating genetic experiment is also easy: either create (new) null

alleles in *hira-1* null worms through Crispr (which may not be possible) or by analysing double mutant animals generated by crossings (or demonstrate non-mendelian segregation of the alleles when in combination). This is a great suggestion. Accordingly, we have strengthened the data on the genetic interaction between CeSPT-2 and HIRA-1. Specifically, we have crossed the *spt-2* KO-B worm strain with the *hira-1*(*uge29*) null strain (PMID: 29636369) and we have observed that *hira-1*;*spt-2* double null worms, although born in Mendelian ratios, are fully sterile. The new data are shown in Fig. 5a of the revised manuscript.

Given the synthetic lethality between *spt-2* and *hira-1* in worms (Figure 5a, b), we tried to assess whether combined loss of SPT2 and HIRA in human cells results in an additive defect in new H3.3-SNAP levels. As shown below in Extra Figure 1, we depleted HIRA in SPT2 KO-4 cells, or in SPT2 KO-4 cells rescued with SPT2 WT, and we measured the levels of new H3.3 in directly fixed or pre-extracted cells. In all three replicates, we observed a consistent, albeit statistically non-significant, decrease of new H3.3 levels in cells lacking both SPT2 and HIRA. Alternative strategies would be needed to answer this question definitively in human cells. Nonetheless, the worm data are very clear.

Extra Figure 1. New H3.3-SNAP intensity was measured in SPT2 KO-4 and SPT2 KO4 + SPT2^{WT} cells 48 hours after HIRA depletion by siRNAs, in directly fixed (a) or pre-extracted (b) cells. Statistics: one-way ANOVA with Dunnett's post-test multiple comparisons. ns = P >

0.05, ***P < 0.001. One representative experiment for directly fixed (c) or pre-extracted cells (d), out of three independent replicates, is shown. Red line represents the median.

Reviewer #2

Remarks to the Author:

The manuscript is well-written, experiments are sound and conclusions are generally convincing. I can recommend considering this paper for publication, although I would still encourage the authors to take into account the detailed points below to make their study an even stronger piece.

We are glad the Reviewer felt so positive about our study.

1) Epigenomic mapping of CeSPT-2 is performed by using a GFP-tagged construct in ChIP-seq. Have the authors confirmed that the a GFP::CeSTP2 construct is functional (i.e that this construct can rescue a mutant and/or that the endogenously tagged worms have a WT phenotype)?

We agree that this is an important control. We tested whether *gfp::spt-2* worms become increasingly sterile across generations when grown at 25C, similar to *spt-2* mutant worms. In the panel S2e in the revised manuscript, we now show that fertility in *gfp::spt-2* worms is comparable to wild type worms, which strongly suggests that GFP-tagged CeSPT-2 is functional.

2) The authors show that high CeSPT-2 occupancy genes display relatively higher transcript counts. Have they conversely checked if all highly active genes are more enriched for CeSPT2? This would be interesting to understand if there is some specificity to CeSPT2 recruitment beyond it being attracted to transcribed regions.

We calculated the expression of all worm genes and extracted those with levels of expression in the top 1% (category A), 5% (category B), and 10% (category C) (Extra Figure 2). From analyzing the data, we found that 82% of category A genes are SPT-2 targets (183/223), 74.8% of category B genes are SPT-2 targets (834/1115) and 56.7% of category C genes are SPT-2 targets (1264/2230). Therefore, not all highly active genes are SPT-2 targets but there is a strong correlation between gene activity and CeSPT-2 targeting. The numbers related to the top 5% genes are now mentioned in the Results section. We have not included the graph below in the revised manuscript but if the Reviewer feels strongly that we should, then we will.

Extra Figure 2. Expression levels of *C. elegans* genes were calculated as average TPM from our mRNA-seq dataset in wild type worms, and genes in the top 1%, top 5% or top 10% were extracted. CeSPT-2 enrichment was compared between the top 1%, 5% or 10% of gene expression versus all other genes.

3) Related to the effect of CeSPT2 mutants in RNAi silencing, have the authors specifically looked at a potential up/downregulation of RNAi pathway effectors?

We have checked the expression levels of genes involved in the RNAi pathway (list from WormBook: Endogenous RNAi pathways in *C. elegans*, Billi *et al.*) in *spt-2* KO worms. No obvious de-regulation of RNAi effectors is seen in *spt-2* mutants compared with wild type worms, and the data are shown in Fig. S4f of the revised manuscript. This strongly suggests that the RNAi defects observed in *spt-2* mutant worms are not due to impaired transcription of RNAi effector proteins. We now mention this point in the text describing Figure 4.

4) The notion that one should expect "more chromatin accessibility to result in increased gene expression" is not obvious. Other studies cited by the authors in contrast find lowered transcriptional activity when chaperones are disrupted (eg the human HIRA complex). This could simply be because a nucleosome-depleted chromatin substrate is less structured and is in fact detrimental to high-rate transcriptional activity. Perhaps the authors should consider this as an alternative explanation to the relatively modest effect of CeSPT2 mutations on transcriptional activity of CeSPT2 target genes?

This is a good point, and we have now amended the text section relating to Fig. 4 accordingly.

5) Have the authors checked that HsSPT2 does not impact deposition of H3.1/.2? The conclusions of the current study stand regardless, but this would enlighten a putatively more general role for SPT2 in histone deposition, versus a dedicated role to transcriptionally active units. I do not believe that steady-state epigenomic occupancy/accessibility data can really tackle this question, but SNAP-tag would be a good tool.

Based on structural data (PMID: 26109053), SPT2 should indeed be able to bind both H3.1 and H3.3; moreover, a recent proteomics study identified SPT2 in both H3.1 and H3.3 complexes (PMID: 36868228). Using the SNAP tag system, we have found that siRNA depletion of SPT2 impacts H3.1 soluble and chromatin-bound levels in EdU-positive cells. We have used 5 independent siRNAs as, due to time constraints we could not manage to generate SPT2 KO clones (with a rescue system) in the H3.1-SNAP cell line. Our analysis shows that depletion of SPT2 leads to a reduction in the levels of chromatin bound H3.1-SNAP (4 siRNAs out of 5 showing a significant decrease), with a more modest effect on the total levels of H3.1-SNAP (3 out of 5 siRNAs showing a decrease, Extra Figure 3a-c). Of note, this is not accompanied by a significant decrease in DNA synthesis levels (Extra Figure 3d), which would in turn impact new histone deposition, nor by a decrease in H3.1-SNAP transcription (Extra Figure 3e). These data support a wider role of SPT2 in preserving soluble and chromatin-bound levels of histone H3. The data are presented here as Extra Figure as we were not sure whether they would fit withing the manuscript, but if the Reviewers feels that they should be incorporated, we will add them.

Extra Figure 3. a, b, SPT2 was depleted from H3.1-SNAP U-2 OS cells with the indicated siRNAs. Newly synthesised H3.1-SNAP histones were labelled as described in our Methods section for H3.3-SNAP, with the difference that we used a chase period of 5 hours instead of 2, and that a 30-minute 10 μ M EdU pulse was performed to identify replicating cells. New H3.1-SNAP intensity in the nucleus of EdU-positive cells is indicated, for either directly fixed (a) or pre-extracted (b) cells. Statistics: one-way ANOVA with Dunnett's post-tests. ns: $P > 0.05$, *: $P < 0.05$, ** $P < 0.01$, *** $P < 0.0001$. **c**, Western blot showing SPT2 depletion levels. **d**, EdU intensity in EdU-positive cells. **e**, H3.1-SNAP mRNA levels.

6) Related to this question, have the authors checked whether *CeSPT2* (specifically) affects the H3.3 variant in worms? This could be done by measuring global levels of tagged H3.3 on chromatin, encouraged by the fact that KD of *HsSPT2* does result in a global decrease in H3.3-SNAP and that the Steiner lab previously reported such an effect in H3.3-GFP-bearing, HIRA-impaired worms (Delaney et al., 2018). Of particular interest, certain of these H3.3 variants appear to be germline-specific. While no direct sterility phenotype was reported for these histone mutants, it is unclear whether such phenotype could be progressively acquired across generations.

We checked GFP-H3.3 (GFP::HIS-72) in worms, quantitating the fluorescence levels in germline cells in *spt-2* mutants versus wild-type, but there was no obvious difference in H3.3 intensity. This observation may seem to contrast with the experiment in the original manuscript showing that siRNA-mediated knockdown of *HsSPT2* caused a decrease in SNAP-H3.3. However, it was new SNAP-H3.3 we measured in that experiment, which we also find to be reduced in SPT2 KO human cells, and in KO cells expressing a histone-

binding SPT2 mutant. In contrast, our preliminary data suggested that parental or “old” histone SNAP-H3.3 levels were not affected by SPT2 loss. Given that it’s a specific pool of histones affected by SPT2 loss, it may be that an equivalent decrease in H3.3 in worms may not be apparent by looking at total GFP-H3.3. Therefore, more work will be needed to address this point.

7) The text states that upregulated genes analyzed in Fig 5g are enriched for stress-response transcripts, but this is not shown in the Figure or a supplement.

We agree that the text relating to these data is not clear. In *C. elegans*, it has been observed that transcripts that are up-regulated in response to pathogen infection, and that therefore fall under the ‘defense response’ terms, are also up-regulated in response to a wide array of stress stimuli, such as UV and IR damage, heat stress etc. We have added relevant references to the Result section, and we have amended the Discussion to better explain this.

Reviewer #3

This manuscript from Saredi et al explores the function of the poorly studied histone chaperone SPT2 in C. elegans and human (U2OS osteosarcoma) cell culture model systems. The main conclusions are that SPT2 is enriched at active genes to help maintain nucleosome occupancy. This involves supporting H3.3 deposition in U2OS cells. SPT2 depletion or deletion leads to general stress response and eventually sterility in C. elegans. The organismal phenotype is not causally linked with the molecular phenotype put forth, nor can the global stress response (concluded based on RNA-seq) be causally linked to loss of SPT2 or its histone binding activity. Overall, this is a loosely associated set of experiments on SPT2 function which would benefit from improved data and statistical analysis.

We would like to be at a more advanced stage in terms of the molecular mechanisms underlying SPT2 function. However, when we started this study, almost nothing was known about SPT2 in Metazoa; there was no known worm orthologue, and there was no evidence that SPT2 affects chromatin structure. The data in our paper represent over 4 years of work from a number of collaborating teams, and we feel that the data we have gathered thus far, showing that the histone binding capacity of SPT2 in worms and human is involved in chromatin structure and protecting aspect of cell and organism function in collaboration with HIRA, is of sufficiently broad interest and novelty for a first paper on the topic. We have provided extra statistical analysis where applicable as described below.

1. Figure 1i is lacking statistics (general point to be addressed throughout the manuscript).

We have now added statistics, both in Figure 1 as well as where relevant throughout our manuscript in Fig. 1i, Fig. 3d, Fig. 4c, f, Fig. 6e, f, Fig. S4b, Fig. S6d, e, l.

2. The authors have spt-2 KO strains generated to assess organismal phenotype. They find that spt-2 KO results in low-penetrance sterility and germline defects.

We believe that the Reviewer refers to Fig. 2b when writing about low penetrance. Fig. 2b shows the brood size of a single generation of worms grown at 25C: these worms were moved from growing at 20C to 25C at the L4 stage, and their brood size was quantified over the following 4-5 days. In a way, this assay represents the equivalent of the first generation of worms grown at 25C in the transgenerational sterility assays in shown Figs. 2c-e and S2e. This observation then prompted us to test whether the sterility of *spt-2* mutant worms increases when the worms are grown at 25C over several generations, and indeed this was the case as we observed that the majority of *spt-2* mutant worm lines passaged at 25C becomes sterile (Figs. 2c-e and S2e). We would argue therefore that the sterility and germline defects are not low penetrance phenotypes.

3. Details of strategy and validation must be provided. Ideally, this would include a western blot, either with the new antibody generated in this study or the commercially available antibody from Abcam, if these antibodies are able to recognize the C. elegans protein.

We had tested the ability of our anti-human SPT2 antibodies to recognise worm SPT-2 but they do not, and so they could not be used to validate the KO worm strains. However, in the revised manuscript we have provided a detailed genotyping strategy for the *spt-2* KO and *spt-2* HBM alleles in the Methods section, including primer sequences. We have also added

a schematic of the genotyping strategy in Fig. S2a, as well as the image of a representative agarose gel showing PCR products for wild type and *spt-2* KO worms.

4. Can the authors please explain the large experimental variability between Fig. 2e and Fig. S2b? Also, what is the biological explanation for the low penetrance observed?

Although the data between biological replicates are consistent in qualitative terms, there is indeed a degree of variation between the different replicates of the transgenerational sterility assay; to better represent the data, we now provide two additional independent replicates of the transgenerational sterility assay in Fig. S2e. We note that these assays, as also stated in our Methods, have been independently performed by researchers in two different labs (GS at University of Dundee, UK, and SR at the Gartner lab at IBS, Ulsan, Rep of S. Korea). For the reasons elaborated in the response to point 2 above, we argue that the sterility and germline defects are not low penetrance phenotypes.

5. The authors list many pleiotropic defects in the germline – can these be quantified?

This is an important point. In the revised manuscript, we have quantified the defects present in the germlines of *spt-2* mutant worms grown at 25C (Fig. 2k).

6a. The authors should provide morphological assessment of the HBM mutants with respect to germline defects.

See response to point 5 and new Fig. 2k.

6b. The authors claim that SPT2 is required for transgenerational maintenance of RNAi. Rationale for studying this connection is that *spt-2* KO and *hrde-1* defective strains have a similar phenotype. The observations presented here are somewhat tangential to the manuscript and do not provide insight into how *spt-2* KO leads to de-silencing of GFP (the reporter transgene used in their assay).

The temperature-dependent sterility of the *spt-2* mutant worms was immediately reminiscent of the range of mutants defective in the transgenerational maintenance of RNAi, this was the primary motivation to do the experiments. Moreover, we noticed that Table 1 of Eric Miska's 2012 paper (PMID 22738725) listing the worm genes required for this pathway included a range of chromatin factors including a range of SET domain proteins as well as histone modifying enzymes. It seemed to us that a histone chaperone could easily have a role to play in this pathway, which motivated us further. We would like to have more mechanistic information on the defect we have seen in *spt-2* worms, but the chromatin changes that are necessary for maintenance of silencing in this system are not well understood. H3K9 methylation has been implicated and to this end we recently carried out H3K9me3 ChIP-seq with help from Julie Ahringer's lab but the results were not conclusive, there was too much variability between biological replicates. Nonetheless, we link the relevant sections with the rest of the text.

7. All data presented here requires statistics.

See response to point 1 above.

8. It is not clear why *spt-2* KO attenuates RNAi-mediated silencing, nor how this data integrates with the rest of the manuscript.

See response to point 6b above.

9. Is transcription required for SPT2 recruitment to chromatin? What does an SPT2 ChIP-seq look like in the presence of molecules that inhibit either transcription initiation or elongation?

This is an interesting question. We used quantitative high-content microscopy to test the effect of the transcriptional inhibitors DRB (CDK9 inhibitor) and Triptolide (TPL, TFIIH inhibitor) on human SPT2 binding to chromatin. Briefly, and as shown in Extra Fig. 4 below, we found that blocking transcription does not affect SPT2 binding to chromatin. We have not included these data in the revised manuscript but if the Reviewer felt strongly that we should, then we will.

Extra Figure 4. a, U-2 OS cells were treated with 50µM DRB or 1µM TPL for 2 hours, and then directly fixed or pre-extracted as described in our Methods section. b, Validation of SPT2 antibody for immunofluorescence. c, d, Controls for the efficacy of DRB and TPL treatment, showing decreased RNA Pol-II CTD Ser5 phosphorylation (c) and total levels of RNA Pol-II (d) in fixed cells. e, Chromatin-bound levels of SPT6 are reduced after DRB and TPL. We used SPT6 as positive control in the assay: SPT6 binds phosphorylated RNA Pol-II and treatment with DRB or TPL was expected to reduce its binding to chromatin. f, Chromatin-bound levels of SPT2 are not affected by DRB and TPL treatment.

10. Figure 4 title states that histone binding activity of SPT2 controls chromatin accessibility. This is not formally demonstrated. The authors should perform ChIP-seq of the HBM to determine whether they can make this claim.

We feel that the observation the *spt-2^{HBM}* mutant worms show alterations in accessibility at the target genes justifies the title of figure. A somewhat separate point is whether the histone binding mutation affects binding of CeSPT-2 to target genes. We have not done the corresponding ChIP-seq experiment because of time constraints, but the GFP-CeSPT-2 HBM shows a localisation pattern similar to wild type. Moreover, we have shown clearly in the revised manuscript that in human cells, the HsSPT2 histone binding domain is dispensable for its localisation to chromatin.

11. Figure is lacking statistics throughout.

See response to point 1.

12. SPT2 enrichment seems generally associated with transcription elongation compared with recruitment to the promoter (see Fig. 4e). However, spt-2 KO effects chromatin accessibility at the promoter. How do the authors characterize / explain the effects on chromatin accessibility in the spt-2 KO?

The ChIP-seq data shows that there is more CeSPT-2 associated with target gene bodies than at promoters, but nonetheless CeSPT-2 is also detected at the promoters. Therefore, we feel that these data are not inconsistent.

13. Fig. 4h – authors state they are displaying RNA-seq of SPT2 non-target genes (line 238) and then state that only 40 of 605 upregulated genes are SPT2 targets.

We thank the Reviewer for pointing out this discrepancy, which we have now corrected in the text.

14. Line 245 – HBM RNA-seq data is under-analyzed and under-represented. Volcano plots such as those shown for spt-2 KO in Fig. 4h should also be shown for the HBM strain. Based on the ATAC-seq shown in Fig. 4d and Fig. S3B, loss of SPT2 or HBM show very similar increases in chromatin accessibility. Why, then, are the changes in transcription more subtle? Authors state on line 247 that “80% of the genes upregulated in spt2-HBM worms are also upregulated in the spt2-KO strain”. What are these numbers?

In the revised manuscript, we present numbers for the differentially expressed genes and we show the volcano plot of the differential gene expression in *spt-2* HBM versus wild type worms (Fig. S4d).

15. Based on Fig. 4f, it appears that the genes that are upregulated shown in Fig. 4h do not show corresponding changes in chromatin accessibility. Fig. 4h displays only a fold-change in expression of spt-2 KO compared to WT, but the relative level of expression for these genes is unclear. I would like to know the expression level of these genes in spt-2 KO compared to the quartiles of expression in WT as shown in Fig. 2c.

We're not entirely sure we understand this question. The genes upregulated in *spt-2* KO strains are not CeSPT-2 target genes, and therefore we would not necessarily expect them to show differences in chromatin accessibility in *spt-2* KO worms. These genes are part of a stress response. In the Extra Figure 5 below we present the data requested.

Extra Figure 5. Presented here are the expression levels of differentially expressed genes between wild type and *spt-2* KO worms, divided between up- and down-regulated, and compared to the levels of expression of all genes divided into quartiles.

16. If I understand the argument that the authors are making correctly, the global stress response is not due to increased chromatin accessibility at active genes. Based on the supplement, non-SPT2 targets show very little change in chromatin accessibility in the absence of SPT2. It is not clear, then, why loss of SPT2 should result in global stress response, and this critical point is never adequately addressed in this manuscript.

It is true that the stress response genes upregulated in *spt-2* KO mutants are not CeSPT-2 target genes; they do not show increased accessibility in *spt-2* KO worms, but there are other ways in which these genes could be up-regulated. The underlying basis for activation of a stress response is not yet clear. In the Discussion we mention that in yeast Spt2 prevents spurious transcription. It could be that aberrant transcripts are produced that are recognized as foreign nucleic acids thus launching a stress response. We hope to establish the seq approaches one would need to examine spurious transcripts in worms (CAGE-seq etc) but we feel that setting this up is beyond the scope of the current study.

The final section of the manuscript transitions to the study of the role of SPT2 in H3.3 deposition in a human osteosarcoma cell line model. There is a general lack of concordance between the *C. elegans* section and the human cell line work. First, I do not understand why the authors do not perform ATAC-seq and/or RNA-seq in the cell line model to validate the *C. elegans* result.

We have now substantially strengthened the human cell work by generating SPT2 KO cells which have been used to assess chromatin accessibility levels compared to wild type (ATAC-seq) and to show (with a rescue complementation system) that SPT2 preserves the levels of soluble and chromatin-bound histones via H3-H4 binding. The data are in Fig. 6d-h and S6i-m of the revised manuscript.

17. The authors use truncation constructs to study the structure-function of SPT2. They use GFP-tagged SPT2 constructs and fluorescence microscopy (either without or with pre-extraction) to conclude functional effects of different parts of SPT2 in recruitment to chromatin. First, I would like to see orthogonal methods presented

here, like the biochemical fractionation shown in Fig. 5a, or SPT2 ChIP to show recruitment to chromatin. Further, could something like FRET be used to gain a more quantitative view of how well and with what kinetics SPT2 is retained on chromatin? Second, the authors state that STP2(1-570) lacking the histone binding domain is still recruited to chromatin to a similar degree as full length protein (line 263) based on staining shown in Fig. S4c. I'd like to see this same experiment with the HBM, and to know what is recruiting / retaining SPT2 at chromatin if it is not interaction with histones.

We thank the Reviewer for the suggestion of using FRAP (we believe this is what was suggested), as it could uncover more subtle differences in SPT2 binding to chromatin than a soluble/chromatin fractionation experiment can. Time constraints prevented us from getting to this experiment. However, we have now performed soluble/chromatin fractionation in GFP-SPT2 U-2 OS cell lines (Fig. S6b, c) to recapitulate our findings from Fig. 6b, namely that SPT2 histone binding domain is dispensable for SPT2 binding to chromatin. We have also included a GFP-SPT2 M641A cell line in the experiments performed in Fig. 6b and Fig. S6b, c. The question of how SPT2 is recruited is an interesting one. It may simply recognize open chromatin perhaps through DNA binding or it may be recruited by an associated factor such as SPT6. However, the new data in Extra Fig. 4 show that whereas transcription inhibitors reduce SPT6 association with chromatin, SPT2 association with chromatin is unaffected. These data imply SPT6 does not recruit SPT2 to chromatin. There's only so much information one can include in the first paper describing the function of a new metazoan factor, this study is over 4 years' work done by multiple teams. Determining the mechanism of SPT2 recruitment to chromatin will be important but we feel that it is beyond the scope of this study.

18. Line 306 – authors refer to SPT2 as a histone chaperone, but I do not think there is sufficient evidence either in the literature or this submission to make that claim.

We feel the evidence in the literature is sufficient to justify referring to SPT2 as a histone chaperone. The plasmid supercoiling assay is regarded as a gold standard test for histone chaperone activity, and this assay was used by Osakabe et al. (PMID 23378026) to show that recombinant human SPT2 functions as a histone chaperone. Moreover, Chen et al. (PMID 26109053) provided the structure of SPT2 bound to histone H3-H4, and extensively discuss the function of yeast Spt2 as a histone chaperone, based on extensive genetic data. We list here other papers and reviews that describe SPT2 as a histone chaperone: PMID 28053344, 36868228, 33857403.

19. Is there any evidence to suggest that SPT2 would have preference for canonical vs H3.3 interaction? To my understanding, the mutant being used in this study is involved in H4 interaction.

This is an interesting point. Based on the published structure from Chen et al., SPT2 should not distinguish between H3.1 and H3.3. We have tested whether SPT2 regulates the levels of soluble and chromatin-bound H3.1 using a reporter H3.1-SNAP cell line. Our analysis shows that depletion of SPT2 leads to a reduction in the levels of chromatin bound H3.1-SNAP (4 siRNAs out of 5 showing a significant decrease), with a more modest effect on the total levels of H3.1-SNAP (3 out of 5 siRNAs showing a decrease, Extra Figure 3a-c). We measured the H3.1-SNAP intensity specifically in EdU positive, and therefore replicating, cells. Importantly, neither the levels of EdU nor the expression of the H3.1-SNAP transgene are affected by SPT2 depletion (Extra Figure 3d, e). The data are presented here as Extra

Figure as we were not sure whether they would fit withing the manuscript, but if the Reviewers feels that they should be incorporated, we will add them.

20. Based on Fig. 5d, the authors conclude that newly synthesized H3.3 protein levels are lower in SPT2 KD compared to WT cells. What role does SPT2 play in synthesizing or stabilizing H3.3 protein? Is reduced H3.3 deposition due to a transcriptional defect in these cells? It is difficult to conclude that SPT2 has a role in H3.3 deposition if SPT2 is affecting H3.3 protein levels.

In the revised (and original) manuscript, we present qPCR controls showing that siRNA depletion of SPT2 does not impact H3.3-SNAP transcription (Fig. S6h); similarly, we show similar expression levels of the H3.3-SNAP transgene in our SPT2 KO complementation system (Fig. S6l). These data suggest that SPT2 histone binding regulates their stability. We did some preliminary cycloheximide pulse chase experiments to investigate this idea, but the data were inconclusive and time constraints prevented us from going further. In the revised manuscript, we avoid making the claim that SPT2 deposits H3.3-H4 into chromatin. In fact, in our revised Discussion we discuss histone binding versus histone deposition with regard to the role of SPT2 in regulating chromatin accessibility.

21. In general, data from the cell line model are difficult to interpret because the genomic assessment of SPT2 KD in this model are incomplete.

To strengthen our data in human cells, we have generated SPT2 KO cell lines and we have re-complemented them with SPT2 cDNA WT or HBM (Fig. 6d-f).

Decision Letter, first revision:

Message: Our ref: NSMB-A47344A

20th Oct 2023

Dear Professor Rouse,

Thank you for submitting your revised manuscript "The histone binding capacity of SPT2 controls chromatin structure and function in Metazoa" (NSMB-A47344A). It has now been seen by the original referees and their comments are below. Two of the reviewers find that the paper has further improved in revision, and therefore we are happy to accept it in principle in Nature Structural & Molecular Biology, pending minor revisions to satisfy the referees' final requests, like those of reviewer #3 with respect to limitations of the study, and to comply with our editorial and formatting guidelines.

To facilitate our work at this stage, it is important that we have a copy of the main text as a word file. If you could please send along a word version of this file as soon as possible, we would greatly appreciate it; please make sure to copy the NSMB account (cc'ed above).

Sincerely,

Dimitris Typas
Associate Editor
Nature Structural & Molecular Biology
ORCID: 0000-0002-8737-1319

Reviewer #1 (Remarks to the Author):

The authors went to great length in improving their manuscript following up the suggestion of the reviewers (including mine) via substantial experimentation, of which the data is now included.

I had some concerns with respect to novel insight into the molecular mechanism but the authors make a to me convincing case in their rebuttal that the current (and improved) manuscript moves the field forward in a way that justifies publication in a journal like NSMB - the quality of the research is certainly there! Therefore, I am positive towards supporting the current version of the manuscript for publication.

Reviewer #2 (Remarks to the Author):

While, as I previously commented, the first version of this manuscript was already a strong piece, this second version is much improved. The authors clarified multiple points raised by this and other reviewers seriously and thoroughly. They further provided quite reasonable explanations for those points that were not addressed. My previous assessment was that this paper meets the NSMB standards, and I stand by my recommendation to publish.

Finally, the authors produce "Extra Figures" for reviewers, which they not wish to include in their paper unless "the Reviewer feels strongly that they should". While I personally do believe that this data is interesting, the authors should feel absolutely free to decide on the content of their article: reviewers are way too influential as it is. Rest assured that this will be a good paper either way.

Reviewer #3 (Remarks to the Author):

In this revised manuscript, the authors (1) identify a *C. elegans* ortholog of the chromatin protein SPT2, (2) demonstrate sterility defects at high temperatures in knockout animals, and (3) show RNAi defects in SPT2 KO animals. The authors then switch to the U2OS human cell line to perform genomics, including RNA-seq and ATAC-seq. The general conclusion of this section is that some SPT2-bound genes become more accessible when this protein is lost, without much effect on transcription. Finally, the authors provide some experiments suggesting that H3.3 levels are reduced in SPT2 KO and infer from their ATAC data that SPT2 might be a general factor that stabilizes H3 proteins (presumably both H3 and H3.3).

Ultimately, while the manuscript has improved in revision, I don't think that this manuscript represents a significant enough advance for it to be an obvious candidate for NSMB. I have no issue with the data provided for the *C. elegans* section, other than the fact that the authors do not provide any mechanistic insight into the fertility defects or the RNAi defect. Moving into human U2OS cells (a cell line that lacks the H3.3 chaperone that deposits H3.3 at repetitive elements, ATRX) does not provide any clarity on SPT2 function that might feed back to the *C. elegans* study. Very few genes that are bound by SPT2 require it to maintain normal transcription levels. In their model, the authors conflate their H3.3 data to imply that it will also be true more globally for all H3 proteins, but this may not be the case. So, we are left again with a number of generic observations that do not greatly advance our understanding of how SPT2 functions on chromatin.

Author Rebuttal, first revision:**Response to Reviewers**

We again thank the Reviewers for carefully assessing our revised manuscript, and we are glad to hear that we have now satisfied the majority of Reviewers' comments. We provide below a response to the last points raised by Reviewer #3.

Reviewer #1:**Remarks to the Author:**

The authors went to great length in improving their manuscript following up the suggestion of the reviewers (including mine) via substantial experimentation, of which the data is now included.

I had some concerns with respect to novel insight into the molecular mechanism but the authors make a to me convincing case in their rebuttal that the current (and improved) manuscript moves the field forward in a way that justifies publication in a journal like NSMB - the quality of the research is certainly there! Therefore, I am positive towards supporting the current version of the manuscript for publication.

Reviewer #2:**Remarks to the Author:**

While, as I previously commented, the first version of this manuscript was already a strong piece, this second version is much improved. The authors clarified multiple points raised by this and other reviewers seriously and thoroughly. They further provided quite reasonable explanations for those points that were not addressed. My previous assessment was that this paper meets the NSMB standards, and I stand by my recommendation to publish.

Finally, the authors produce "Extra Figures" for reviewers, which they not wish to include in their paper unless "the Reviewer feels strongly that they should". While I personally do believe that this data is interesting, the authors should feel absolutely free to decide on the content of their article: reviewers are way too influential as it is. Rest assured that this will be a good paper either way.

Reviewer #3:**Remarks to the Author:**

In this revised manuscript, the authors (1) identify a *C. elegans* ortholog of the chromatin protein SPT2, (2) demonstrate sterility defects at high temperatures in knockout animals, and (3) show RNAi defects in SPT2 KO animals. The authors then switch to the U2OS human cell line to perform genomics, including RNA-seq and ATAC-seq. The general conclusion of this section is that some SPT2-bound genes become more accessible when this protein is lost, without much effect on transcription. Finally, the authors

provide some experiments suggesting that H3.3 levels are reduced in SPT2 KO and infer from their ATAC data that SPT2 might be a general factor that stabilizes H3 proteins (presumably both H3 and H3.3).

Ultimately, while the manuscript has improved in revision, I don't think that this manuscript represents a significant enough advance for it to be an obvious candidate for NSMB. I have no issue with the data provided for the C. elegans section, other than the fact that the authors do not provide any mechanistic insight into the fertility defects or the RNAi defect. Moving into human U2OS cells (a cell line that lacks the H3.3 chaperone that deposits H3.3 at repetitive elements, ATRX) does not provide any clarity on SPT2 function that might feed back to the C. elegans study. Very few genes that are bound by SPT2 require it to maintain normal transcription levels. In their model, the authors conflate their H3.3 data to imply that it will also be true more globally for all H3 proteins, but this may not be the case. So, we are left again with a number of generic observations that do not greatly advance our understanding of how SPT2 functions on chromatin.

We thank the Reviewer for pointing out that U-2 OS cells carry DAXX-ATRX mutations. DAXX-ATRX have been shown to regulate H3.3 deposition mainly in heterochromatin (PMID 20504901, 20651253, 20211137) and we therefore think that these mutations are unlikely to affect how SPT2 regulates chromatin structure in actively transcribed regions. However, we agree that this remains to be formally proven, and we have therefore added a sentence to discuss this important point at the end of our Result section.

SPT2 has been identified as an interactor of both H3.1 and H3.3 (PMID 36868228) in human cells. Moreover, as we showed in Extra Figure 1 in our previous Response to Reviewers, SPT2 siRNA knock-down reduces the levels of total and chromatin-bound new histone H3.1, not just H3.3. While we are only presenting the data for Reviewers, these experiments show that SPT2 can promote the stability/deposition of more histone H3 variants than just H3.3, and we therefore believe that we are not conflating our data. We have added a sentence in our Discussion to describe this.

Final Decision Letter:

Message 14th Dec 2023

:

Dear Professor Rouse,

We are now happy to accept your revised paper "The histone chaperone SPT2 regulates chromatin structure and function in Metazoa" for publication as an Article in Nature

Structural & Molecular Biology.

As soon as your article is published, you can generate your shareable link by entering the DOI of your article here: `http://authors.springernature.com/share`. Corresponding authors will also receive an automated email with the shareable link

Your paper will be published online soon after we receive proof corrections and will appear in print in the next available issue. You can find out your date of online publication by contacting the production team shortly after sending your proof corrections.

Please note that *Nature Structural & Molecular Biology* is a Transformative Journal (TJ). Authors may publish their research with us through the traditional subscription access route or make their paper immediately open access through payment of an article-processing charge (APC). Authors will not be required to make a final decision about access to their article until it has been accepted. <https://www.springernature.com/gp/open-research/transformative-journals> Find out more about Transformative Journals

Sincerely,

Dimitris Typas
Associate Editor
Nature Structural & Molecular Biology
ORCID: 0000-0002-8737-1319